# Current challenges of implementing anthropogenic land-use and land-cover change in models contributing to climate change assessments

Reinhard Prestele[1], Almut Arneth[2], Alberte Bondeau[3], Nathalie de Noblet-Ducoudré[4], Thomas A.M. Pugh[2,5], Stephen Sitch[6], Elke Stehfest[7], and Peter H. Verburg[1,8]

[1]Environmental Geography Group, Department of Earth Sciences, Vrije Universiteit Amsterdam, De Boelelaan 1087, 1081 HV Amsterdam, The Netherlands
[2]Karlsruhe Institute of Technology, Department of Atmospheric Environmental Research (IMK-IFU), Kreuzeckbahnstr. 19, 82467 Garmisch-Partenkirchen, Germany
[3]Institut Méditerranéen de Biodiversité et d'Écologie marine et continentale, Aix-Marseille Université, CNRS, IRD, Avignon Université, Technopôle Arbois-Méditerranée, Bâtiment Villemin, BP 80, 13545 Aix-en-Provence CEDEX 4, France
[4]Laboratoire des Sciences du Climat et de l'Environnement, 91190 Gif-sur-Yvette, France
[5]School of Geography, Earth & Environmental Science and Birmingham Institute of Forest Research, University of Birmingham, Birmingham, B15 2TT, UK
[6]School of Geography, University of Exeter, Exeter, UK
[7]PBL Netherlands Environmental Assessment Agency, P.O. Box 303, 3720 AH Bilthoven, The Netherlands
[8]Swiss Federal Research Institute WSL, Zürcherstr. 111, CH-8903 Birmensdorf, Switzerland

*Correspondence to*: Reinhard Prestele (reinhard.prestele@vu.nl)

**Abstract.** Land-use and land-cover change (LULCC) represents one of the key drivers of global environmental change. However, the processes and drivers of anthropogenic land-use activity are still overly simplistically implemented in Terrestrial Biosphere Models (TBMs). The published results of these models are used in major assessments of processes and impacts of global environmental change, such as the reports of the Intergovernmental Panel on Climate Change (IPCC). Fully coupled models of climate, land use and biogeochemical cycles to explore land use – climate interactions across spatial scales are currently not available. Instead, information on land use is provided as exogenous data from the land-use change modules of Integrated Assessment Models (IAMs) to TBMs. In this article, we discuss, based on literature review and illustrative analysis of empirical and modeled LULCC data, three major challenges of this current LULCC representation and their implications for land use – climate interaction studies: (I) provision of consistent, harmonized, land-use time series spanning from historical reconstructions to future projections while accounting for uncertainties associated with different land-use modeling approaches, (II) accounting for sub-grid processes and bi-directional changes (gross changes) across spatial scales and (III) the allocation strategy of independent land-use data at the grid cell level in TBMs. We discuss the reasons that hamper the development of improved land-use representation that sufficiently accounts for uncertainties in the land-use modeling process. We propose that LULCC data-provider and –user communities should engage in the joint development and evaluation of enhanced LULCC time series, which account for the diversity of LULCC modeling, and increasingly include empirically based information about sub-grid processes and land-use transition trajectories, to improve the representation of land use in TBMs.

Moreover, we suggest to concentrate on the development of integrated modeling frameworks that may provide further understanding of possible land-climate-society feedbacks.

**Keywords.** land-climate interaction, gross transitions, land-use allocation, Earth System model, global vegetation model, land-use harmonization

## 5   1 Introduction

Anthropogenic land-use and land-cover change (LULCC; for a list of acronyms used in the paper see Supplement S0) is a key cause of alterations in the land surface (Ellis, 2011; Ellis et al., 2013; Turner et al., 2007), with manifold impacts on biogeochemical and biophysical processes that influence climate (Arneth et al., 2010; Brovkin et al., 2004; Mahmood et al., 2014; McGuire et al., 2001; Sitch et al., 2005), and affect food security (Hanjra and Qureshi, 2010; Verburg et al., 2013), fresh

water availability and quality (Scanlon et al., 2007), as well as biodiversity (Newbold et al., 2015). Hence, LULCC is now being increasingly included in Terrestrial Biosphere Models (TBMs), including Dynamic Global Vegetation Models (DGVMs) and Land Surface Models (LSMs) (Fisher et al., 2014), to quantify historical and future climate impacts both in terms of biophysical (surface energy and water balance) and biogeochemical variables (carbon and nutrient cycles) (Le Quéré et al., 2015; Luyssaert et al., 2014; Mahmood et al., 2014). For example, LULCC has been estimated to act as a strong carbon source

since pre-industrial times (Houghton et al., 2012; Le Quéré et al., 2015; McGuire et al., 2001). Livestock husbandry, rice cultivation, and the large-scale application of agricultural fertilizers further contributed to the increase in atmospheric $CH_4$ and $N_2O$ concentration (Davidson, 2009; Zaehle et al., 2011), turning the land to a potential net source of greenhouse gases to the atmosphere (Tian et al., 2016). Local and regional observational studies suggest impacts of LULCC on biophysical surface properties, e.g. surface albedo and water exchange, eventually affecting temperature and precipitation patterns (Alkama and

Cescatti, 2016; Pielke et al., 2011).

TBMs have been originally designed to study the interactions between natural ecosystems, biogeochemical cycles and the atmosphere. The short history of implementing land-use change in TBMs (~10 years; Canadell et al., 2007), along with the need to include external data (e.g., maps of global cropland or pasture distribution) to represent land-use change, have led to several issues that complicate the quantification of land-use change impacts on climate and biogeochemical cycles using

TBMs. For example, carbon fluxes related to land-use change that increase the atmospheric concentration of greenhouse gases are the largest source of uncertainty in the global carbon budget (Ballantyne et al., 2015; Le Quéré et al., 2015). Similarly, biophysical impacts of land-use change on climate are not yet sufficiently understood and quantified (Pielke et al., 2011). The lack of process understanding and reliable quantification of impacts can be attributed to a separated history of land-use research and land-cover research and the current 'offline' coupling of different models, where external land-use information from

Integrated Assessment Models (IAMs) or dedicated land-use change models (LUCMs) is imposed on the natural vegetation scheme of TBMs. This current land-use representation is, amongst others, sensitive to the definition of individual land-use categories (e.g., what exactly defines a 'pasture'), inconsistencies in the definition of the land-use carbon flux (Pongratz et al.,

2014; Stocker and Joos, 2015), the implementation and parameterization of land use in TBMs (Brovkin et al., 2013; de Noblet-Ducoudré et al., 2012; Di Vittorio et al., 2014; Hibbard et al., 2010; Jones et al., 2013; Pitman et al., 2009; Pugh et al., 2015), the structural differences across IAMs and LUCMs (Alexander et al., 2016; Prestele et al., 2016; Schmitz et al., 2014), and the uncertainty about land-use history (Ellis et al., 2013; Klein Goldewijk and Verburg, 2013; Meiyappan and Jain, 2012).

Currently reported uncertainties of the outputs of land use – climate interaction studies may be underestimated by insufficiently accounting for the aforementioned sources of uncertainty. The current land-use representation therefore requires improvement to narrow down the uncertainty range in reported results of land use – climate studies and eventually increase the confidence level of climate change assessments. Assessments of the global water cycle, freshwater quality, biodiversity and non-$CO_2$ greenhouse gases would also benefit from an improved land-use representation.

The overall objective of this article is to review three important challenges faced in connecting models to assess land use – climate interactions and feedbacks, discuss the underlying mechanisms and constraints that have hampered improved representations until now, and propose pathways to improve the land-use representation. We review recent literature from the land use, land cover, carbon cycle and climate modeling communities and support our arguments by illustrative analysis of satellite land-cover products and outputs of the land-use change model CLUMondo (Van Asselen and Verburg, 2013). Each

of the following sections presents one of the three challenges we identify to be crucial in future land use – climate interaction studies and reviews the issue and its implications for the results of modeling studies, based on previously published literature and in the context of the widely applied Land Use Harmonization (LUH) dataset published by Hurtt et al. (2011). In section 5 we propose pathways to improve the current LULCC representation for each of the challenges and conclude with an outlook on future research priorities.

**2 Challenge I: Spatially explicit, continuous and consistent time series of land-use change**

*2.1 Background and emergence*

Current TBMs require consistent, continuous and spatially explicit time series of land-use change, covering at least the period since the industrial revolution (~1750) to disentangle the contributions of land use and fossil fuel combustion to carbon cycling and radiative forcing (Le Quéré et al., 2015; Shevliakova et al., 2009). Without time series of at least this length, important

legacy fluxes will be missed in the calculations. The application of discontinuous land-use change time series in TBMs to quantify the interactions and feedbacks between land use and climate would lead to large artificially induced changes ('jumps') in land use. Corresponding jumps in carbon and nutrient pools in the transition period would distort legacy fluxes working on decadal to centennial time scale, rendering the simulations useless for the quantification of climate impacts.

However, observational data on LULCC is not available at global scale with the required temporal and spatial resolution,

consistency, and historical coverage (Verburg et al., 2011). Instead, models are utilized to represent global land use and produce the required land-use change time series. Land-use modeling is typically split up into historical backcasting approaches and future scenario modeling. Both forward and backward looking models apply a range of different modeling approaches,

assumptions about drivers and the spatial allocation of land-use changes (National Research Council, 2014; Yang et al., 2014), and are often initialized with different representations of present-day land use (Prestele et al., 2016). Thus, even the models within one community (future or historical) do not provide consistent information on land use and land-use change over time, and a variety of independent datasets at spatially explicit or world region level is provided to the user community (e.g. climate modeling) (see Supplement S1 and Table S1 for the example of the historical data). These historical and future datasets are not connected and consistent in the transition period and entail a variety of uncertainties (Klein Goldewijk and Verburg, 2013) (Figure 1). In consequence, these datasets disagree about the amount and the spatial pattern of land affected by human activity. Moreover, varying detail of classification systems, inconsistent definition of individual categories (e.g., forest or pasture), and individual model aggregation techniques, amplify the discrepancies among models (Alexander et al., 2016; Prestele et al., 2016).

*2.2 Current approach to provide consistent data: The Land Use Harmonization (LUH)*

Large efforts have been undertaken to connect the different sources of land-use data and provide consistent time series for climate modeling applications during the 5th phase of the Coupled Model Intercomparison Project (CMIP5; Taylor et al., 2012) by the Land Use Harmonization project (Hurtt et al., 2011). The resulting dataset (hereafter referred to as LUH data) is commonly used in modeling studies dealing with land use – climate interactions and feedbacks. It has recently been updated for the upcoming 6th Phase of the Coupled Model Intercomparison Project (CMIP6; Eyring et al., 2016; Lawrence et al., 2016) and data for the historical period have been published (hereafter referred to as LUH2). Due to the lack of comprehensive documentation of the updated version at the time this paper was written and as, to our best knowledge, the points we demonstrate using LUH will be still valid with the new product, we primarily refer to the CMIP5 version in the remainder of this paper.

Hurtt et al. (2011) extended their Global Land Use Model (GLM; Hurtt et al., 2006) to produce a consistent time series of land-use states (= fraction of each land-use category in a grid cell) and transitions (= changes between land-use categories in a grid cell) for the time period 1500-2100. The cropland, pasture and wood harvest projections of four IAMs were smoothly connected to the History Database of the Global Environment (HYDE) historical reconstruction of agricultural land use (Klein Goldewijk et al., 2011) and historical wood harvest estimates by applying the decadal spatial patterns from the projections to the HYDE map of 2005 (Figure 1). This harmonization process tries to conserve the original patterns, rate and location of change as much as possible, and to reduce the differences between the models due to definition of cropland, pasture and wood harvest. To achieve the final harmonized time series and explicit transitions, the pre-processed land-use time series are used as input into the GLM model and constrained by further data and assumptions about the occurrence of shifting cultivation, the spatial pattern of wood harvest, priority of the source of agricultural land and biomass density (Hurtt et al., 2011). The harmonization ensured for the first time consistent land-use input for climate model intercomparisons and thus facilitated the implementation of anthropogenic impact on the land in climate models. Beyond this inarguable success, several uncertainties are to date not, or only partially, addressed in the LUH data. In the following section we discuss the main uncertainties and

how they may propagate into TBMs, impacting the amplitude, and possibly even the sign of land-use interactions and feedbacks.

*2.3 Open issues in the LUH data and their implications for climate change assessments*

The first major uncertainty of the LUH data evolves from the exclusive consideration of the HYDE baseline dataset for the historical period. The HYDE reconstruction is erroneously regarded as observational data, rather than as model output accompanied by various sources of uncertainty (Klein Goldewijk and Verburg, 2013). Importantly, the LUH2 data will additionally include the HYDE low and high estimates of land use for the historical period (Lawrence et al., 2016). However, alternative spatially explicit reconstructions have been proposed (Kaplan et al., 2010; Pongratz et al., 2008; Ramankutty and

Foley, 1999) (see Supplement S1 and Table S1 for additional information on these reconstructions), and shown to differ substantially both in terms of the total cultivated area and spatial pattern over time (Meiyappan and Jain, 2012). These differences originate in the scarcity of historical input data (i.e., mainly population estimates) for historical times, the assumption about the functional relationship between population density and land use (e.g., linear or non-linear) and the allocation scheme used to distribute regional or national estimates of agricultural land to specific grid cell locations (Klein

Goldewijk and Verburg, 2013).

The uncertainty about land-use history has several implications for land use – climate interactions (Brovkin et al., 2004). For instance, Meiyappan et al. (2015) found the difference in cumulative land-use emissions among three historical reconstructions for the 21th century modeled by one TBM to be about 18 PgC or ~11 % of the mean land-use emission. Another study, using three commonly-used net land-use datasets in one TBM, revealed differences of about 20 PgC or ~9 % of the mean land-use

emission since 1750 (Bayer et al., 2017). Jain et al. (2013) further found contrasting trends in land-use emissions at regional scale during the past three decades, which originate in different amounts and rates of land-use change in different realizations of historical land use. Further, as biophysical climate impacts of land use are known to be substantial, especially on a regional scale (Alkama and Cescatti, 2016; Pielke et al., 2011; Pitman et al., 2009), an inappropriate representation of the uncertainty about land-use history is likely to affect model outcomes regarding changes in local to regional climate. Using the HYDE

reconstruction exclusively implies high confidence about land-use history in many large scale assessments and comparison studies (Kumar et al., 2013; Le Quéré et al., 2015; Pitman et al., 2009), which is in fact lacking. As a result, important uncertainties are being excluded from climate change mitigation and adaptation policies developed based on these studies (Mahmood et al., 2015).

Second, large inconsistencies exist between estimates of present-day land use. The LUH approach does not consider the

differences between different data regarding the current state of land use as it connects the future projections exclusively to the HYDE end map (Figure 1). The present-day starting maps of historical reconstructions and future projections are based on maps derived from the integration of remotely sensed land-cover maps and (sub-)national statistics of land use (e.g., Erb et al., 2007; Fritz et al., 2015; Klein Goldewijk et al., 2011; Ramankutty et al., 2008). The land-cover maps in turn disagree about extent and spatial pattern of agricultural land (Congalton et al., 2014; Fritz et al., 2011) due to both inconsistent definitions of

individual land-use and land-cover categories (e.g., Sexton et al., 2015) and difficulties in identifying them from the spectral response (Friedl et al., 2010). These differences propagate into the starting maps of the various land-use change models, including the IAMs providing data for the LUH (Prestele et al., 2016). Removing these differences can result in substantial deviations of the seasonal and spatial pattern of surface albedo, net radiation and partitioning of latent and sensible heat flux

(Feddema et al., 2005) and affect carbon flux estimates proposed by TBMs across spatial scales (Quaife et al., 2008).

Finally, the future projections used in the LUH are provided by different IAMs, whereby each of them represents an individual scenario of the four representative concentration pathways (RCP) in CMIP5 or the five shared socioeconomic pathways (SSP) in CMIP6 (O'Neill et al., 2015; van Vuuren et al., 2011). These are referred to as 'marker scenarios' in case of the SSPs. A 'marker scenario' entails the implementation of a SSP by one IAM that was elected to represent the characteristics of the

qualitative SSP storyline best, while additional implementations of the same SSP in other IAMs are 'non-marker scenarios' (Popp et al., 2016; Riahi et al., 2016). Alternative RCP or SSP implementations were not considered in LUH. Land-use change model intercomparisons and sensitivity studies, however, indicate that the uncertainty range emerging from different assumptions in the models, input data, and spatial configuration substantially impacts the model results (Alexander et al., 2016; Di Vittorio et al., 2016; Schmitz et al., 2014). Due to the large range across model outcomes per scenario, the problems of

using 'marker scenarios' from different models are evident. However, no better alternative to this approach seems to be currently available, and representing uncertainty across models is valuable (Popp et al., 2016). Model comparisons further revealed that while land-use change models represent the future development of cropland area more consistently, the representation of pastures and forests (if modeled) is poor. For example, the projections of 11 IAMs and LUCMs show large variations in pasture areas in 2030 for many world regions (Figure 2, background map; Supplement S2.1). These projections

were based on a wide range of scenarios, and thus variation in outcomes was to be expected (Prestele et al., 2016). The variation attributed to the difference in model structure exceeds the variation due to different scenarios in most regions (Figure 2, bar plots), while the main part of the variation relates to the different starting points of the models, i.e. deviation from FAO pasture areas in the year 2010. This implies that in many cases the different land-use projections actually do not represent different outcomes resulting from different scenario assumptions, but rather differences between land-use data input used to calibrate

the models and the implementation of drivers and processes in the models. Consequently, differences in future climate impacts of land use are likely also affected by the structural differences across land-use change models.

## 3 Challenge II: Considering gross land-use changes

### 3.1 Background and emergence

Typically, net land-use changes are applied in TBMs. Net land-use changes refer to the summed grid-cell difference in land-

use categories between two subsequent time steps at a certain spatial and temporal resolution. Gross change representations provide additional information about land-use changes on a sub-grid scale. The total area in a grid cell, which has been affected by change, can be calculated by the sum of all individual changes (i.e., area gains and area losses). Gross changes have been

shown to be substantially larger than net changes due to bi-directional change processes happening at the same time step (Fuchs et al., 2015a; Hurtt et al., 2011) that are obscured in net change representations. For example, 20 km$^2$ cropland at time $t_1$ and 40 km$^2$ at time $t_2$ within a grid cell does not necessarily mean that this change resulted from clearing exactly 20 km$^2$ of forest. Equally plausible would be clearance of forest of larger spatial extent, while at the same time also a certain amount of cropland

was abandoned, resulting in the same net areal change.

'Gross changes' are not consistently defined across communities. Commonly, shifting cultivation (mostly occurring in parts of the tropics nowadays), and cropland-grassland dynamics (i.e., the bi-directional process of cropland expansion and abandonment) are referred to as gross changes (Fuchs et al., 2015a; Hurtt et al., 2011). Moreover, in the carbon cycle and climate modeling communities, wood harvest (in addition to forest cleared for agricultural land) is sometimes included in gross

changes (Hurtt et al., 2011; Stocker et al., 2014; Wilkenskjeld et al., 2014). A more general definition would include all area changes (i.e., gains and losses across all categories represented in a product) that are not depicted in land-use change products (Fuchs et al., 2015a). The larger the averaging unit (be it in terms of grid cell or time), the greater the discrepancy between gross and net changes becomes. Re-gridding of high-resolution (e.g., 5 arc minutes) land-use information to the TBM grid (~0.5 degree) thus entails additional loss of information on land-use transitions unless gross changes are considered.

These sub-grid dynamics have been shown to be of importance when modeling change of carbon and nutrient stocks in response to land-use change in recent TBM studies (Bayer et al., 2017; Fuchs et al., 2015b; Stocker et al., 2014; Wilkenskjeld et al., 2014). For example, Bayer et al. (2017) found the global cumulative land-use carbon emission to be ~33 % higher over the time period 1700-2014. Stocker et al. (2014) likewise report increased carbon emissions in recent decades and for all RCPs when accounting for shifting cultivation and wood harvest. Similarly, Wilkenskjeld et al. (2014) found a 60 % increase in the

annual land-use emission for the historical period (1850-2005) and a range of 16-34 % increase for future scenarios, when accounting for gross changes. Recently, Arneth et al. (2017) demonstrated uniformly larger historical land-use change carbon emissions across a range of TBMs when shifting cultivation and wood harvest were included, which has implications for understanding of the terrestrial carbon budget as well as for estimates of future carbon mitigation potential in regrowing forest. Except for such sensitivity studies, gross changes have hardly been considered so far in land use – climate interaction studies

(a notable exception being Shevliakova et al., 2013), mainly due to two reasons. First, gross change estimates have not been available until recently. Deriving estimates of historical and future gross change is a difficult task, since gross changes vary with spatial and temporal scale (Fuchs et al., 2015a), i.e. they are dependent on the scale of the underlying net change product used for modeling and to what extent gross change processes are included in the individual land-use change models. Second, the implementation of bi-directional changes below the native model grid often entails substantial technical modification to

TBM structure, meaning that many TBMs are currently not ready to include information on gross changes or only started recently to include it.

*3.2 Example: Gross changes due to re-gridding in the CLUMondo model*

To illustrate the amount of land-use and land-cover changes that might be missed in net representations, we conducted an analysis based on the output of a dedicated high-resolution LUCM (CLUMondo; 5 arc minutes spatial resolution; Eitelberg et al., 2016; Van Asselen and Verburg, 2013). We tracked all changes between five land-use and land-cover categories (cropland,

pasture, forest, urban, and bare) at the original resolution over the time period 2000 to 2040. Aggregating to ca. 0.5 degree resolution allowed to differentiate the gross area from the net area affected by change (see Supplement S2.2 for methodological details). The results, shown in Figure 3, indicate that gross changes are substantially higher than net changes all over the globe, including the temperate zone and high latitudes. It has to be noted that Figure 3 is only based on one realization of a single LUCM, i.e. not necessarily representing the full extent and spatial pattern of global scale gross changes. The analysis only

depicts the loss of information while re-gridding from 5 arc minutes to 0.5 degree resolution. Thus, bi-directional changes below the spatial resolution of the original data are still not captured.

*3.3 Current approaches to provide gross change information: LUH and analysis of empirical data*

To provide estimates of gross change, the land-use change modeling community currently follows two different approaches.

First, Hurtt et al. (2011), within the framework of LUH, propose a matrix that provides explicit transitions between cropland, pasture, urban, and natural vegetation. Sub-grid scale information is added to net transitions (that are derived from historical or projected land-use data and referred to as 'minimum transitions') through assumptions about the extent of shifting cultivation practices and the spatial pattern of wood harvest. In each grid cell, where shifting cultivation appears according to a map of Butler (1980), an average land abandonment rate is added to each transition from and to agricultural land. In LUH2 an updated

shifting cultivation estimate based on the analysis of Landsat imagery will be included and replace the aforementioned simple assumption (Lawrence et al., 2016). Wood harvest is regarded as gross change, if the wood harvest demand from statistics (historical) or IAMs (future) is not met by deforestation for agricultural land in the net transitions or the GLM model is run in a configuration where deforestation for agricultural land is not counted towards wood harvest demand.

The second approach derives gross/net ratios and a transition matrix directly from empirical data such as historical maps or

high-resolution remote sensing products. These ratios can be subsequently applied to existing historical or future net representations to provide estimates of additional area affected by change (Fuchs et al., 2015a).

*3.4 Open issues in the current approaches*

The LUH gross transitions account for some aspects of gross changes. However, the values are dependent on what one includes

in the definition of gross changes and are based on overly simplistic assumptions. Most of the gross transitions appear in parts of the tropics, where shifting cultivation is assumed to be an important agricultural practice (Bayer et al., 2017; their Figure S1). Gross changes outside of these areas are mainly related to wood harvest, i.e. the (additional) area deforested to meet external wood harvest demands. Although these are regarded as gross changes in some literature (e.g., Hurtt et al., 2011; Stocker et al., 2014), we argue that wood harvest not leading to an actual areal change of land-cover (e.g., forest to cropland)

should be rather referred to as land management than gross change. Excluding wood harvest from the LUH data restricts the occurrence of gross changes to the areas of shifting cultivation. However, our analysis of CLUMondo output (Figure 3), along with the European analysis of Fuchs et al. (2015a), suggests substantial amounts of gross changes (below the 0.5 degree LUH grid) also in the temperate zone and the high latitudes. Consequently, the LUH approach heavily depends on the resolution of the original land-use data (provided by IAMs or historical reconstructions) and their ability to represent land-use change dynamics on a sub-grid scale.

The data-based approach avoids the process uncertainty which hinders high-resolution model projections of land use, but is limited to the time period where empirical data through remote sensing is available. Additional sources such as historical land-use and land-cover maps and statistics (Fuchs et al., 2015c) may contribute to cover longer time periods, although with limited spatio-temporal resolution and spatial coverage, and an associated increase in uncertainty. It is thus difficult to develop multi-century reconstructions or future scenarios including gross changes using data-based approaches, since the derived gross/net ratios are only valid for periods of data coverage and are expected to change over time (Fuchs et al., 2015a).

## 4 Challenge III: Allocation of managed land in TBMs

### 4.1 Background and emergence

The LSMs in most Earth System Models (ESMs) in CMIP5 treated the land surface as a static representation of current land-use and land-cover distribution typically derived from remote sensing products (Brovkin et al., 2013; de Noblet-Ducoudré et al., 2012). DGVMs, some of which are incorporated in the land surface component of ESMs, were originally designed to model potential natural vegetation as a dynamic function of monthly climatology, bioclimatic limits, soil type and the competitiveness of different wood- or grass-shaped plant functional types (PFTs) (Prentice et al., 2007). Thus, the early TBMs were not able to sufficiently account for anthropogenic activity on the land surface and consequently the impact of land use on climate and biogeochemical cycles (Flato et al., 2013). However, over the last decade, representation of human land-cover change, and also some land management aspects, have increasingly been added to these models, albeit with levels of complexity which vary from crops as grassland to more detailed agricultural representations (Bondeau et al., 2007; Le Quéré et al., 2015; Lindeskog et al., 2013). Crop functional types (CFTs) and management options have been introduced in some models, explicitly parameterizing the phenology, biophysical and biogeochemical characteristics of major crop types, and distinguishing important management options such as irrigation, fertilizer application, occurrence of multiple cropping or processing of crop residues (Bondeau et al., 2007; Lindeskog et al., 2013). However, since TBMs do not include representations of human activity as a driver of changes on the land surface, information about the extent and exact location of managed land is required from external data sources such as IAMs or LUCMs.

IAMs and LUCMs usually provide land-cover information (e.g., forest, grassland, shrubland) along with land-use information (e.g., cropland and pasture). However, as modeling changes in natural vegetation type is one of the primary functions of many TBMs, only land-use information has been used in the LUH (Hurtt et al., 2011). Hence, TBM modelers have to decide in

which way the natural vegetation in a grid cell has to be reduced (in case of expansion of managed land) or increased (in case of abandonment of managed land). This has resulted in a range of different strategies, which we show as an illustration in Table 1 for a non-exhaustive list of models. The decision is important as it impacts the distribution of the natural vegetation in a grid cell, as well as the mean length of time that land has been under a particular use, with consequences for both the biogeochemical and biophysical properties (Reick et al., 2013). For example, new cropland expanding on forest would lead to a large and relatively rapid loss of ecosystem carbon due to deforestation, while cropland expanding on former grassland would have a less immediate impact on ecosystem carbon stocks due to the long time lag (years to centuries) for the resulting changes in soil carbon to be realized (Pugh et al., 2015). Likewise, the albedo and partitioning of energy differs strongly between forest and grassland land covers (Mahmood et al., 2014; Pielke et al., 2011). In the following sections we illustrate, based on literature review and analysis of empirical and modeled data, that the previously described simple allocation algorithms, applied globally, within TBMs do not account well for the spatio-temporal variation of land-use and land-cover change.

## 4.2 Spatial heterogeneity of cropland transitions – empirical evidence

Table 2 summarizes dominant sources of cropland expansion for several world regions and demonstrates the heterogeneity in the spatial pattern of expanding agriculture. For Europe, the CORINE land-cover product (Bossard et al., 2000) indicates over two consecutive time periods (1990-2000, 2000-2006) shrubland systems to be the main source of expanding agricultural land, followed by low productivity grasslands and forests (Figure 4a). In contrast, over a similar time period, the NLCD (Homer et al., 2015) for the USA shows low productivity grasslands as the dominant source of new croplands, while pastures are predominantly converted from forest or shrubland systems and grasslands only account for around 20 % of new pastures (Figure 4b). A large-scale study by Graesser et al. (2015) covering Latin America and based on the interpretation of MODIS images for the time period 2001-2013, identified the dominant trajectory of forests being first converted to pastures and subsequently to cropland. They show, however, varying patterns on national and ecoregion scale. This regional variation is also emphasized by Ferreira et al. (2015), who describe a satellite-based transition matrix as input for a modeling study for different states in Brazil. They do not distinguish non-forest natural vegetation such as the Cerrado systems, which might be another important source for agricultural land (Grecchi et al., 2014). A study conducted by Gibbs et al. (2010) investigating agricultural expansion in the tropics in the 1980s and 1990s based on data from Food and Agriculture Organization of the United Nations (2000) (i.e., areas with less than 10 % forest cover are not considered) concludes that more than 80 % of new agricultural land originates from intact or degraded forests. Gibbs et al. (2010) further found large variability in agricultural sources across seven major tropical regions, e.g., substantially higher conversions from shrublands and woodlands to agricultural land in South America and East Africa. Grasslands have been detected as the main source of agricultural land in Northern China, e.g., by Li (2008), Liu et al. (2009) and Zuo et al. (2014), while in the Yangtze River basin woodlands contribute most (Wu et al., 2008) (Table 2). All the mentioned studies indeed combine different approaches to derive changes, cover different time periods and are not representative of current agricultural change hotspots (Lepers et al., 2005). However,

this kind of aggregated analysis already indicates that the spatial pattern of agricultural change dynamics varies across world regions and a single global algorithm to replace natural vegetation by managed land in TBMs is likely to be overly simplistic.

*4.3 Example: Spatial heterogeneity of cropland transitions in the CLUMondo model*

As it is not possible to compare the land-use allocation strategies of TBMs with historical change data at global scale due to the lack of accurate global land-use and land-cover products (though  products with higher resolution (up to ~30m), more frequent temporal coverage, and increasing thematic detail are just emerging; Ban et al., 2015), we additionally tested to what extent cropland expansion simulated by the land-use change model CLUMondo (Eitelberg et al., 2016; Van Asselen and Verburg, 2013) represents one or more of the simplified algorithms currently considered in TBMs (Table 1).

CLUMondo models the spatial distribution of land systems over time, instead of land use and land cover directly. Land systems are amongst others characterized by a mosaic of land use and land cover within each grid cell. The land systems are allocated to the grid in each time step 'based on local suitability, spatial restrictions, and the competition between land systems driven by demands for different goods and services' (Eitelberg et al., 2016; Van Asselen and Verburg, 2013). Thus, the determination of the source land use or land cover upon cropland expansion can be interpreted as a complex algorithm taking into account

external demands, the land-use distribution of the previous time step, local suitability in a grid cell and neighborhood effects (i.e., cropland expansion in a grid cell also depends on the availability of suitable land in the surrounding grid cells). This strategy differs from the one in TBMs in a way that not one simple rule is applied to each grid cell equally, but accounts for the spatial heterogeneity of drivers of land-use change.

In order to compare the sources of cropland expansion in CLUMondo to the globally applied rules in TBMs, we reclassified

the outputs of the same CLUMondo simulation utilized in section 3.2 (FAO3D; Eitelberg et al.; 2016) according to their dominant land-use or land-cover type to derive transitions (Table S6) and classified the changes  within each ca. 0.5 x 0.5 degree grid cell as either grassland first, forest first, proportional, or a complex reduction pattern (Table 3; Figure S2-3 and additional explanation in Supplement S2.4). Additionally, a grid cell was labeled 'undefined', if grassland or forest was not available in the source map.

Figure 5 shows the results of this analysis for decadal time steps between 2000 and 2040. Based on the CLUMondo data it is clear that a single simple algorithm does not account for the temporal and spatial heterogeneity of cropland expansion in a detailed land-use change model. The majority of grid cells with substantial cropland expansion (> 10 % of grid cell area) where we could detect an algorithm (i.e., the grid cell was not classified 'undefined') show a complex reduction pattern of the remaining land-use and land-cover categories, i.e., any algorithm applied to these grid cells in a TBM could be seen as equally

good or bad. The remaining grid cells account only for 24-27 % globally. Moreover, the spatial distribution of grid cells that are classified to the same algorithm is very heterogeneous and changing over time. It has to be noted that this analysis builds on only one realization of one LUCM and results may differ if using another data source in terms of overall cropland expansion and the exact grid cell location of changes. However, the analysis does not aim at identifying the exact location of a particular algorithm but rather emphasizing the heterogeneous pattern of cropland expansion.

*4.4 Current approach to provide allocation information: the transition matrix*

In CMIP5, most ESMs implemented a proportional reduction of natural vegetation rather arbitrarily due to reasons of simplicity or internal model constraints; others converted grassland preferentially and/or treated croplands differently from pastures upon transformation (de Noblet-Ducoudré et al., 2012). However, none of them depicts the complex interplay of biophysical and socioeconomic parameters leading to a heterogeneous spatial pattern of land-use change within the coarse grid-resolution used in ESMs. As we have shown in the previous sections, empirical evidence and land-use change models suggest that this complexity is poorly represented by simplistic, globally applied, algorithms. The efforts of LUH thus included the provision of a transition matrix, i.e. the explicit identification of source and target categories between agricultural land and natural vegetation at the grid cell level. For each annual time step, the exact fraction of a grid cell that has changed from one land-use category to another is determined, thus providing the option to replace the simple allocation options by detailed information about land-use transitions within each grid cell (Hurtt et al., 2011).

*4.5 Open issues of transition matrices*

The provision of transition matrices, however, generally brings up a sequence of additional challenges, which we illustrate using the example of LUH in the following. First, the decision which land-cover type should be replaced upon cropland or pasture expansion (or introduced in case of abandonment) is in fact only shifted from the TBM community to the IAM/LUCM community and the accuracy of the transitions are heavily dependent on the sophistication (i.e., knowledge about and depiction of land-use change drivers and processes at the grid scale) of the land-use allocation algorithm in the original model providing the land-use data. Many current models simulate land-use changes on a world-region level and downscale these aggregated results to the required grid cell level (Hasegawa et al., 2016; Schmitz et al., 2014). In the LUH approach these downscaled data are used to derive the minimum transitions between agricultural land use and natural vegetation. Additional assumptions are made to allocate changes in land-use states to explicit transitions, not accounting for the spatial and temporal heterogeneity of the multiple drivers of land-use change. For example, urban expansion is applied proportionally to cropland, pasture and (secondary) natural vegetation. Upon transitions between natural vegetation and agricultural land, choices in the model configuration have to be made, whether primary or secondary land is converted preferentially. These choices are similar to the grassland or forest first reduction algorithms applied in TBMs.

Moreover, due to the lack of empirical long-term, high-accuracy land-use and land-cover change information and the inconsistencies between agricultural land-use data and land-cover information from satellites, global IAMs and LUCMs are rarely evaluated against independent data (Verburg et al., 2015). It is thus not clear yet to what extent the spatial land-use patterns simulated by these models and provided to LUH represent a good estimate of real past and future land-use changes. In consequence, transitions derived from these modeled time series are necessarily uncertain.

Hence, it is evident that more and improved empirical information on land-use transitions is required to improve land-use change modeling, and to estimate the natural systems at risk under agricultural expansion. However, the specific problem of allocating new agricultural land in DGVMs and LSMs also has a strong model and data-structure component. In many DGVMs, the grass and forest PFTs on non-agricultural land in a grid cell are mostly not considered different systems, but are part of one complex vegetation structure thus not representing spatial-horizontal heterogeneity. Therefore, when agriculture expands into such natural systems, all natural PFTs need to be reduced proportionally. If handled otherwise (i.e., when removing a specific PFT preferentially), the vegetation dynamics would slowly converge again towards the initial PFT mix (if all boundary conditions like climate and soil properties remain unchanged).

For LSMs coupled to ESMs, the situation is slightly more complex. Most ESMs (if not incorporating dynamic vegetation through a DGVM) are using a remote sensing product such as the ESA CCI-LC (ESA, 2014), and a translation to PFTs, e.g., Poulter et al. (2011), as background vegetation map on which agricultural land is imposed. Due to inaccuracies in global remote sensing land-cover products and differences in historical reconstructions (as discussed in section 2), fractions of agricultural land on a grid-scale are subject to differences between the background map and the external land-use dataset. Consequently, the PFT composition outside the prescribed agricultural land can represent either the real heterogeneity in natural vegetation, or represent a mix of natural and anthropogenic land cover due to differences in the datasets. However, these cases are difficult to distinguish and empirically justified transition matrices, together with more accurate present-day land-cover products, would provide a useful tool for reducing uncertainties due to allocation decisions in ESMs.

## 5 Recommendations for improving the current LULCC representation across models

### 5.1 Tackling uncertainties in the harmonization

The Land Use Harmonization (Hurtt et al., 2011) has allowed to include anthropogenic impacts on the land surface for the first time in the CMIP5 climate change assessments. As we have shown in section 2, three major sources of uncertainty, which include the uncertainty about land-use history, inconsistencies in present-day land-use estimates, and structural differences across IAMs and LUCMs, are poorly addressed through the almost exclusive implementation of the LUH dataset within the climate modeling community. A wider range of harmonized time series is therefore likely to substantially influence the outcomes of studies on land use - climate interactions. The actual impact of alternative harmonized time series on carbon cycle (and other ecosystem processes) and climate has never been tested, mainly due to the lack of alternative provision of such products. One would need a multi-model ensemble design to properly account for and disentangle the individual contribution of different historical reconstructions, the multitude of present-day land-use products and varying future land-use change modeling approaches. Different future scenario models would need to be connected to different instances of historical reconstructions, both constrained by different plausible realizations (i.e., based on previously published, peer-reviewed approaches) of current land use and land cover. Such an approach would ensure a comprehensive coverage of the uncertainties

accumulating across temporal and spatial scales prior to feeding land-use data into climate models and allow for testing climate model sensitivity to different realizations of land-cover and land-use information.

The high computational demands of complex ESMs probably do not allow for multiple runs including all the uncertainties in land-use forcing. However, to derive robust results from climate model intercomparisons, a sufficient quantification of

uncertainty in the land-use forcing dataset is urgently required. If this proves impractical through ESM simulations, we recommend to utilize less computational expensive models such as DGVMs and offline LSMs to assess the full range of uncertainty and determine a limited set of simulations, which appears to significantly affect biogeochemical cycles and climate. These can be subsequently used to test the uncertainty range in ESMs.

Simultaneously, we suggest that the land-use and remote-sensing communities should engage to reduce uncertainties in land-

use and land-cover products by:

(1) Developing diagnostics for the evaluation of land-use reconstructions based on satellite data and additional proxy data such as pollen reconstructions (Gaillard et al., 2010) or archeological evidence of early land use (Kaplan et al., 2016).

(2) Developing systematic approaches to evaluate results of land-use change models against independent data sources,

utilizing the full range of high-resolution satellite data (e.g., the Landsat archive and the European Sentinel satellites), reference data obtained from (sub-)national reporting schemes under international policy frameworks (e.g., Kohl et al., 2015) and innovative methods such as volunteered geographic information and crowd-sourcing (Fritz et al., 2012). Although satellite data is also not directly measured empirical data, but goes through a mathematical conversion process prior to a final land-cover product, it can improve representations of present-day land cover. If not yet possible

at the global scale due to the limitations discussed in section 2, we recommend the implementation of regional scale evaluation schemes using smaller scale, high accuracy remote sensing products as a starting point for later integration into global applications.

*5.2 Gross change representations*

The full extent of gross changes is still not well understood (see section 3). Thus, the land-use community should explore high-resolution remote-sensing imagery regarding their ability to derive gross change estimates and improve understanding of sub-grid dynamics which are not yet captured by their models. Regions where driving factors of small-scale land-use change processes are more complex, and not easy to determine due to frequent land-use changes, should receive special attention. Based on such analyses, multi-century reconstructions and projections for climate and ecosystem assessments could be

enhanced for at least the satellite era. As models extend further into the past, the detailed information could be gradually replaced by model assumptions, supported by additional reference data such as historical maps and statistics.

*5.3 Transition matrix from empirical data*

Explicit information of land-use transitions instead of annual land-use states is essential for questions regarding carbon and nutrient cycling. We argue that simple, globally applied, assumptions about these transitions or the shift of the responsibility from TBMs to land-use models may not solve the problem (section 4). Thus, the development of dedicated transition matrices increasingly based on empirical data (as soon as new products emerge) and sophisticated land-use change allocation models,

which account for the spatio-temporal heterogeneity of land-use change drivers is essential.

Simultaneously, TBMs must ensure to use the full detail of information provided by the implementation of explicit transition information in their land modules. Due to internal model structure proportional reduction of PFTs need to be applied in models with internally simulated dynamic vegetation. However, we recommend the utilization of explicit transition information to further evaluate discrepancies between the potential natural vegetation scheme and LULCC data provided by LUCMs and

IAMs.

## 6 Outlook: towards model integration across disciplines

The 'ways forward' listed in the previous section will only be the first stage of a process towards improved LULCC representation in climate change assessments. Rather than improving de-coupled data products and models on an individual basis and connecting them 'offline' through the exchange of files, we argue that land use, land cover and the climate system

need to be studied in an integrated modeling framework. As we have shown in this paper, most of the challenges and related uncertainties originate in the disparate disciplinary treatment of the individual aspects. Although sophisticated models have been developed during the past decades within each community, the current 'offline' coupling seems overly limited, accumulating an increasing level of uncertainty along the modeling chain. Integration of these different types of models, where anthropogenic activity on the land system is considered as an integral part of ESMs, instead of an external boundary condition,

might help to reduce these uncertainties, although it will certainly further complicate the interpretation of model responses. For example, Di Vittorio et al. (2014) report preliminary results of the iESM (Collins et al., 2015), an advanced coupling of an IAM and an ESM, implementing two-way feedbacks between the human and environmental systems, and show how this improved coupling can increase the accuracy of information exchange between the individual model components. In the long term, additionally including behavioral land system models (e.g., agent-based approaches) in the coupling, may provide further

understanding of possible land-climate-society feedbacks (Arneth et al., 2014; Verburg et al., 2015), since the current modeling chain rarely accounts for the complexity of human-environmental relationships and feedbacks (Rounsevell et al., 2014).

## Acknowledgements

The research in this paper has been supported by the European Research Council under the European Union's Seventh Framework Programme project LUC4C (Grant No. 603542), ERC grant GLOLAND (No. 311819) and BiodivERsA project

TALE (No. 832.14.006) funded by the Dutch National Science Foundation (NWO). This research contributes to the Global Land Project (www.globallandproject.org). This is paper number 26 of the Birmingham Institute of Forest Research.

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

**Table 1: Examples of allocation rules at grid cell level to implement agricultural land in different TBMs**

| Model | Land use /cover types | Allocation strategy | Reference |
|---|---|---|---|
| LPJ-GUESS | natural, cropland, pasture | proportional reduction | Lindeskog et al. (2013) |
| HadGEM2-JULES | natural (tree, shrub, grass) cropland, pasture | grassland first | Clark et al. (2011) |
| ORCHIDEE | natural (tree, grass), cropland, pasture | proportional reduction | Krinner et al. (2005) |
| LPJ-mL | natural (tree, grass), cropland, pasture | proportional reduction | Bondeau et al. (2007) |

**Table 2: Case studies and continental scale remote sensing studies that report main sources of agricultural expansion or allow for land cover change detection**

| Region | Temporal coverage | Main source of new cropland | Main source of new pasture | Reference |
|---|---|---|---|---|
| Europe | 1990-2000 / 2000-2006 | | Shrubland / Shrubland* | Bossard et al. (2000) |
| USA | 2001-2006 / 2006-2011 | Grassland / Grassland | Shrubland / Forest | Homer et al. (2015) |
| Latin America | 2001-2013 | Pasture | Forest | Graesser et al. (2015) |
| Northern China | 1989-1999 / 1999-2003 | Grassland / Grassland | - | Li (2008) |
| | 1986-2000 | Grassland | - | Liu et al. (2009) |
| | 1995-2010 | Grassland | - | Zuo et al. (2014) |
| Yangtze River Basin | 1980-2000 | Woodland | - | Wu et al. (2008) |
| Brazil | 1994-2002 | Forest | Forest | Ferreira et al. (2015) |
| Tropics | 1980-2000 | Forest* | | Gibbs et al. (2010) |

* source refers to all new agricultural land, i.e. cropland and pasture combined

**Table 3: Definition of classified algorithms in the CLUMondo exercise (section 4.3). CLUMondo data were preprocessed as described in the text and Supplement S2.4. Each ca. 0.5 x 0.5 degree grid cell was assigned a label according to the distribution of changes seen in the higher resolution (5 arc minute) CLUMondo data. Land types according to the reclassification of CLUMondo land systems shown in Table S6; mosaics refer to a mixture of vegetation within a grid cell (e.g. forest and grassland).**

| Label | Within a 0.5 x 0.5 degree grid cell… |
|---|---|
| UNDEFINED | …*forest* or *grassland* were not available for conversion to cropland.* |
| UNVEGETATED FIRST | …*urban* or *bare* were converted to cropland, although vegetation was available. |
| FOREST FIRST | …*forest* was predominantly converted to cropland, although *grassland* and *mosaics* were available. |
| GRASSLAND FIRST | …*grassland* was predominantly converted to cropland, although *forest* and *mosaics* were available. |
| PROPORTIONAL | (1) …*mosaics* were predominantly converted to cropland, although *forest* and *grassland* were available <br> (2) …*forest* and *grassland* were converted proportionally to cropland. |
| COMPLEX | …*forest, grassland*, and *mosaics* were simultaneously converted without a preference to one of the classes or proportional reduction. |

* If one of the two classes is not available for conversion, either of the preferential algorithms (unvegetated, forest or grassland first) could be correct, but not executed because of the lack of the source that should be converted 'first'.

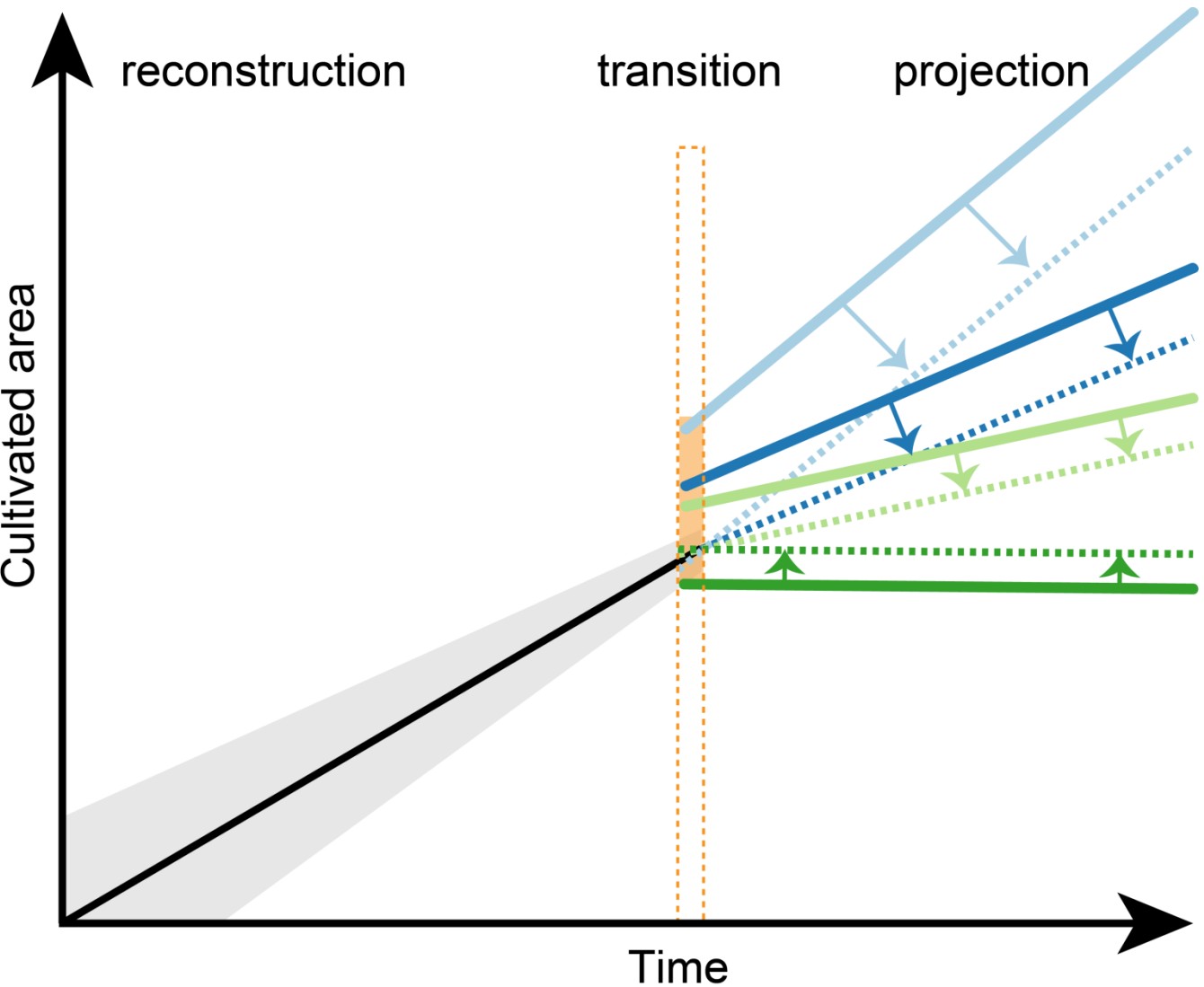

**Figure 1: Simplified scheme of the harmonization process. Future projections from different models (solid colored lines) are smoothly connected (dashed colored lines) to the HYDE historical reconstruction (black line; grey shading represents the uncertainty range of LULCC history). Uncertainty about extent and pattern of current land use and land cover (orange shading) is removed and the total areas of cultivated land projected by the different models are changed.**

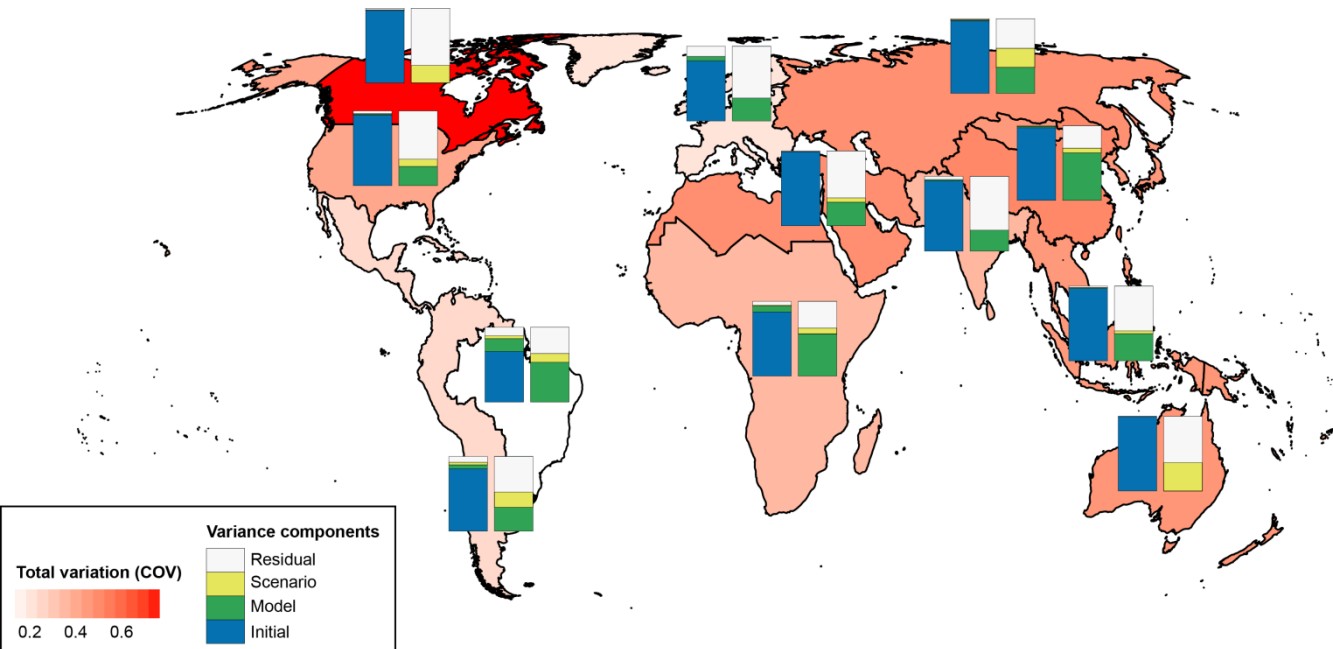

**Figure 2: Variation (expressed as coefficients of variation) of pasture projections for 12 world regions in 2030 (shading of the background map). The left bar plots show the relative contribution (in %) of initial variation (pasture area in relation to values reported by FAOSTAT (2015) for the year 2010), model related variation (model type and spatial configuration) and scenario related variation to the total variation in a region. The right bar plots show the relative contribution (in %) of variance components to the part of total variation that cannot be attributed to initial variation. The figure is based on 11 regional and spatially explicit land-use change models as described in Prestele et al. (2016). Methodological details can be found in Supplement S2.1 (Table S2) and in Alexander et al. (2016).**

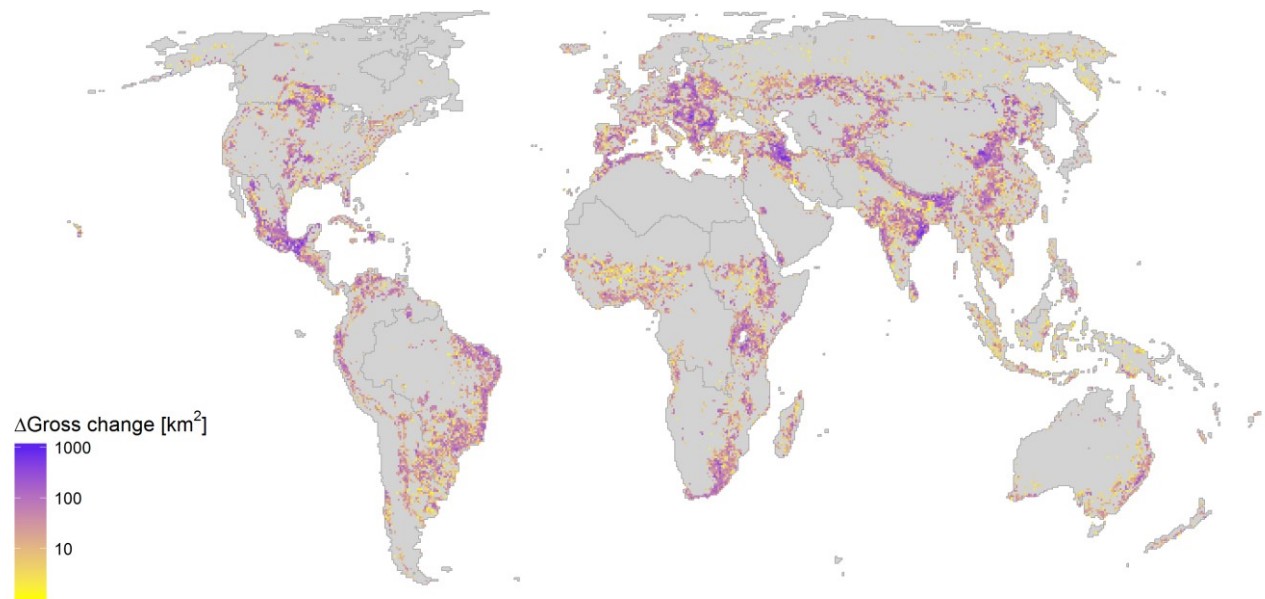

**Figure 3: Difference between gross versus net area affected by change at grid cell level (ca. 0.5 x 0.5 degree) as shown by one realization of a single LUCM (CLUMondo; FAO 3 demand scenario). Areas affected by net or gross change have been accumulated over a 40 year simulation period (2000-2040). Net changes are calculated at ca. 0.5 x 0.5 degree resolution, while gross changes also account for bi-directional changes at the 5 arcminute native CLUMondo resolution (Supplement S2.2; Figure S1). More intense colors indicate a larger difference between the area changed under a net and a gross change view at ca. 0.5 x 0.5 degree grid level. Note the logarithmic scale.**

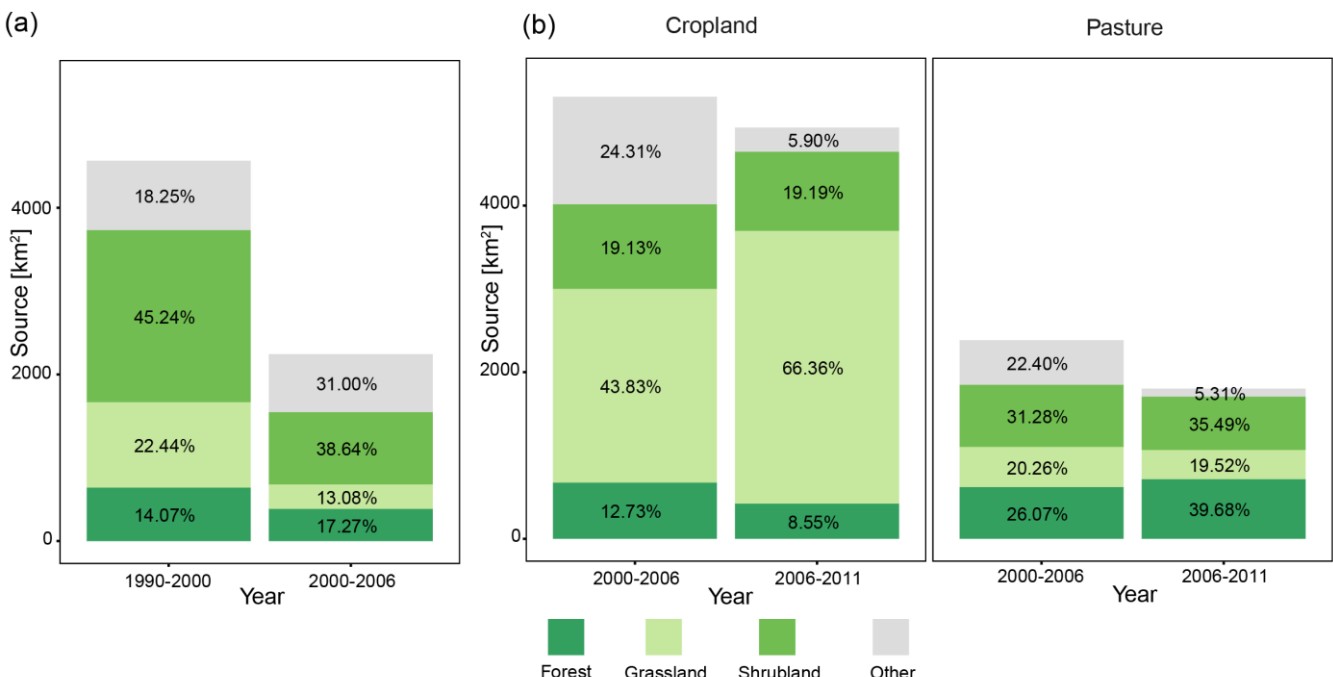

**Figure 4: Sources of agricultural land (cropland and pasture combined) for two time periods in Europe based on the CORINE land cover data (a) and sources of cropland and pasture for two time periods in the USA based on the NLCD land cover data (b) (Supplement S2.3, Table S3). Changes between different agricultural classes are not considered as expansion of agricultural land. Aggregation of CORINE and NLCD legends to forest, grassland and shrubland according to Tables S4-5. Other includes urban land, wetlands, water and bare land.**

**2000-2010**

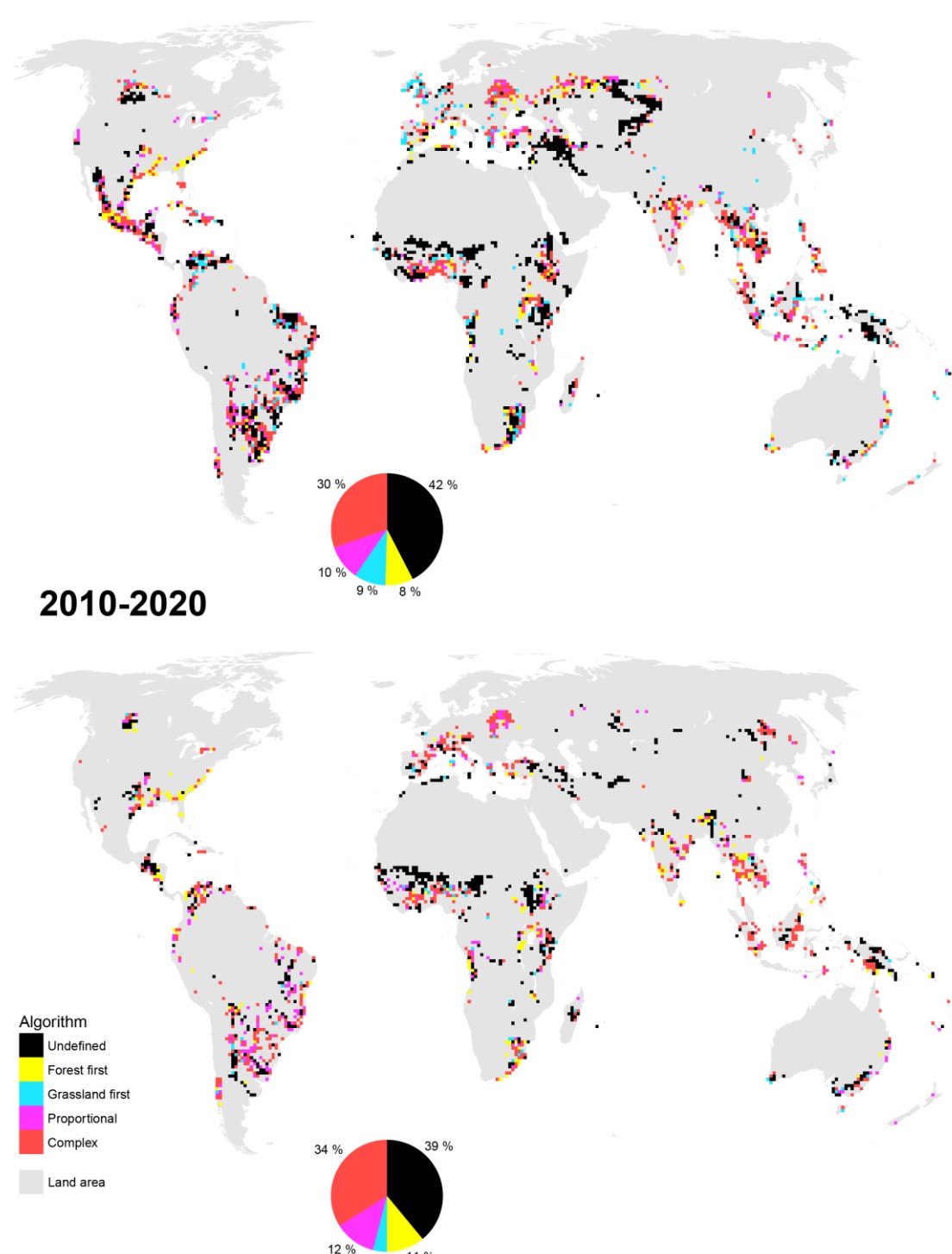

**2010-2020**

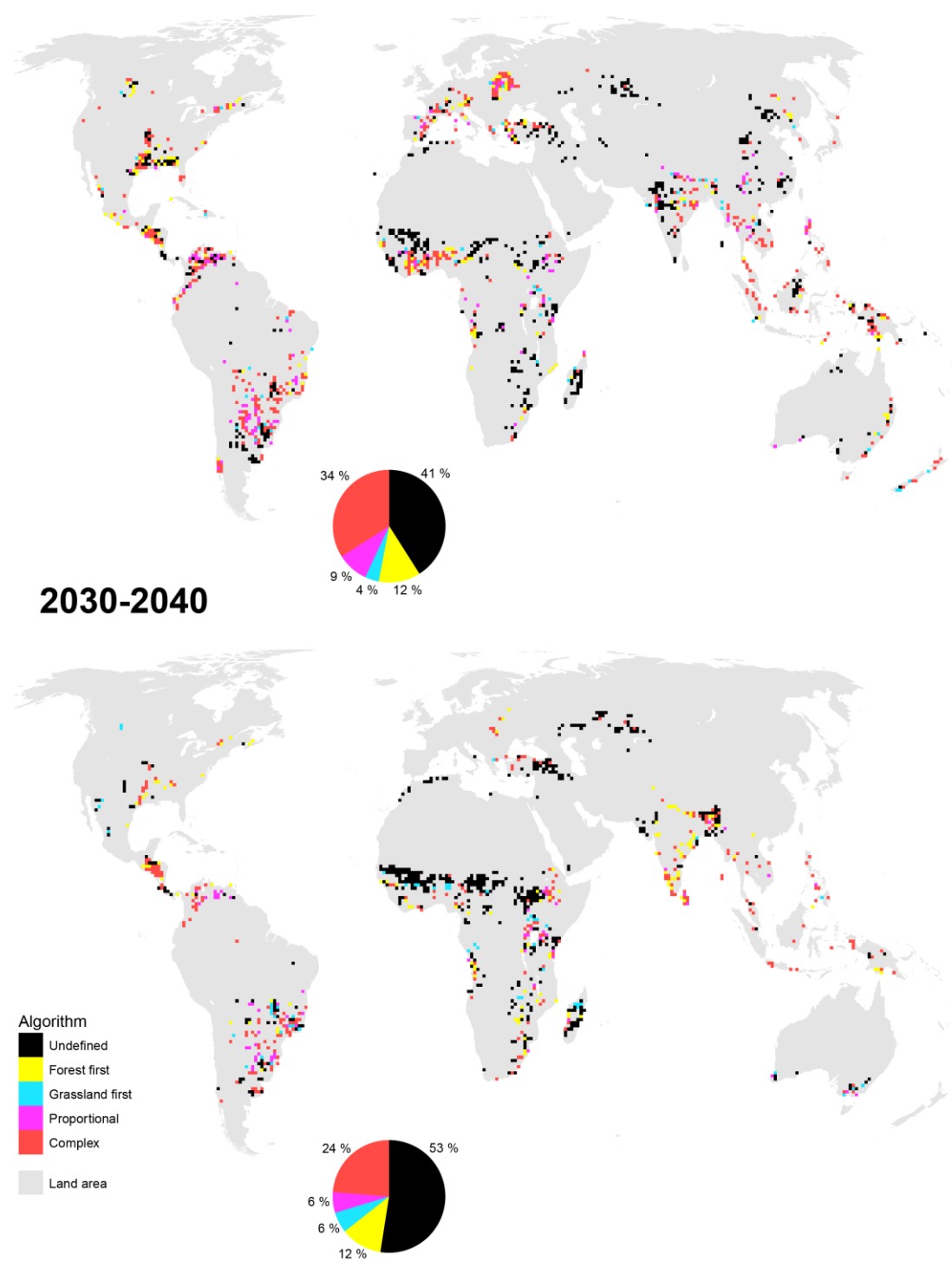

**Figure 5: Transitions from natural vegetation to cropland as shown by the CLUMondo model (FAO 3 demand scenario) from 2000 to 2040 in decadal time steps. Colored grid cells represent areas with at least 10 % of cropland expansion within a ca. 0.5 x 0.5 degree grid cell. Grid cells are classified to forest first (yellow), grassland first (cyan), proportional (magenta) and complex (red) reduction algorithm as described in the text (for details see Supplement S2.4). Black grid cells denote areas where the validity of none algorithm could be detected. Grid cells classified to unvegetated first (Table 3) are not shown due to very small contribution (< 0.1 %). Grid cells in this figure have been aggregated to ca. 1.0 x 1.0 degree following a majority resampling for reasons of readability. A high-resolution version of the maps including the full detail of the classification results can be found in the supplementary material (Figure S4).**