# Peer review of "Current challenges of implementing anthropogenic land-use and land-cover change in models contributing to climate change assessments"

_Earth System Dynamics, 2016_

## Referee Comment (RC1) · A. Di Vittorio (Referee) · 3 Nov 2016

Review of esd-2016-39: Current challenges of implementing land-use and land-cover change in climate assessments

The authors examine current practices of providing LULCC data and simulating LULCC effects on the earth system in global models and identify 3 main issues related to reliable provision and use of LULCC data. They then move on to the limitations in data for gross land use/cover transitions, and finally discuss land conversion assumptions in modeling as a source of uncertainty. They also make 3 suggestions on how to improve the provision and use of LULCC data.

I appreciate this paper and am pleased that it discusses relatively overlooked, yet very important, issues regarding LULCC and global modeling. I generally agree with the

assessment, but I think that the paper needs some reorganization and some additional discussion to fully and clearly make its case. The main issues requiring attention are summarized here, with specific comments/suggestions following:

1) The paper needs a consistent framing and argument. The three main issues are different between the abstract, text, and conclusion. It appears (see abstract lines 28-34 and page 3 lines 5-10) that the point is to show that the 3 main issues are indeed main issues, based on literature, an example, and discussion of two underlying factors ( 1) gross transitions and 2) land use change to land cover change translation ), and provide suggestions for moving forward. But the main issues are not referred to in the later sections, and these two aspects are not introduced up front so that they can be discussed in the context of the main issues. And the suggestions are not related to the issues.

2) It isn't clear that lack of information on gross transitions is a fundamental factor for the 3 main issues. While there is a lot of uncertainty in estimating gross transitions, and there is a need to improve related data, this seems more like an example of a more fundamental driver. One thing that cuts through the three issues and incorporates gross transitions is data quality. In fact, that is largely what issue 2 in the text describes. And ultimately issue 3 as well (initial, present-day data sets for future projections). Maybe there are only two main issues (single historical product with no uncertainty and uncharacterized/large model uncertainty in future land projections) and two underlying factors (data quality and independent land use and land cover implementation). Then the underlying factors provide guidance for the two communities to work together to address the two issues as they apply to both the human and dgv/es models.

3) The underlying factor of the traditional separation of land use research from land cover research is not addressed until section 4, even though it cuts through the main issues and there is also a main point in the conclusion that land use modeling needs to be integrated with land cover/ecosystem modeling. And one of the suggestions calls for specific land use to land cover conversion information in place of just land use

information. You also use land cover products for figure 5, which are not necessarily consistent with agricultural land use data. Furthermore, this separation is not explicitly discussed, with LULCC being a whole throughout the text, even when discussing how each land model has to make land cover conversion assumptions to accommodate independent land use data. You mainly focus on the land cover conversion uncertainty, but the separation of land use and land cover is the underlying source. There is some additional literature addressing this specific issue that would be useful to the authors. I would also be happy to discuss this further with the authors, as I am trying to finish a manuscript looking at how land cover conversion uncertainty affects carbon and climate projections. Look me up if you are coming to AGU in San Francisco this year.

Specific comments and suggestions:

Abstract

page 1, lines 31-32: this subgrid and gross transition source is not on page 5 as a main source of uncertainty. The second main source in the text is inconsistencies of present day data. You do later discuss gross transitions, and make a statement in the conclusion, however.

page 2, lines 1-2: I think I know who you mean (providers and users), but it is unclear who is included in the "joint development and evaluation" here.

Introduction

page 2, line 12: What do you include as a DGVM here? Some consider any model having vegetation growth in response to environmental conditions as a DGVM. For others a DGVM specifically includes prognostic biogeography (i.e., the extent of vegetation types change according to environmental conditions) and/or successional vegetation processes (e.g., stages of forest stand growth following a forest clearing disturbance).

page 2, line 26: you should also cite: Meiyappan and Jain, 2012. Three distinct global estimates of historical land-cover change an land-use conversions for over 200 years,

Frontiers in Earth Science, 6(2):122-139 (I noticed you cite it later)

page 2, lines 29-30: you should also cite: Di Vittorio et al., 2014. From land use to land cover: restoring the afforestation signal in a coupled integrated assessment-earth system model and the implications for CMIP5 RCP simulations, Biogeosciences, 11, 6435-6450

Provision of LULCC

page 4, lines 11-13: The CMIP5 product harmonizes only land use, and as such the land cover (forest, grass, etc.) and how it is altered by land use is determined independently by the DGVMs/ESMs, and can be dramatically different between models for a given scenario (in fact, prescribed scenarios can be substantially altered in ESMs by this, see Di Vittorio paper listed above). The CMIP6 product is also including forest cover in the harmonization, both for the historical period (with reference to satellite data) and for the IAM scenarios (which actually project all land cover).

page 4, lines 14-15: Only land use is input to and output from GLM for CMIP5, and forest cover is included for CMIP6.

page 5, line 12: It isn't clear here that you have shifted away from the harmonization group of models to a more general group providing present-day lulc data for future projections.

pages 5-6, lines 24-5: Two points here: 1) In the IPCC context, only land use was used, with forest cover coming into play for CMIP6, even though the IAMs project land types for the entire terrestrial surface. This introduces uncertainty beyond just the model structure/assumptions and different input data (see the Di Vittorio paper listed above). 2) The starting point of lulcc determination isn't just about which land-use input data or how processes are implemented. The spatial configuration of these data and the model are key factors in determining model outcomes. And each model has a unique spatial configuration. Gridded models/data do not necessarily resolve this

spatial issue because regional values are often just resampled to the grid. See: Di Vittorio et al, 2016. What are the effects of Agro-Ecological Zones and land use region boundaries on land resource projection using the Global Change Assessment Model, Environmental Modelling and Software, 85:246-265.

page 6, line 2: How were the variables normalized? Could the dominance of initial pasture area be due to it just being the largest difference in relation to the other variables? Also, it would be more clear if you were specific in the text and the caption in describing that the "starting point" and "initial" are the pasture area in relation to fao in 2010, and that "model" is actually model type and presumably the spatial resolution/configuration.

page 6, lines 6-14: I completely agree! While recent feedback on LUMIP has prompted the provision of LU-forest uncertainty along with the CMIP6 LUH product, it still falls short of the comprehensive approach discussed here.

page 6, lines 15-22: The separation of land use and land cover is a critical factor omitted from this discussion. While land use and land cover are often said in the same breath and the LULC(C) acronym is widely used, in nearly all cases people are referring to either land use or land cover. Research is clearly split along these lines, and land use data are remarkably inconsistent with land cover data. Land use and land cover need to be studied together, as an integrated process, in order to reduce LULCC uncertainties and inconsistencies between these two groups of data.

Considering gross land use changes

How does this relate to the three issues in the previous section? Is this really a major driver of the 3 issues, or something along for the ride? It is clearly present in issues 1 and 2 (although the present day isn't discussed, only past and future), while its absence in IAM projection may be the relevant link (as the transitions are determined by a single independent model, which is part of issue 1)

page 7, lines 21-23: Just a note: You are well aware that gross transition information

is highly uncertain, and current work suggests that the CMIP5 LUH data product may actually overestimate gross transitions in the tropical regions.

page 7, line 32: there are no land cover categories in CMIP5 LUH, only primary and secondary land. wood harvest is associated with forest or non-forest, but this land cover designation is based on a threshold of a potential biomass model, rather than more commonly used land cover or potential vegetation data sets.

page 8, line 14: "…increasingly been captured…"

Allocation of managed land in ESMs and DGVMs

Ah, finally! This aspect of separate land use and land cover information/modeling is a factor in all 3 of your main sources of uncertainty, and as such needs to be mentioned up front and related to these uncertainty sources.

page 8, lines 27-28: and scenarios and over relatively short time periods (see Di Vittorio et al 2014)

page 9, lines 26-27: this is consistent with land cover being studied separately from land use, and your examples also relate to your second main source of uncertainty

pages 9-10, lines 30-12: glad you did this! But what determines the source of land use in CLUMondo? It is important to clearly state how this model differs in this selection versus those that use the methods by which you classify the changes. Generally, more info is needed regarding how the different classified algorithms are defined, in relation to how they are implemented in dgvms/esms. The reader should be able to understand what is going on without digging through the supplemental material. Maybe a table of the definitions?

page 10, lines 5-8: what about the undefined category, which is the dominant category according to the figures (not the complex)? what does it stand for? are you grouping this with the complex category?

page 10, line 16: the IAM community has been projecting land use AND land cover for some time, although not necessarily gross transitions.

Conclusion and recommendations

page 11, lines 17-20: this is an important point, but it hasn't been clearly demonstrated in the text, largely because the paper generally refers to LULCC as a whole.

page 11, lines 20-23: while not an individual, agent-based behavioral model, GCAM has been integrated with CESM as the iESM, implementing two-way feedbacks between the human and environmental systems, particularly for terrestrial systems and the effects on land projection. See the oft-noted paper above, which runs the iESM, and: Collins et al, 2015, The integrated Earth system model version 1: formulation and functionality. Geoscientific model development, 8,:2203-2219. There should also be a paper coming out soon on a complete experiment using the iESM to examine the effects of the feedbacks.

page 12, lines 1-19: It isn't clear how these suggestions relate to your three proposed primary issues with LULCC (which should also be restated in this conclusion - 1) uncertainty in lulcc data products is lacking due to not enough different products generated, 2) present-day lulc data are inconsistent and thus contain high uncertainty, and 3) uncertainty in lulcc projections is largely driven by initial data uncertainty over other model-specific sources). Issues 2 and 3 appear to mainly consist of data quality issues. Also note that your second main source on page 5 does not refer to gross or subgrid transitions at all, just to inconsistencies in present day data. Please make your conclusion/suggestions more consistent with the theme of the paper.

Supplemental material

Figure S1 seems a lot more complicated with more steps than described in the text, which makes more sense.

---

## Referee Comment (RC2) · Anonymous Referee #2 · 7 Nov 2016

Review esd-2016-39: Current challenges of implementing land-use and land-cover change in climate assessments

The authors discuss in their manuscript the major uncertainties and shortcomings associated with the implementation of land-use and land-cover changes (LULCC) in climate change assessments. Additionally, three major challenges are identified and the reasons for them are discussed.

General Comment: Generally, I think this paper raises some important issues related to implementation of LULCC in the modelling community. Raising awareness to this issues with the help of an extended literature review will be beneficial in tacking this problems. However, I think the manuscript would benefit from not only raising awareness of the issues and the reasons behind it but also providing a 'way forward'. This

is sometimes done in the individual sections but I think the suggestions get lost in the wealth of information presented in the manuscript. I therefore suggest adding a separate recommendation section in which the authors clearly state in short and precise form what could/should be done to overcome the challenges and which part of the community they see in a better position to take the lead (if possible), instead of having the conclusions and recommendations combined in one section. The abstract and P3L5-10 ('overall objectives') do not match. Please make sure that these points are consistent throughout the document.

This brings me to my general comment; The paper covers a lot of material but very often the structure of the document gets lot. I therefore suggest the authors to go through the document and make sure that the reader can follow the train of thought easily. This could be achieved for example by adding subheading to the sections and having a similar structure within each of the sub-sections. This also applies to the Supplements which should also convey a clear structure within the sub-sections.

Specific Comments: In the title of the manuscript it should become clear that the document is about 'anthropogenic' land-use and land cover changes and that the assessments are with regard to 'climate change'. I therefore suggest revising the title.

P1L20-23: Suggested to split sentence

P2L22-26: I think as this is an interdisciplinary journal, it would be beneficial for readers outside of the community, if examples of the uncertainties could be given (e.g. it might not be clear to the readers what 'definition issues' are (see also P5L 16)).

P3L1: which 'sources'? Please specify

P3L27-31 Please be more specific and elaborate on the dataset

P4L6&L9: What does 'LUH' and 'HYDE' stand for?

P5L18: 'large differences' in which variable?

P5L23: what are the 'all relevant processes' to the authors? Elaborate. What is still missing.

P5L26: It is not clear what a 'marker scenario' entails

P5L33: 'large variations' in which variable?

P6L5: The differences in output arising from different models the input and calibration etc. is not only an issue in the assessment of LULCC but generally applies to all models... Maybe if you look at other modelling communities and how they quantify these uncertainties.

P 6L11: the problem is how to define 'plausible' realisations.

P7L 2: Elaborate why sub-grid dynamics 'have been shown important'.

P7L6: what are 'under-determined mathematical systems' in this context?

P7L8: what are 'minimum-transitions' in this context?

P7L21-23: Rephrase sentence

P10: rephrase 'allocation issue' do you mean the shifts between the communities and their perceived responsibilities

P10 Maybe you would like to add to your discussion that 'satellite data' is also not 'directly measured data' but also goes through a mathematical conversion process.

P11L24: Can you elaborate what 'improved communication' should entail in an ideal case. Additionally, I think it is not only the 'understanding' but first the 'awareness' of different assumptions and constrains needs to be achieved and before one can understand and tackle the problems.

Figures: P24: add a 'log' label to the legend and add colour areas without change in grey to make the light yellow areas better stand out.

Supplement SP2L7: can you provide more details on the updated version, i.e. reference

SP5: Can you comment on the uncertainties associated with the CORINE data

SP6L8: Can you elaborate on what the 'thematic accuracy' entails.

---

## Referee Comment (RC3) · Anonymous Referee #3 · 23 Nov 2016

The manuscript by Prestele et al., "Current challenges of implementing land-use and land-cover change in climate assessments", provides an overview of recent publications on interactions among land-use, carbon cycling, and different aspects of climate. First, the manuscript aims "...to identify existing shortcomings of the current LULCC representation within DGVMs and ESMs, reveal the underlying mechanisms and constraints that have hampered improved representations until now, and propose pathways to improve current representations" (page 3, lines 5-7). Second, based on the literature review, the manuscript attributes the lack of progress in including LULCC into climate assessments, to 1) the failure to account for uncertainty in reconstruction and future scenarios of gridded LULCC; 2) resolving sib-grid changes in land-use activities (e.g. gross transitions); 3) allocation of primary lands to managed lands in DGVMs and ESMs. Manuscript reviews a number of studies and discusses a wide range of

limitations, specifically in CMIP5 historical reconstruction and future scenario. It has interesting discussion of how to use remote sensing data in improving treatment of LULCC processes in scenario development and its implementation into DGVMs and ESMs. However, the title is not appropriate because climate assessments such as IPCC do not implement LULCC – IPCC assessments review literature. CMIPs are not part of the IPCC, although their model simulations provide input to IPCC.

The manuscript has four major shortcomings: 1) While the manuscript reviews and synthesizes a number of recent studies on the development of scenarios of LULCC and of LULCC for climate and carbon cycling, it does not actually provide new insights or synthesis of LULCC implementation in ESMs and DGVMs. The manuscript provides a discussion of how the CMIP5 scenario was constructed and its limitations, but does not discuss differences in land use components of different ESMs or DGVMs. Or how they implemented the CMIP5 LULCC scenario. Table 1 gives 4 examples: 3 DGVMs (2 of which are variants of LPJ model) and a new HadGEM2-Jules ESM. There is no comprehensive analysis of CMIP5 ESMs or TRENDY DGVMs used in the AR5 in respect to LULCC. Thus, the manuscript's first goal is not supported by new insights beyond those previously published in the literature. 2) The manuscript claims that the limited characterization of uncertainty in CMIP5 and CMIP6 LU reconstructions and scenarios is responsible for the lack of progress on LULCC in climate assessments. There is no reason to believe that's true. CMIP is designed to compare climate models and ESMs under a common set of forcings and capture model structural uncertainty. CMIPs never claimed to capture all uncertainty due to input forcing. It's a well-established practice in climate MIPs to provide a standard scenario for all forcings – greenhouse gases, short-lived species, solar, constants, volcanoes and LULCC, particularly over historical periods. Such GCM or ESM simulations are extremely computationally expensive. Permutation of alternative forcings datasets is not likely something that many climate centers will be able to engage and afford. The idea of multiple LULCC reconstructions advocated by the paper for CMIPs is not practical.

[Figure]

If some modeling group/center wants to explore uncertainty due to LULCC, there is more than one scenario that is available even from the GLM model: Hurtt et al. (2006) included both scenarios based on SAGE and HYDE datasets. Hurtt et al. 2011 examines different assumptions in the GLM model. The main bottleneck for improving LULCC characterization in the CMIP is poor representation of LULCC processes in GCMs and ESMs. Most CMIP5 ESMs or TRENDY DGVMs can't use the information available in CMIP5 or CMIP6 historical reconstructions or future scenarios. For example, most of the CMIP5 models use only information about land use fractions, and not gross transitions provided by the Hurtt et al. (2011) data set. With the exception of very few models, ESMs do not represent shifting cultivation or wood harvesting. Another unsupported assumption in the manuscript is that, by making additional ESMs or GCMs with alternative representations of LULCC history, one would get a better handle on the uncertainty in climate feedback of LULCC. It's not necessarily true: most studies with and without LULCC typically find a small difference in global climate and small regions with statically distinguishable differences in climate characteristics. One would need a large ensemble of such simulations to find differences between the biogeophysical effects of alternative LULCC reconstructions and scenarios, unless they are really different as in future scenarios. Biogeophysical differences should be more pronounced, but the problem is that CMIP5 or even CMIP6 ESMs are incapable of representing major LU processes such as shifting cultivation, wood or crop harvesting..

3) The manuscript questions assumptions in CMIP5 Hurtt et al. 2011 reconstruction and future scenario. The Hurtt et al. (2011) effort, for the first time, harmonized historical reconstruction with the 4 Representative Concentration pathways (RCP) scenario and took into account gross transitions between different LU types in both tropics and extra-tropics. The authors are mistaken in their assumption that no-shifting cultivation in the extra-tropics implies no gross-transitions in the extra-tropics; for example, non-zero transitions between pastures to crops and crops to pastures. Furthermore, for CMIP6 (Lawrence et al. 2016), there will be a focus and additional LUH reconstructions available, as well as more details about the relationship between land cover

and land use categories. I think a lot of criticism of the CMIP5 LULCC reconstruction and scenario is valid but the authors are overlooking improvements in the new reconstruction for CMIP6, which is publicly available now on the CMIP6 website. While it's possible to construct more detailed scenarios for recent periods with satellite coverage or for specific countries (e.g., Table 2 in the manuscript), particularly in the Northern Hemisphere, it is difficult if not impossible to develop multi-century reconstructions on a global scale with consistent sets of assumptions. Making simple assumptions in ESM is not an unreasonable approach for global, multi-century analyses. Assuming transitions based on the satellite era for the entire CMIP-style experiments may be problematic, as well, for pre-industrial or future periods

4) The rationale for including analysis from the CLUMondo model is not clear – it demonstrates how spatio-temporal variations could be different within the grid. It does not show that such patterns will affect climate or carbon cycling. Besides the CLUMondo analysis, there is no new analysis in this manuscript. So, there are no new insights/analysis, just a synthesis of other studies, which are already partially covered by the authors in related publications (e.g., Alexander et al. 2016, Bayer et al, 2016, Prestele et al. 2016).

I think the most interesting part of the paper is the section on remotely sensed data (high and low resolution) in development of new diagnostics for evaluation of global LULCC reconstructions or models. Perhaps the authors can re-frame their analysis and demonstrate how such data can be used to improve or evaluate reconstructions (e.g. the one in CMIP6) or to create new diagnostics to evaluate ESMs and DGVMS.

References

Alexander, P., Prestele, R., Verburg, P.H., Arneth, A., Baranzelli, C., Batista e Silva, F., Brown, C., Butler, A., Calvin, K., Dendoncker, N. and Doelman, J.C., 2016. Assessing uncertainties in land cover projections. Global Change Biology.

Bayer, A. D., Lindeskog, M., Pugh, T. A., Fuchs, R., & Arneth, A. Uncertainties in the

land use flux resulting from land use change reconstructions and gross land transitions. Earth Syst. Dynam. Discuss., doi:10.5194/esd-2016-24, 2016

Hurtt, George C., et al. "The underpinnings of land‐use history: Three centuries of global gridded land‐use transitions, wood‐harvest activity, and resulting secondary lands." Global Change Biology 12.7 (2006): 1208-1229.

Hurtt, George C., et al. "Harmonization of land-use scenarios for the period 1500–2100: 600 years of global gridded annual land-use transitions, wood harvest, and resulting secondary lands." Climatic change 109.1-2 (2011): 117-161.

Prestele, R., Alexander, P., Rounsevell, M.D., Arneth, A., Calvin, K., Doelman, J., Eitelberg, D.A., Engström, K., Fujimori, S., Hasegawa, T. and Havlik, P., 2016. Hotspots of uncertainty in land‐use and land‐cover change projections: a global‐scale model comparison. Global change biology.

---

## Author Comment (AC1) · 15 Dec 2016

First of all we would like to thank the three reviewers for their time and effort they spent to review our manuscript. It is certainly possible to improve the manuscript based on the detailed and constructive comments.

The first two reviewers are supportive of the paper, acknowledge its importance and we are able to revise the manuscript in such a way that all comments and requests are met. The comments of reviewer #3 have a different nature. We are able to address the more specific comments of reviewer #3, but disagree with the overall comment of this reviewer on lack of relevance of the paper.

Reviewer #1 suggests a reframing of the manuscript from our 'three challenges' and paying more attention to the distinction between land use and land cover or their separate research history, respectively. These are valid points which will help us to improve the manuscript. We intend to resolve this by more clearly describing the framing of the paper. For example, there is  room for misinterpretation of the structure in the original manuscript: while we intended to present one important challenge ('main issue') per section, the reviewer identifies the three main challenges already in section 2 (*Provision of spatially explicit, continuous and consistent time series of LULCC*) and thus the other two sections seem to be disconnected. We therefore suggest to include a guiding paragraph at the end of the introduction to clarify our underlying structure as following:

*'Each of the following sections thereby presents one of the three challenges we identify to be crucial in future land use – climate interaction research, reviews the issue and its implications for the results of climate change assessments based on previously published literature and in the context of the widely applied land use harmonization (LUH) dataset by Hurtt et al. (2011). Moreover, in sections 3 and 4 we perform additional analysis using data from the CLUMondo land-use change model (Van Asselen and Verburg, 2013) to illustrate our arguments. Section 5 synthesizes the three challenges and provides recommendations to move forward in land use – climate interaction studies.'*

Other revisions in line with the comments of this reviewer are outlined in detail below.

Reviewer #2 comments broadly concentrate around hiddenness of the 'ways forward' in the individual sections. We propose to address this in a revised version following the reviewer's suggestions and will include sub-headings, re-organize the (sub-)sections towards a common structure and move the 'ways forward' to a separate section and also slightly expanding on these.

Reviewer #3 concludes that we do not provide 'new insights, synthesis or analysis' and suggests a reframing of our manuscript towards the 'development of new diagnostics for evaluation of global LULCC reconstructions or models'. We do not share this view. The current manuscript is a review, supported by data (e.g., CORINE and NLCD land cover data in section 4) and new analysis (e.g., allocation of managed land as shown by CLUMondo in section 4), of the most dominant issues related to land use and climate modeling. The manuscript brings together, to our knowledge for the first time, prominent challenges in land use – climate interaction studies. This view is shared by the other reviewers. The purpose of the paper is to review the current problems and progress, and identifying possible ways forward for the community. We do agree with the reviewer's assessment that identifying more conclusively the ways forward to overcome existing stumbling blocks will add to the value of the manuscript and will revise the paper to strengthen that aspect.

For a detailed list of how we are aiming to address the individual reviewer's comments and improve the manuscript, please see our following responses inline in blue.

**Reviewer #1 (A. Di Vittorio)**

The authors examine current practices of providing LULCC data and simulating LULCC effects on the earth system in global models and identify 3 main issues related to reliable provision and use of LULCC data. They then move on to the limitations in data for gross land use/cover transitions, and finally discuss land conversion assumptions in modeling as a source of uncertainty. They also make 3 suggestions on how to improve the provision and use of LULCC data.

I appreciate this paper and am pleased that it discusses relatively overlooked, yet very important, issues regarding LULCC and global modeling. I generally agree with the assessment, but think that the paper needs some reorganization and some additional discussion to fully and clearly make its case. The main issues requiring attention are summarized here, with specific comments/suggestions following:

Response: We thank the reviewer for the kind words and the appreciation of our work. We are pleased that the reviewer generally agrees with our assessment. We will revise the paper to provide a more clear structure as we feel the current framing provided some misunderstanding: we did not aim to first identify the three main issues and then 'move on to limitations in data for gross land use/cover transitions, and finally discuss land conversion assumptions in modeling', but the three main issues are intended to be the three sections following the introduction, namely (1) *Provision of a consistent LULCC time series*, (2) *Consideration of gross transitions*, and (3) *Allocation of managed land in DGVMs/ESMs*. We will refer to this and explain more in the responses to following comments.

1) The paper needs a consistent framing and argument. The three main issues are different between the abstract, text and conclusion. It appears (see abstract lines 28-34 and page 3 lines 5-10) that the point is to show that the 3 main issues are indeed main issues, based on literature, an example, and discussion of two underlying factors ( 1) gross transitions and 2) land use change to land cover change translation ), and provide suggestions for moving forward. But the main issues are not referred to in the later sections, and these two aspects are not introduced up front so that they can be discussed in the context of the main issues. And the suggestions are not related to the issues.

Response: Based on the assumptions the reviewer made regarding the three main issues and the underlying factors, we entirely agree that there is a lack of consistency and disconnection throughout the manuscript. We propose to add a paragraph at the end of the introduction that will clarify the structure (please see the general response for a suggestion). We will also remove the confusing sentences that give the impression that we present the 'main issues' in the first section (after the introduction) and discussing 'underlying factors' subsequently. A revised manuscript will make sure that each section (2, 3, 4) treats one of the challenges outlined in the abstract and picked up in the recommendation section individually, connected by the overarching topic of 'shortcomings/issues in land use – climate interaction studies'.

In conjunction with the changes outlined later on in this document regarding structure requested by reviewer #2, we aim to address the concern raised.

2) It isn't clear that lack of information on gross transitions is a fundamental factor for the 3 main issues. While there is a lot of uncertainty in estimating gross transitions, and there is need to improve related data, this seems more like an example of a more fundamental driver. One thing that cuts through the three issues and incorporates gross transitions is data quality. In fact, that is largely what issue 2 in the

text describes. And ultimately issue 3 as well (initial, present-day data sets for future projections). Maybe there are only two main issues (single historical product with no uncertainty and uncharacterized/large model uncertainty in future land projections) and two underlying factors (data quality and independent land use and land cover implementation). Then the underlying factors provide guidance for the two communities to work together to address the two issues as they apply to both the human and dgv/es models.

Response: We will rephrase the section in such a way that we no longer present the lack of information on gross transitions as a fundamental factor for other issues/challenges. We will make more clear that gross transitions are a challenge in itself, since these additional changes are hardly considered in climate models, but may have a substantial impact, e.g. on the resulting carbon signal. Indeed, it closely links to both of the other challenges, to the harmonization (section 2) in terms of data quality and to the allocation (section 4) in terms of how the DGVMs/ESMs are able to implement the transitions. However, the reframing suggested by the reviewer would make the gross transitions to one out of several 'data quality' issues, which we feel is not appropriate. The topic of gross transitions goes beyond data quality issues due to their scale dependency, i.e. even if we would be able to reduce uncertainties due to technical restrictions (e.g., in remote sensing products) or model assumptions, sub-grid processes would need to be addressed separately. Similarly, the suggested reframing would exclude the uncertainty from present-day land-use (and land-cover) products which different land-use model intercomparison exercises have identified as an important source of variation across land-use models (Alexander et al., 2016; Schmitz et al., 2014). We however agree that challenge 1 (*Provision of consistent time series*) is mainly related to data quality, which we will emphasize in a revised version of the manuscript.

3) The underlying factor of the traditional separation of land use research from land cover research is not addressed until section 4, even though it cuts through the main issues and there is also a main point in the conclusion that land use modeling needs to be integrated with land cover/ecosystem modeling. And one of the suggestions calls for specific land use to land cover conversion information in place of just land use information. You also use land cover products for figure 5, which are not necessarily consistent with agricultural land use data. Furthermore, this separation is not explicitly discussed, with LULCC being a whole throughout the text, even when discussing how each land model has to make land cover conversion assumptions to accommodate independent land use data. You mainly focus on the land cover conversion uncertainty, but the separation of land use and land cover is the underlying source. There is some additional literature addressing this specific issue that would be useful to the authors. I would also be happy to discuss this further with the authors, as I am trying to finish a manuscript looking at how land cover conversion uncertainty affects carbon and climate projections. Look me up if you are coming to AGU in San Francisco this year.

Response: The reviewer has a valid point that we did not carefully separate between land use and land cover throughout the manuscript and especially in section 4. We will check again and change accordingly. As mentioned later on in the response to reviewer #2, we will additionally clarify the structure such that a recommendation section will focus on the short term improvements (i.e., how to tackle the individual issues within the current coupling) and an outlook section will discuss the model integration. We do not necessarily agree that the separation of land use from land cover research 'cuts through the main issues', but it rather applies to our issue 3 (*Allocation of managed land in ESMs and DGVMs*). In addition to this, we discuss the issues arising in the data used for the current coupling in section 2. We will clarify this in a revised manuscript as outlined in the response to the reviewers first general comment.

Specific comments and suggestions:

Abstract

page 1, lines 31-32: this subgrid and gross transition is not on page 5 as a main source of uncertainty. The second main source in the text is inconsistencies of present day data. You do later discuss transitions, and make a statement in the conclusion, however.

Response: See response to this reviewers main comments. The revised manuscript will ensure consistency.

page 2, lines 1-2: I think I know who you mean (providers and users), but it is unclear who is included in the "joint development and evaluation" here.

Response: Thanks for pointing to that. We will rephrase this to clarify as following:

*'We propose that LULCC data-provider and –user communities should engage on the joint development and evaluation of enhanced LULCC time series, […]'*

page 2, line 12: What do you include as a DGVM here? Some consider any model having vegetation growth in response to environmental conditions as a DGVM. For others a DGVM specifically includes prognostic biogeography (i.e., the extent of vegetation types change according to environmental conditions) and/or successional vegetation processes (e.g., stages of forest stand growth following a forest clearing disturbance).

Response: We agree with the reviewer that the DGVM terminology is misleading in our manuscript. We suggest to replace it with the term *'terrestrial biosphere model (TBM)',* which covers models with both static and dynamic vegetation. Additionally, we will add a short explanation that will make clear that this term is meant to be inclusive of a range of modeling approaches, as basically any terrestrial biosphere model that is able to include land use from an external data set will face the challenges discussed in our manuscript.

page 2, line 26: you should also cite: Meiyappan and Jain, 2012. Three distinct global estimates of historical land-cover change and land-use conversions for over 200 years, Frontiers in Earth Science, 6(2):122-139 (I noticed you cite it later).

Response: Good point. We will include the reference.

page 2, lines 29-30: you should also cite: Di Vittorio et al., 2014. From land use to land cover: restoring the afforestation signal in a coupled integrated assessment-earth system model and the implications for CMIP5 RCP simulations, Biogeosciences, 11, 6435-6450.

Response: We apologize that we have not been aware of this work during the preparation of the manuscript. We will include the reference.

Provision of LULCC

page 4, lines 11-13: The CMIP5 product harmonizes only land use, and as such the land cover (forest, grass, etc.) and how it is altered by land use is determined independently by the DGVMs/ESMs, and can be dramatically different between models for a given scenario (in fact, prescribed scenarios can be substantially altered in ESMs by this, see Di Vittorio paper listed above). The CMIP6 product is also

including forest cover in the harmonization, both for the historical period (with reference to satellite data) and for the IAM scenarios (which actually project all land cover).

Response: We agree that the term LULCC is not appropriate in this context. We propose to improve this paragraph by exchanging the term 'LULCC' by 'land use'. Furthermore, we suggest to replace the sentence

*'This strategy tries to conserve the original patterns, rate and location of change as much as possible, and to reduce the differences between the models due to definition of land-use categories (e.g., what constitutes a forest).'*

by

*'This strategy tries to conserve the original patterns, rate and location of change as much as possible, and to reduce the differences between the models due to inconsistent definitions of cropland, pasture and wood harvest.'*

page 4, lines 14-15: Only land use is input to and output from GLM for CMIP5, and forest cover is included for CMIP6.

Response: We will revise as mentioned in the response to the previous comment.

page 5, line 12: It isn't clear here that you have shifted away from the harmonization group of models to a more general group providing present-day lulc data for future projections.

Response: It is not really a shift from the harmonization group of models to a more general group, but we are not aware of work that has compared the differences in the starting maps of the IAMs contributing to the CMIP5 harmonization only. The cited Prestele et al. (2016) paper indeed includes additional (non-IA) models, but still three out of four models providing data to the CMIP5 harmonization. These do not agree about initial areas as well. We will clarify as following:

*'This uncertainty, however, is represented also in the starting maps of the different land-use change models providing land-use data to climate models, including the IAMs providing data for the harmonization.'*

pages 5-6, lines 24-5: Two points here: 1) In the IPCC context, only land use was used, with forest cover coming into play for CMIP6, even though the IAMs project land types for the entire terrestrial surface. This introduces uncertainty beyond just the model structure/assumptions and different input data (see the Di Vittorio paper listed above). 2) The starting point of lulcc determination isn't just about which land-use input data or how processes are implemented. The spatial configuration of these data and the model are key factors in determining model outcomes. And each model has a unique spatial configuration. Gridded models/data do not necessarily resolve this spatial issue because regional values are often just resampled to the grid. See: Di Vittorio et al, 2016. What are the effects of Agro-Ecological Zones and land use region boundaries on land resource projection using the Global Change Assessment Model, Environmental Modelling and Software, 85:246-265

Response:

1) We agree with the reviewer about the missing land cover information being a major constraint in the CMIP5 LUH product and propose to clarify and expand the discussion about the missing land cover information in CMIP5 LUH in the section *Allocation of managed land in ESMs and DGVMs*. We do not

think that the discussion about uncertainty emerging from the fact, that only land use is provided to DGVMs/ESMs and they utilize different implementation schemes on their background land cover, would fit in section 2 where we discuss the uncertainties arising during land-use modeling and the harmonization.

2) We agree that the spatial configuration of the individual models even increases the uncertainty range and appreciate the reference to the Di Vittorio et al. (2016) paper. We propose to include this additional uncertainty issue by rephrasing page 5, lines 26-28 to:

*'Land-use change model intercomparisons and sensitivity studies, however, indicate that the uncertainty range emerging from different assumptions in the models, input data and spatial configuration substantially impacts the model results (Alexander et al., 2016; Schmitz et al., 2014; Di Vittorio et al., 2016).'*

page 6, line 2: How were the variables normalized? Could the dominance of initial pasture area be due to it just being the largest difference in relation to the other variables? Also, it would be more clear if you were specific in the text and the caption in describing that the "starting point" and "initial" are the pasture area in relation to fao in 2010, and that "model" is actually model type and presumably the spatial resolution/configuration.

Response: We will include a clear mentioning how the data were processed. The areas are indeed in absolute terms, which indeed makes it likely to have larger deviations. However, in the text we will describe the range relative to the average value in more detail to address this point and indicate that the larger variation cannot be explained by the larger initial area. We will improve the caption as suggested.

page 6, lines 6-14: I completely agree! While recent feedback on LUMIP has prompted the provision of LU-forest uncertainty along with the CMIP6 LUH product, it still falls short of the comprehensive approach discussed here.

Response: We thank the reviewer and also appreciate initial consideration of uncertainty along with LULCC data provided to LUMIP, e.g. that additional high and low estimates of historical land use will be included (Lawrence et al., 2016). However, we think a more ambitious approach should be taken into consideration for future MIPs.

page 6, lines 15-22: The separation of land use and land cover is a critical factor omitted from this discussion. While land use and land cover are often said in the same breath and the LULC(C) acronym is widely used, in nearly all cases people are referring to either land use or land cover. Research is clearly split along these lines, and land use data are remarkably inconsistent with land cover data. Land use and land cover need to be studied together, as an integrated process, in order to reduce LULCC uncertainties and inconsistencies between these two groups of data.

Response: We agree with the reviewer that there is a large discrepancy between land use and land cover research and data products and that this will be an important challenge to overcome in future by integrating the research lines. In fact, model integration is one of the key points of our conclusions. We will highlight it more clearly in a revised manuscript by rephrasing our three conclusion points in a 'recommendation' section (for the more short-term improvement potential) and discussing the model integration in an 'outlook' section.

Considering gross land use changes

How does this relate to the three issues in the previous section? Is this really a major driver of the 3 issues, or something along for the ride? It is clearly present in issues 1 and 2 (although the present day isn't discussed, only past and future), while its absence in IAM projection may be the relevant link (as the transitions are determined by a single independent model, which is part of issue 1)

Response: See earlier responses. We will clarify in the text that we do not argue that missing gross transition representation is a major driver of the uncertainty in land-use data sets used for the harmonized time series. Separate from the uncertainties discussed in the previous section we argue that gross transitions are an issue that have not got enough attention throughout the communities apart from shifting cultivation in the tropics.

page 7, lines 21-23: Just a note: You are well aware that gross transition information is highly uncertain, and current work suggests that the CMIP5 LUH data product may actually overestimate gross transitions in the tropical regions.

Response: Indeed, gross transition information is uncertain, especially if it is provided by models that have multiple other sources of uncertainty (as discussed in section 2) or simple assumptions about the spatial distribution of shifting cultivation have to be made (as in the CMIP5 harmonization). For this reason, we call for additional work on this topic, based on empirical data, such as the updated shifting cultivation estimate for the CMIP6 LUH product (which the reviewer probably refers to here) or the recent work by Fuchs et al.

page 7, line 32: there are no land cover categories in CMIP5 LUH, only primary and secondary land. wood harvest is associated with forest or non-forest, but this land cover designation is based on a threshold of a potential biomass model, rather than more commonly used land cover or potential vegetation data sets.

Response: We will remove 'land-cover' from the sentence, as it is indeed misleading along with the CMIP5 LUH data. But even if forest/non-forest land cover is provided for CMIP6, the decision how to derive the transitions (e.g., forest to crop or non-forest to crop) has to be made? We thus believe our main argument in this paragraph about the simplistic assumptions to derive transitions remain unaffected.

page 8, line 14: "…increasingly been captured…"

Response: Will be corrected accordingly.

Allocation of managed land in ESMs and DGVMs

Ah, finally! This aspect of separate land use and land cover information/modeling is a factor in all 3 of your main sources of uncertainty, and as such needs to be mentioned up front and related to these uncertainty sources.

Response: See the general response, previous responses and the response to reviewer #2 how we will clarify the structure in a revised manuscript to address this comment. In short, section 2 will focus on the uncertainties within the land-use and land-cover data provider community ('data quality') and its implications for climate assessments upon coupling to DGVMs/ESMs. Section 3 will pick up an important, but so far hardly considered challenge (gross transitions), while section 4 will discuss the issues arising

when using external land-use data in DGVMs/ESMs, including the underlying source of the separated land-use and land-cover research lines.

page 8, lines 27-28: and scenarios and over relatively short time periods (see Di Vittorio et al 2014)

Response: We can remove 'over long time periods'.

page 9, lines 26-27: this is consistent with land cover being studied separately from land use, and your examples also relate to your second main source of uncertainty.

Response: The reviewer is correct that the overview of these studies also show that there is a mix of approaches in studying current land-use and land-cover changes that necessarily leads to some kind of uncertainty. However, this statement was intended to emphasize that we did not conduct a systematic review, but rather use the studies as an illustration to support our argument that the change pattern is spatially heterogeneous.

pages 9-10, lines 30-12: glad you did this! But what determines the source of land use in CLUMondo? It is important to clearly state how this model differs in this selection versus those that use the methods by which you classify the changes. Generally, more info is needed regarding how the different classified algorithms are defined, in relation to how they are implemented in dgvms/esms. The reader should be able to understand what is going on without digging through the supplemental material. Maybe a table of the definitions?

Response: We will include some more detail in the main text (P9L33 onwards; see a suggestion below), add a table with the definitions (from S2.4), and expand the detailed description in the supplementary materials. In the original manuscript we aimed to keep methodological descriptions as short as possible, as we apply the analysis mainly for illustrative purposes.

*'CLUMondo models the spatial distribution of land systems over time, instead of land use and land cover directly. Land systems are among others characterized by a mosaic of land use and land cover (cropland, grassland, forest, urban, bare) within each grid cell. The land systems are allocated to the grid in each time step 'based on local suitability, spatial restrictions, and the competition between land systems driven by the demands for different goods and services' (Van Asselen and Verburg, 2013; Eitelberg et al., 2016). Thus the determination of the source land use can be interpreted as a complex algorithm taking into account external demands, the previous time step, local suitability in a grid cell and neighborhood effects. This strategy differs from the one in DGVMs/ESMs in a way that not one simple rule (e.g., grassland first) is applied to each grid cell equally, but accounts for spatial heterogeneity of land-use sources. We reclassified the CLUMondo outputs according to their dominant land use or land cover type to derive the transitions and classified the changes in the simulated data within each ca. 0.5 x 0.5 degree grid cell as either grassland first, forest first, proportional, or a complex reduction pattern (Table 3; see SI for details).'*

page 10, lines 5-8: what about the undefined category, which is the dominant category according to the figures (not the complex)? what does it stand for? are you grouping this with the complex category?

Response: Good point. We seem to have missed the undefined category somewhere on the line and will add this piece of information to the text, including a table that will explain the individual categories. In principle, undefined means that it was not possible to detect on of the algorithms (forest first, grassland first, etc.). We are not grouping it with the complex category, as this category basically entails the

opposite: all major land-cover types have been available, but there is no priority which one is reduced upon cropland expansion.

page 10, line 16: the IAM community has been projecting land use AND land cover for some time, although not necessarily gross transitions.

Response: Agreed. But it is about gross transitions and transition matrices here and as IAMs/LUCMs provide the transition matrices, they need to derive the exact source category somehow as well. How will this be done? As the reviewer mentioned earlier, there are not many IAMs which actually project land use and land cover at a regular grid scale, but instead resample/downscale aggregated results. Thus, it very much depends on the sophistication of the allocation procedure, how 'accurate' the derived transition matrices will be. Additionally, these downscaled maps are usually not evaluated against observational data. Thus, by using transition matrices (and the ability of DGVMs/ESMs to incorporate them), the issue would be solved for ESMs/DGVMs, but did not disappear and requires further research.

Conclusions and recommendations

page 11, lines 17-20: this is an important point, but it hasn't been clearly demonstrated in the text, largely because the paper generally refers to LULCC as a whole.

Response: We will resolve this issue by applying the revisions mentioned in our previous responses and the response to reviewer #2, i.e. refining land-use and land-cover terminology and reorganization of the sections. Additionally, we will move the discussion about model integration into a 'outlook' section and thus separating it from the more short-term recommendations.

page 11, lines 20-23: while not an individual, agent-based behavioral model, GCAM has been integrated with CESM as the iESM, implementing two-way feedbacks between the human and environmental systems, particularly for terrestrial systems and the effects on land projection. See the oft-noted paper above, which runs the iESM, and: Collins et al, 2015, The integrated Earth system model version 1: formulation and functionality. Geoscientific model development, 8, :2203-2219. There should also be a paper coming out soon on a complete experiment using the iESM to examine the effects of the feedbacks.

Response: We will accommodate this comment in the 'outlook' section of a reorganized manuscript by including a short discussion about efforts of model integration so far.

page 12, lines 1-19: It isn't clear how these suggestions relate to your three proposed primary issues with LULCC (which should also be restated in this conclusion – 1) uncertainty in lulcc data products is lacking due to not enough different products generated, 2) present-day lulc data are inconsistent and thus contain high uncertainty, and 3) uncertainty in lulcc projections is largely driven by initial data uncertainty over other model-specific sources). Issues 2 and 3 appear to mainly consist of data quality issues. Also note that your second main source on page 5 does not refer to gross or subgrid transitions at all, just to inconsistencies in present day data. Please make your conclusions/suggestions more consistent with the theme of the paper.

Response: See previous responses. The main challenges are meant to be (1) *Provision of a spatially explicit, continuous and consistent time series of LULCC*, (2) *Considering gross land-use changes* and (3) *Allocation of managed land in ESMs and DGVMs.* Related to these challenges we suggest to (1) Develop enhanced harmonized time series including uncertainty, (2) supplement the time series with estimations

of gross transitions and (3) the development of transition matrices based on sophisticated land-use models and empirical data.

Supplemental material

Figure S1 seems a lot more complicate with more steps than described in the text, which makes more sense.

Response: Please see response to comment pages 9-10, lines 30-12.

**Reviewer #2 (Anonymous)**

The authors discuss in their manuscript the major uncertainties and shortcomings associated with the implementation of land-use and land-cover changes (LULCC) in climate change assessments. Additionally, three major challenges are identified and the reasons for them are discussed.

General Comment: Generally, I think this paper raises some important issues related to implementation of LULCC in the modelling community. Raising awareness to this issues with the help of an extended literature review will be beneficial in tacking this problems.

Response: We thank the reviewer for the kind words and the overall positive evaluation of our manuscript.

However, I think the manuscript would benefit from not only raising awareness of the issues and the reasons behind it but also providing a 'way forward'. This is sometimes done in the individual sections but I think the suggestions get lost in the wealth of information presented in the manuscript. I therefore suggest adding a separate recommendation section in which the authors clearly state in short and precise form what could/should be done to overcome the challenges and which part of the community they see in a better position to take the lead (if possible), instead of having the conclusions and recommendations combined in one section.

Response: Indeed, the manuscript intends to provide guidance for the involved communities (i.e., IAM, LUCM, DGVM, ESM, remote sensing) on the 'ways forward' based on the different challenges identified. We aimed to do this by discussing each of the challenges within one section and bringing them together in the conclusion section with additional recommendations. However, we see that this could be done better with an improved guidance for the reader throughout the manuscript. We think that the reviewer makes useful suggestions how to reorganize the manuscript in a way that the intended structure becomes clearer. We therefore decided to follow these suggestions and propose to improve the manuscript by:

- Streamlining the discussion within the individual sections into 'description of the challenge', 'underlying reasons' and 'examples'
- Adding subheadings accordingly (e.g., 'Background', 'Underlying reasons', 'Example(s)'
- Moving the recommendations from the individual sections to a separate 'recommendation' section preceding the overall conclusion section, including an indication which part of the community we think is in charge of taking the lead

We feel that this reorganization of the structure will help to simultaneously clarify the main concerns of reviewer #1.

The abstract and P3L5-10 ('overall objectives) do not match. Please make sure that these points are consistent throughout the document.

Response: We will ensure consistency throughout the document by rephrasing the particular part of the abstract and adding a paragraph giving guidance on the structure of the manuscript according to the response to reviewer #1.

This brings me to my general comment; The paper covers a lot of material but very often the structure of the document gets lot. I therefore suggest the authors to go through the document and make sure that the reader can follow the train of thought easily. This could be achieved for example by adding subheading to the sections and having similar structure within each of the sub-sections. This also applies to the Supplements which should also convey a clear structure within the sub-sections.

Response: We will clarify the structure as outlined in the response to the reviewer's previous comments and hope to resolve the issue raised here in this way. Additionally, we will make sure to clarify the structure of the supplementary material.

Specific Comments: In the title of the manuscript it should become clear that the document is about 'anthropogenic' land-use and land cover changes and that the assessments are with regard to 'climate change'. I therefore suggest revising the title.

Response: We can add 'anthropogenic' to the title and exchange 'climate assessments' by 'climate change assessments' to be unambiguous. We suggest to revise the title (in addition to the changes suggested by reviewer #3) into

*'Current challenges of representing anthropogenic land-use and land-cover change in models contributing to climate change assessments'*

P1L20-23: Suggested to split sentence

Response: We suggest to split the sentence into

*'However, the processes and drivers of anthropogenic land-use activity are still overly simplistically implemented in Dynamic Global Vegetation Models (DGVMs) and Earth System Models (ESMs). The published results of these models are used in major assessments of processes and impacts of global environmental change such as the reports of the Intergovernmental Panel on Climate Change (IPCC).'*

P2L22-26: I think this is an interdisciplinary journal, it would be beneficial for readers outside of the community, if examples of the uncertainties could be given (e.g. it might not be clear to the readers what 'definition issues' are (see also P5L 16)).

Response: We agree that the 'definition issues' were not sufficiently explained in the text. We will thus add an explanation and examples, e.g.

*'Carbon fluxes are understood quite well for some compartments of the global carbon cycle, e.g. fossil fuel combustions and the ocean sink (Le Quéré et al., 2015), but the quantification of LULCC flux suffers from high uncertainties (Ballantyne et al., 2015) due to different definitions of individual land-use categories (e.g., what exactly is a 'pasture'), the definition of the land-use carbon flux (Pongratz et al., 2014; Stocker and Joos, 2015), the simplistic representation of LULCC in models that are used to quantify these fluxes (de Noblet-Ducoudre et al., 2012; Jones et al., 2013; Pugh et al., 2015) as well as the uncertainty about LULCC history (Ellis et al., 2013; Klein Goldewijk and Verburg, 2013; Meiyappan et al., 2012).'*

Accordingly on page 5:

*'While the rising operational application of remote sensing during recent decades has opened a powerful resource to map land cover on global scale, the exact definition of individual land-use and land-cover categories (e.g., Sexton et al., 2015), and difficulties in the distinction of them from the spectral response*

*(Friedl et al., 2010), lead to a variety of global land-cover products that do not agree about extent and spatial pattern of individual classes (Ban et al., 2015; Congalton et al., 2014).'*

P3L1: which 'sources'? Please specify

Response: The sources are listed in the previous paragraph. We will add a reference to make this unambiguous.

P3L27-31 Please be more specific and elaborate on the dataset

Response: Unfortunately, there is not a single dataset we could elaborate on here and we will add a clarifying statement in the revised manuscript. We argue that the diversity of models produce a tremendous amount of data, which is not necessarily consistent and comparable due to the underlying reasons listed in the preceding paragraph (i.e., two independent land-use change modeling communities (historical, future), different data sources, varying assumptions and drivers in the models.)

P4L6&L9: What does 'LUH' and 'HYDE' stand for?

Response: The reviewer is absolutely correct that we missed to spell out the acronyms. We will include the full terms in the revised manuscript.

*LUH* stands for *Land Use Harmonization*, a commonly applied acronym for the harmonized land use data set developed by Hurtt et al. (2011). *HYDE* refers to the *History Database of the Global Environment* (Klein Goldewijk et al., 2011), a historical reconstruction of population and land-use activity.

P5L18: 'large differences' in which variable?

Response: We refer to global land-cover products in this sentence, i.e. the variable is land cover or the individual classes (such as forest, grassland, etc.), respectively. We will rephrase to remove ambiguity. See the suggestion in response to comment P2L22-26.

P5L23: what are the 'all relevant processes' to the authors? Elaborate. What is still missing.

Response: This is indeed a reasonable comment and the elaboration on this question would probably easily fill another publication, which we think is out of the scope of our manuscript. To address this comment we will add a summarizing sentence with a list of references that have discussed these relevant processes in detail, e.g. Verburg et al. (2011) and Erb et al. (2016).

Some of the relevant processes might include temporal dynamics based on 'observational' data for non-forest land types, the distinction of 'natural' grasslands vs. intensively used pastures and additional information on land management practices (see Erb et al., 2016), urban/peri-urban development or a consistent change product including the main classes (forest, shrubland, grassland, cropland, pastures, urban,..) in general.

P5L26: It is not clear what a 'marker scenario' entails.

Response: We suggest to add a clarifying sentence and references in this paragraph, e.g.

*'A 'marker scenario' entails the implementation of a SSP scenario by one IAM that was elected to represent the characteristics of the qualitative SSP storyline best, while additional implementations of the same SSP, but by other IAMs, are 'non-marker scenario'. See Popp et al. (2016) and Riahi et al. (2016) for details.'*

P5L33: 'large variations' in which variable?

Response: The variable is 'pasture areas'. We will rephrase the sentence accordingly to make this clear.

*'For example, the projections of 11 IAMs and LUCMs show large variations in pasture areas in 2030 for many world regions (Figure 2, background map).'*

P6L5: The differences in output arising from different models the input and calibration etc. is not only an issue in the assessment of LULCC but generally applies to all models…Maybe if you look at other modelling communities and how they quantify these uncertainties.

Response: We entirely agree that these issues are not unique to the LULCC modeling community. However, it is not the purpose of the paper to describe detailed methods/metrics of how the uncertainties could be quantified. But in the paragraphs following this statement, we outline what would conceptually be needed to (1) quantify the uncertainty range and (2) suggestions how it could be reduced.

We argue that little attention has been paid to the evaluation of land use models and the quantification of the uncertainty in their projections. This leads to a situation where often model and input data related differences dominate the scenario related uncertainties. We acknowledge that there are activities starting towards quantifying the uncertainties related to model assumptions (Riahi et al., 2016). However, two main issues are not resolved yet: (1) the 'marker' implementation of SSPS are intended to be used for climate change assessments without providing an error/uncertainty range due to different model interpretations, which would allow to quantify the uncertainty propagation into the final assessments and (2) all 'marker scenarios' are interpreted by a very similar kind of models (IAMs), which – especially when it comes to the spatial pattern of LULCC projections (= input to DGVMs/ESMs) – largely differ from alternative realizations of land-use allocation models (Prestele et al., 2016).

P 6L11: the problem is how to define 'plausible' realisations.

Response: It is indeed a difficult task. However, there are different products available for global land cover only (e.g., the ESA-CCI product, the MODIS product) or integrated with land-use statistics (e.g., Fritz et al., 2015; Klein Goldewijk et al., 2011; Ramankutty et al., 2008), which show substantial variations in the extent and especially the spatial pattern of land-use distribution. To elect one of these to the 'best' might be difficult due to technical and analytical constraints. Nonetheless, we would assume them to be 'plausible' since they are based on sound science (either remote sensing directly and/or remote sensing and statistics) and we will add this deliberation to the manuscript.

P7L 2: Elaborate why sub-grid dynamics 'have been shown important'

Response: We will add a few examples summarizing the results in the referenced publications.

P7L6: what are 'under-determined mathematical systems' in this context?

Response: We will clarify this in the revised manuscript.

We refer to Hurtt et al. (2011) here who describe the derivation of land-use transitions as a 'large, under-determined mathematical system', i.e. the final state of LULC in year t2 depends on the state of LULC in year t1 and all the transitions in between. As usually no or insufficient information is available on the transitions, we have to make assumptions (e.g., cropland expands on grassland in a grid cell) or constrain the system by empirical data.

P7L8: what are 'minimum-transitions' in this context?

Response: Minimum transitions entail the changes between LULC categories only accounting for one-directional changes. We will add this explanation to the text.

P7L21-23: Rephrase sentence

Response: We suggest to rephrase as follows:

*'It is thus important that Figure 3 indicates also in large parts of the temperate zone and high latitudes substantial gross changes that may be underestimated by the currently used LUH dataset.'*

P10: rephrase 'allocation issue' do you mean the shifts between the communities and their perceived responsibilities

Response: 'Allocation issue' actually refers to the decision which land-cover type is replaced upon cropland and pasture expansion in the model. We suggest to rephrase to:

*'Providing such transition matrices, however shifts the decision which land-cover should be replaced upon cropland or pasture expansion from the DGVM/ESM community to the IAM/LUCM community.'*

P10 Maybe you would like to add to your decision that 'satellite data' is also not 'directly measured data' but also goes through a mathematical conversion process.

Response: That is indeed a good point and we suggest to add a sentence that explicitly mentions that satellite data is not directly measured data, too. However, we think compared to the modeled LULCC data, satellite data entails a much more 'directly measured' component in most cases and can contribute to the evaluation of land-use change models.

P11L24: Can you elaborate what 'improved communication' should entail in an ideal case. Additionally, I think it is not only the 'understanding' but first the 'awareness' of different assumptions and constrains needs to be achieved and before one can understand and tackle the problems.

Response: We believe improved communication eventually entails engaging in model integration as outlined in our conclusion. Simultaneously the individual challenges discussed in the manuscript need to be resolved through joint engagement across the communities in the individual tasks. Indeed, raising awareness of different assumptions and constraints is a first step and a major objective of the manuscript. By reorganizing the manuscript as described in our previous responses, we will ensure this becomes clear in a revised version, including recommendations which part of the community could most likely take the lead.

Figures: P24: add a 'log' label to the legend and add colour areas without change in grey to make the light yellow areas better stand out.

Response: We will revise as suggested.

Supplement SP2L7: can you provide more details on the updated version, i.e. reference

Response: We will add reference as far as possible.

The updated version is based on Ramankutty et al. (2008) for the static map of cropland distribution. There was however no additional publication related to the updated dataset. The dataset was available

from http://www.geog.mcgill.ca/nramankutty/Datasets/Datasets.html, but the webpage has recently been removed.

SP5: Can you comment on the uncertainties associated with the CORINE data

Response: We can add a short paragraph of uncertainties related to the CORINE data. We do not think that it would add important information to the manuscript, if we elaborate comprehensively on the uncertainty in the CORINE data. We actually only use the change product for illustration in terms of cropland transition trajectories. These aggregated results are probably not heavily affected by uncertainty in the CORINE data.

SP6L8: Can you elaborate on what the 'thematic accuracy' entails.

Response: 'Thematic accuracy' entails the capability of CORINE land cover maps to represent the 'true' land-cover class as compared to an independent validation data set (EEA, 2006). We will add an explanatory sentence and the reference in the respective section of the supplementary material.

**Reviewer #3 (Anonymous)**

The manuscript by Prestele et al., "Current challenges of implementing land-use and land-cover change in climate assessments", provides an overview of recent publications on interactions among land-use, carbon cycling, and different aspects of climate. First, the manuscript aims "…to identify existing shortcomings of the current LULCC representations within DGVMs and ESMs, reveal the underlying mechanisms and constraints that have hampered improved representations until now, and propose pathways to improve current representations" (page 3, lines 5-7). Second, based on the literature review, the manuscript attributes the lack of progress in including LULCC into climate assessments, to 1) the failure to account for uncertainty in reconstruction and future scenarios of gridded LULCC; 2) resolving sib-grid changes in land-use activities (e.g. gross transitions); 3) allocation of primary lands to managed lands in DGVMs and ESMs. Manuscript reviews a number of studies and discusses a wide range of limitations, specifically in CMIP5 historical reconstruction and future scenario. It has interesting discussion of how to use remote sensing data in improving treatment of LULCC processes in scenario development and its implementation into DGVMs and ESMs.

Response: The reviewer presents a good summary of the objectives of our manuscript. We thank the reviewer for the time spent on our manuscript and we appreciate the positive evaluation of our discussion about implementing remote sensing data in land – climate interaction studies.

For some of the following comments we sometimes split the original comments of the reviewer to address individual points.

However, the title is not appropriate because climate assessments such as IPCC do not implement LULCC – IPCC assessments review literature. CMIPs are not part of the IPCC, although their model simulations provide input to IPCC.

Response: The reviewer has a valid point here. Our current title is misleading, although we did not intend to equate the CMIP simulations with the IPCC assessment. We thus propose (in addition to the changes suggested by reviewer #2) to change the title into:

*'Current challenges of representing anthropogenic land-use and land-cover change in models contributing to climate change assessments'*

The manuscript has four major shortcomings: 1) While the manuscript reviews and synthesizes a number of recent studies on the development of scenarios of LULCC and of LULCC for climate and carbon cycling, it does not actually provide new insights or synthesis of LULCC implementation in ESMs and DGVMs. The manuscript provides a discussion of how the CMIP5 scenario was constructed and its limitations, but does not discuss differences in land use components of different ESMs or DGVMs. Or how they implemented the CMIP5 LULCC scenario. Table 1 gives 4 examples: 3 DGVMs (2 of which are variants of LPJ model) and a new HadGEM2-Jules ESM. There is no comprehensive analysis of CMIP5 ESMs or TRENDY DGVMs used in the AR5 in respect to LULCC. Thus, the manuscript's first goal is not supported by new insights beyond those previously published in literature.

Response: The reviewer raises a valid point that the wording of our main objectives leaves room for interpretation which requires clarification. It is not the purpose of the manuscript to focus on the technical details of land modules and/or LULCC implementation of ESMs/DGVMs. This has been done in

related publications (e.g., De Noblet-Ducoudré et al., 2012) as the reviewer states correctly. Instead, we provide a synthesis of issues arising along the chain of activities regarding land-use in climate change assessments (remote sensing, modeling, scenario development, implementation in DGVMs/ESMs). For that purpose we review the literature and identify the three challenges that comprise our three main sections. For each challenge, we show why it is a challenge, what implications this challenge can have for carbon cycling and climate assessments, and discuss limitations in the current approach to overcome this particular challenge.

In this way, the manuscript provides important guidance to the communities involved, as we bring together the individual points for the first time, including recommendations for potential ways forward. In fact, Table 1 should be regarded as an illustration to support our argument rather than a comprehensive analysis. We feel accommodating changes as outlined in the response to the other two reviewers will better clarify the value of our manuscript.

Additionally, please note: We intentionally did not submit a 'research article' but a 'short communication' type of manuscript, since it is intended as a guidance or perspective for future research, rather than new fundamental research.

2) The manuscript claims that the limited characterization of uncertainty in CMIP5 and CMIP6 LU reconstructions and scenarios is responsible for the lack of progress on LULCC in climate assessments. There is no reason to believe that's true. CMIP is designed to compare climate models and ESMs under a common set of forcings and capture model structural uncertainty. CMIPs never claimed to capture all uncertainty due to input forcing. It's a well-established practice in climate MIPs to provide a standard scenario for all forcings – greenhouse gases, short-lived species, solar, constants, volcanoes and LULCC, particularly over historical periods. Such GCM or ESM simulations are extremely computationally expensive. Permutation of alternative forcings datasets is not likely something that many climate centers will be able to engage and afford. The idea of multiple LULCC reconstructions advocated by the paper for CMIPs is not practical. If some modeling group/center wants to explore uncertainty due to LULCC, there is more than one scenario that is available even from the GLM model: Hurtt et al. (2006) included both scenarios based on SAGE and HYDE datasets. Hurtt et al. 2011 examines different assumptions in the GLM model.

Response: We do not seek to claim that 'the limited characterization of uncertainty in LU reconstructions and scenarios is responsible for lack of progress on LULCC in climate assessments' in our manuscript. We argue that the current characterization of uncertainty is insufficient and errors unaccounted for propagate into climate assessments. For example, LUMIP (contributing to the goals of CMIP6) aims to answer the scientific question '*What are the global and regional effects of land-use and land-cover change on climate and biogeochemical cycling (past-future)?*' (Lawrence et al., 2016), which we think can only be done if there is a sufficient quantification of uncertainty in the land-use forcing data set in place as well.

We agree with the point that CMIP is designed to compare climate models instead of forcing data sets, but it is similarly true that CMIP – due to its highly structured design – acts as a prototype for activities outside of CMIP and its forcing datasets (and as such the LUH) are widely used as a standard outside of CMIP, too. Please note, we do not restrict our arguments to the CMIP comparisons, but use it as an example at several places in the manuscript due to its high impact and pioneer role in the community.

We will make sure to clarify these distinctions in a revised version.

In our view a 'well-established' practice is not necessarily the same as 'best practice'. The fact that climate modeling centers cannot (or in some cases do not prioritize to) explore uncertainty in LULCC does not necessarily imply that it should not be done at all. If not practical in such a comprehensive way as proposed in section 2 of our manuscript, then the communities need to come up with alternative strategies to tackle these uncertainties, e.g. determining a minimum LULCC accuracy required for climate assessments using less computationally expensive DGVMs or offline land surface models (see our conclusion section). Harmonization to a common input for climate models is a major first step to have LULCC included in the climate simulations; in a next step the communities need to find a way to systematically approach the related uncertainties. Here our recommendations and conclusions could provide guidance on how to move forward and we will rephrase them to be more specific.

The main bottleneck for improving LULCC characterization in the CMIP is poor representation of LULCC processes in GCMs and ESMs. Most CMIP5 ESMs or TRENDY DGVMs can't use the information available in CMIP5 or CMIP6 historical reconstructions or future scenarios. For example, most of the CMIP5 models use only information about land use fractions, and not gross transitions provided by the Hurtt et al. (2011) data set. With the exception of very few models, ESMs do not represent shifting cultivation or wood harvesting.

Response: We agree that one of the major issues in land use – climate interaction studies is the 'poor representation' of LULCC in GCMs and ESMs. In fact, we identify it as one of the main issues in our manuscript as well (section 4) and will emphasize it additionally following the suggestions of reviewer #1 and #2. Given the low certainty in inputs of additional products such as wood harvest and shifting cultivation (Erb et al., 2016; Hurtt et al., 2011), inclusion of the processes in ESMs is not necessarily the only bottleneck. Simultaneously, we do not agree that a 'main bottleneck' (e.g., the poor representation) justifies neglecting other important issues (such as the uncertainty in LU modeling and the gross transitions) we raise in the manuscript. Instead, the LU modeling community should clearly communicate these issues as well and take the lead on improving the products.

Another unsupported assumption in the manuscript is that, by making additional ESMs or GCMs with alternative representations of LULCC history, one would get a better handle on the uncertainty in climate feedback of LULCC. It's not necessarily true: most studies with and without LULCC typically find a small difference in global climate and small regions with statistically distinguishable differences in climate characteristics. One would need a large ensemble of such simulations to find differences between the biogeophysical effects of alternative LULCC reconstructions and scenarios, unless they are really different as in future scenarios. Biogeophysical differences should be more pronounced, but the problem is that CMIP5 or even CMIP6 ESMs are incapable of representing major LU processes such as shifting cultivation, wood or crop harvesting..

Response: Regarding biogeochemical effects it has been shown that alternative reconstructions make a large difference to carbon emissions (e.g., Bayer et al., 2016; Meiyappan et al., 2015). From these findings we derive that alternative reconstructions are 'likely to substantially impact' LULCC – climate interactions and propose to determine a minimum accuracy that would be required from LULCC time series to not significantly impact on ESM output. For example, the work of Kaplan et al. (2011) and Fuchs et al. (2013) has shown that historical reconstructions can substantially differ from HYDE, thus we think it is an important scientific question how alternative reconstructions could affect the climate signal. Additionally, the uptake of a high/low estimate of historical land use in LUMIP (Lawrence et al., 2016) after feedback from the community (see the open discussion of the LUMIP paper, Lawrence et al. (2016),

in *Geoscientific Model Development*), indicates that the uncertainty in LU products is indeed an important issue that needs to be further explored.

3) The manuscript questions assumptions in CMIP5 Hurtt et al. 2011 reconstruction and future scenario. The Hurtt et al. (2011) effort, for the first time, harmonized historical reconstruction with the 4 Representative Concentration pathways (RCP) scenario and took into account gross transitions between different LU types in both tropics and extra-tropics. The authors are mistaken in their assumption that no-shifting cultivation in the extra-tropics implies no gross-transitions in the extra-tropics; for example, non-zero transitions between pastures to crops and crops to pastures. Furthermore, for CMIP6 (Lawrence et al. 2016), there will be a focus and additional LUH reconstructions available, as well as more details about the relationship between land cover and land use categories. I think a lot of criticism of the CMIP5 LULCC reconstruction and scenario is valid but the authors are overlooking improvements in the new reconstruction for CMIP6, which is publicly available now on the CMIP6 website.

Response: We acknowledge the effort of the Hurtt et al. (2011) harmonization activity and its contribution to enhance LULCC representation in land use – climate interaction studies. But, as the reviewer states later on, there are also limitations in the CMIP5 product. We do not assume that gross transitions can only appear in the tropics due to shifting cultivation (see P7L17-20), but argue that due to the resolution of the minimum transitions, gross transitions, especially in the temperate zone and the high latitudes, might be missed (P7L11ff.).

In terms of CMIP6 and the LUH2 product, we do not overlook the improvements, but explicitly mention the update (P5L6). However, we admit it is extremely difficult at the moment to follow the improvements compared to the LUH CMIP5 product, since documentation of the new products is restricted to a rather generic description (Lawrence et al., 2016) and thus we might miss some details. We are aware that the new historical product is publicly available, but the final dataset does not allow to trace back how the individual processes were implemented. To our best knowledge – apart from the indisputable improvements between the two products (e.g., additional focus on land management, improved shifting cultivation estimate) – some of our main criticisms (e.g., gross transitions in the extra-tropics, derivation of LU transitions) will be untouched even with the new product.

While it's possible to construct more detailed scenarios for recent periods with satellite coverage or for specific countries (e.g., Table 2 in the manuscript), particularly in the Northern Hemisphere, it is difficult if not impossible to develop multi-century reconstructions on a global scale with consistent sets of assumptions. Making simple assumptions in ESM is not an unreasonable approach for global, multi-century analyses. Assuming transitions based on the satellite era for the entire CMIP-style experiments may be problematic, as well, for pre-industrial or future periods.

Response: We agree that making simple assumptions is a reasonable approach to get started with the land-use implementation in DGVMs/ESMs and that the satellite era might not be representative for multi-century analysis as well. However, at least for this era the model assumptions should be carefully evaluated. Based on sufficient transition information for present-day, these assumptions could be, e.g., gradually replaced over time (backward and forward) based on scenario assumptions or regional characteristics (e.g., Fuchs et al., 2015). Sensitivity analysis could provide additional insights how individual decisions affect the land-use pattern, even over long time periods. In such a way the models would account for spatio-temporal variability in land-use transitions.

4) The rationale for including analysis from the CLUMondo model is not clear – it demonstrates how spatio-temporal variations could be different within the grid. It does not show that such patterns will affect climate or carbon cycling. Besides the CLU-Mondo analysis, there is now new analysis in this manuscript. So, there are no new insights/analysis, just a synthesis of other studies, which are already partially covered by the authors in related publications (e.g., Alexander et al. 2016, Bayer et al, 2016, Prestele et al. 2016).

Response: We will add a sentence to our objectives , which explicitly mentions the illustrative purpose of the CLUMondo analysis (see our general response for a suggestion). Specifically, the rationale for including CLUMondo analysis is to show that a simple allocation algorithm (such as forest will be cleared upon cropland expansion) applied globally might not sufficiently account for the spatio-temporal heterogeneity in the change patterns. Previous publications have shown that these decisions can affect regional climate and carbon cycling (e.g., de Noblet-Ducoudre, 2012), and thus our analysis should be taken as an illustration that further research is required on how these decisions affect the ESM results.

As mentioned in previous responses, the manuscript does not aim to present comprehensive new analysis, but rather use illustrative analysis using the CLUMondo model to support our arguments. In doing so, we provide a synthesis of currently untackled, or insufficiently tackled, challenges at the interface of land-use and climate modeling, and try to present guidance for the communities involved.

I think the most interesting part of the paper is the section on remotely sensed data (high and low resolution) in development of new diagnostics for evaluation of global LULCC reconstructions or models. Perhaps the authors can re-frame their analysis and demonstrate how such data can be used to improve or evaluate reconstructions (e.g. the one in CMIP6) or to create new diagnostics to evaluate ESMs and DGVMS.

Response: In our view, our manuscript brings together three major challenges/issues at the interface of land-use and climate modeling, which can serve as a guidance on the 'ways forward' to the communities involved – we therefore do not share the opinion that the remotely-sensed data should be the chief focus of the paper. Certainly there is also a need to develop new diagnostics as mentioned by the reviewer, but this is beyond the scope of this current paper. However, the reviewer raises a fair point and we will add this need to the 'outlook' section of the revised manuscript.

[revised manuscript text omitted]

---

## Author Response (AR1)

Dear Dr. Perdigão,

Thank you very much for the time and effort spent on our manuscript and the open discussion, as well as the provision of additional helpful comments how to improve our manuscript.

We revised the manuscript as detailed in the response to the reviewers during the open discussion including

- A revision of the title to *Current challenges of implementing anthropogenic land-use and land-cover change in models contributing to climate change assessments*,
- Clarification of the structure (including subsections) and providing additional guidance about the framing of the manuscript in the introduction and throughout the individual sections,
- A clearer distinction between sections 3 (*Challenge 2: Considering gross land-use changes*) and 4 (*Challenge 3: Allocation of managed land in TBMs*),
- A separate recommendation section (*Recommendations for improving the current LULCC representation across models*) that summarizes the 'ways forward' we propose to the individual challenges,
- An outlook section (*Outlook: towards model integration across disciplines*) that discusses model integration as a long-term goal
- Additional discussion and justification on controversial statements in the original manuscript where possible (e.g., on the gross change challenge)

Please find below a short summary of major changes to the manuscript and how they relate to the major concerns raised by the referees, followed by a detailed point-to-point reply to all individual comments and a marked-up version showing all changes eventually made to the manuscript. Page/line references in our response refer to the clean version of the revised manuscript.

With kind regards,

Reinhard Prestele

**General Response**

First of all we would like to thank the three reviewers for their time and effort they spent to review our manuscript. The detailed and constructive comments certainly helped us to improve the manuscript.

The first two reviewers are supportive of the paper, acknowledge its importance and we were able to revise the manuscript in such a way that all comments and requests are met. The comments of reviewer #3 have a different nature. We were able to address the more specific comments of reviewer #3, but disagree with the overall comment of this reviewer on lack of relevance of the paper.

Reviewer #1 suggests a reframing of the manuscript from our 'three challenges' and paying more attention to the distinction between land use and land cover or their separate research history, respectively. These are valid points which helped us to improve the manuscript. We resolved this by more clearly describing the framing of the paper. For example, there was room for misinterpretation of the structure in the original manuscript: while we intended to present one important challenge ('main issue') per section, the reviewer identifies the three main challenges already in section 2 (*Provision of spatially explicit, continuous and consistent time series of LULCC*) and thus the other two sections seem to be disconnected. We therefore included a guiding paragraph at the end of the introduction to clarify our underlying structure as following:

*'Each of the following sections presents one of the three challenges we identify to be crucial in future land use – climate interaction research, reviews the issue and its implications for the results of modeling studies based on previously published literature and in the context of the widely applied Land Use Harmonization (LUH) dataset published by Hurtt et al. (2011). In section 5 we propose pathways to improve the current LULCC representation for each of the challenges and conclude with an outlook on future research priorities.'*

Reviewer #2 comments broadly concentrate around hiddenness of the 'ways forward' in the individual sections. We addressed this in the revised version following the reviewer's suggestions and included sub-headings, re-organized the (sub-)sections towards a common structure and moved the 'ways forward' to a separate section and also slightly expanded on these.

Reviewer #3 concludes that we do not provide 'new insights, synthesis or analysis' and suggests a reframing of our manuscript towards the 'development of new diagnostics for evaluation of global LULCC reconstructions or models'. We do not share this view. The current manuscript is a review, supported by data (e.g., CORINE and NLCD land cover data in section 4) and new analysis (e.g., allocation of managed land as shown by CLUMondo in section 4) of the most dominant issues related to land use and climate modeling. The manuscript brings together, to our knowledge for the first time, prominent challenges in land use – climate interaction studies. This view is shared by the other reviewers. The purpose of the paper is to review the current problems and progress, and identifying possible ways forward for the community. We do agree with the reviewer's assessment that identifying more conclusively the ways forward to overcome existing stumbling blocks adds to the value of the manuscript and revised the paper to strengthen that aspect.

For a detailed list of how we addressed the individual reviewer's comments and improved the manuscript, please see our following responses inline in blue.

**Reviewer #1 (A. Di Vittorio)**

The authors examine current practices of providing LULCC data and simulating LULCC effects on the earth system in global models and identify 3 main issues related to reliable provision and use of LULCC data. They then move on to the limitations in data for gross land use/cover transitions, and finally discuss land conversion assumptions in modeling as a source of uncertainty. They also make 3 suggestions on how to improve the provision and use of LULCC data.

I appreciate this paper and am pleased that it discusses relatively overlooked, yet very important, issues regarding LULCC and global modeling. I generally agree with the assessment, but think that the paper needs some reorganization and some additional discussion to fully and clearly make its case. The main issues requiring attention are summarized here, with specific comments/suggestions following:

Response: We thank the reviewer for the kind words and the appreciation of our work. We are pleased that the reviewer generally agrees with our assessment. We revised the paper to provide a more clear structure as we feel the current framing provided some misunderstanding: we did not aim to first identify the three main issues and then 'move on to limitations in data for gross land use/cover transitions, and finally discuss land conversion assumptions in modeling', but the three main issues are intended to be the three sections following the introduction, namely (1) *Provision of continuous and consistent LULCC time series*, (2) *Consideration of gross land-use change*, and (3) *Allocation of managed land in TBMs*. We refer to this and explain more in the responses to following comments.

1) The paper needs a consistent framing and argument. The three main issues are different between the abstract, text and conclusion. It appears (see abstract lines 28-34 and page 3 lines 5-10) that the point is to show that the 3 main issues are indeed main issues, based on literature, an example, and discussion of two underlying factors ( 1) gross transitions and 2) land use change to land cover change translation ), and provide suggestions for moving forward. But the main issues are not referred to in the later sections, and these two aspects are not introduced up front so that they can be discussed in the context of the main issues. And the suggestions are not related to the issues.

Response: Based on the assumptions the reviewer made regarding the three main issues and the underlying factors, we entirely agree that there is a lack of consistency and disconnection throughout the manuscript. We added a paragraph at the end of the introduction that clarifies the structure (page 3, lines 13-19; please see the general response). We removed the confusing sentences that give the impression that we present the 'main issues' in the first section (after the introduction) and discussing 'underlying factors' subsequently. The revised manuscript ensures that each section (2, 3, 4) treats one of the challenges outlined in the abstract and picked up in the recommendation section individually, connected by the overarching topic of 'open issues in land use – climate interaction studies'.

In conjunction with the changes outlined later on in this document regarding structure requested by reviewer #2, we believe that we have addressed the concern raised.

2) It isn't clear that lack of information on gross transitions is a fundamental factor for the 3 main issues. While there is a lot of uncertainty in estimating gross transitions, and there is need to improve related data, this seems more like an example of a more fundamental driver. One thing that cuts through the three issues and incorporates gross transitions is data quality. In fact, that is largely what issue 2 in the text describes. And ultimately issue 3 as well (initial, present-day data sets for future projections). Maybe there are only two main issues (single historical product with no uncertainty and uncharacterized/large model uncertainty in future land projections) and two underlying factors (data quality and independent land use

and land cover implementation). Then the underlying factors provide guidance for the two communities to work together to address the two issues as they apply to both the human and dgv/es models.

Response: We rephrased the section in such a way that we no longer present the lack of information on gross transitions as a fundamental factor for other issues/challenges. We strengthened our view that gross transitions are a challenge in itself, since these additional changes are hardly considered in climate models, but may have a substantial impact, e.g. on the resulting carbon signal. Indeed, it closely links to both of the other challenges, to the harmonization (section 2) in terms of data quality and to the allocation (section 4) in terms of how the DGVMs/ESMs are able to implement the transitions. However, the reframing suggested by the reviewer would have made the gross transitions to one out of several 'data quality' issues, which we feel is not appropriate. The topic of gross transitions goes beyond data quality issues due to their scale dependency, i.e. even if we would be able to reduce uncertainties due to technical restrictions (e.g., in remote sensing products) or model assumptions, sub-grid processes would need to be addressed separately. Similarly, the suggested reframing would exclude the uncertainty from present-day land-use (and land-cover) products which different land-use model intercomparison exercises have identified as an important source of variation across land-use models (Alexander et al., 2016; Schmitz et al., 2014). We however agree that challenge 1 (*Provision of consistent time series*) is mainly related to data quality, which we emphasized in the revised version of the manuscript.

3) The underlying factor of the traditional separation of land use research from land cover research is not addressed until section 4, even though it cuts through the main issues and there is also a main point in the conclusion that land use modeling needs to be integrated with land cover/ecosystem modeling. And one of the suggestions calls for specific land use to land cover conversion information in place of just land use information. You also use land cover products for figure 5, which are not necessarily consistent with agricultural land use data. Furthermore, this separation is not explicitly discussed, with LULCC being a whole throughout the text, even when discussing how each land model has to make land cover conversion assumptions to accommodate independent land use data. You mainly focus on the land cover conversion uncertainty, but the separation of land use and land cover is the underlying source. There is some additional literature addressing this specific issue that would be useful to the authors. I would also be happy to discuss this further with the authors, as I am trying to finish a manuscript looking at how land cover conversion uncertainty affects carbon and climate projections. Look me up if you are coming to AGU in San Francisco this year.

Response: The reviewer has a valid point that we did not carefully separate between land use and land cover throughout the manuscript and especially in section 4. We checked again and changed accordingly by spelling out land use and/or land cover where applicable. As mentioned later on in the response to reviewer #2, we additionally clarified the structure such that a recommendation section focuses on the short term improvements (i.e., how to tackle the individual issues within the current offline coupling of models) and an outlook section discusses the model integration. We do not necessarily agree that the separation of land use from land cover research 'cuts through the main issues', but it rather applies to our issue 3 (*Allocation of managed land in TBMs*). In addition to this, we discuss the issues arising in the data used for the current coupling in section 2. We clarified this in a revised manuscript as outlined in the response to the reviewers first general comment.

Specific comments and suggestions:

Abstract

page 1, lines 31-32: this subgrid and gross transition is not on page 5 as a main source of uncertainty. The second main source in the text is inconsistencies of present day data. You do later discuss transitions, and make a statement in the conclusion, however.

Response: See response to this reviewers main comments.

page 2, lines 1-2: I think I know who you mean (providers and users), but it is unclear who is included in the "joint development and evaluation" here.

Response: Thanks for pointing to that. We rephrased this sentence to clarify as following (page 1, lines 32-35):

*'We propose that LULCC data-provider and –user communities should engage in the joint development and evaluation of enhanced LULCC time series, […]'*

page 2, line 12: What do you include as a DGVM here? Some consider any model having vegetation growth in response to environmental conditions as a DGVM. For others a DGVM specifically includes prognostic biogeography (i.e., the extent of vegetation types change according to environmental conditions) and/or successional vegetation processes (e.g., stages of forest stand growth following a forest clearing disturbance).

Response: We agree with the reviewer that the DGVM terminology is misleading in our manuscript. We replaced it with the term *'terrestrial biosphere model (TBM)',* which covers models with both static and dynamic vegetation. Additionally, we added a short explanation that makes clear that this term is meant to be inclusive of a range of modeling approaches (page 2, lines 10-12), as basically any terrestrial biosphere model that is able to include land use from an external data set will face the challenges discussed in our manuscript. As models with dynamic vegetation can substantially differ from land surface models coupled to ESMs in the exact implementation of external data, we differentiate between DGVMs and ESMs in our section 4.

page 2, line 26: you should also cite: Meiyappan and Jain, 2012. Three distinct global estimates of historical land-cover change and land-use conversions for over 200 years, Frontiers in Earth Science, 6(2):122-139 (I noticed you cite it later).

Response: Good point. We included the reference (page 3, line 3).

page 2, lines 29-30: you should also cite: Di Vittorio et al., 2014. From land use to land cover: restoring the afforestation signal in a coupled integrated assessment-earth system model and the implications for CMIP5 RCP simulations, Biogeosciences, 11, 6435-6450.

Response: We apologize that we have not been aware of this work during the preparation of the manuscript. We included the reference (page 2, line 32).

Provision of LULCC

page 4, lines 11-13: The CMIP5 product harmonizes only land use, and as such the land cover (forest, grass, etc.) and how it is altered by land use is determined independently by the DGVMs/ESMs, and can be dramatically different between models for a given scenario (in fact, prescribed scenarios can be substantially altered in ESMs by this, see Di Vittorio paper listed above). The CMIP6 product is also including forest cover in the harmonization, both for the historical period (with reference to satellite data) and for the IAM scenarios (which actually project all land cover).

Response: We agree that the term LULCC is not appropriate in this context. We improved this paragraph by exchanging the term 'LULCC' by 'land use'. Furthermore, we replaced the sentence

*'This strategy tries to conserve the original patterns, rate and location of change as much as possible, and to reduce the differences between the models due to definition of land-use categories (e.g., what constitutes a forest).'*

by

*'The harmonization tries to conserve the original patterns, rate and location of change as much as possible, and to reduce the differences between the models due to inconsistent definitions of cropland, pasture and wood harvest.'* (page 4, lines 26-28)

page 4, lines 14-15: Only land use is input to and output from GLM for CMIP5, and forest cover is included for CMIP6.

Response: We revised as mentioned in the response to the previous comment.

page 5, line 12: It isn't clear here that you have shifted away from the harmonization group of models to a more general group providing present-day lulc data for future projections.

Response: It is not really a shift from the harmonization group of models to a more general group, but we are not aware of work that has compared the differences in the starting maps of the IAMs contributing to the CMIP5 harmonization only. The cited Prestele et al. (2016) paper indeed includes additional (non-IA) models, but still three out of four models that provide data to the CMIP5 harmonization. These do not agree about initial areas as well. We clarified as following (page 6, lines 1-2):

*'These differences propagate into the starting maps of the various land-use change models providing land-use data to climate models, including the IAMs providing data for the LUH.'*

pages 5-6, lines 24-5: Two points here: 1) In the IPCC context, only land use was used, with forest cover coming into play for CMIP6, even though the IAMs project land types for the entire terrestrial surface. This introduces uncertainty beyond just the model structure/assumptions and different input data (see the Di Vittorio paper listed above). 2) The starting point of lulcc determination isn't just about which land-use input data or how processes are implemented. The spatial configuration of these data and the model are key factors in determining model outcomes. And each model has a unique spatial configuration. Gridded models/data do not necessarily resolve this spatial issue because regional values are often just resampled to the grid. See: Di Vittorio et al, 2016. What are the effects of Agro-Ecological Zones and land use region boundaries on land resource projection using the Global Change Assessment Model, Environmental Modelling and Software, 85:246-265

Response:

1) We agree with the reviewer about the missing land cover information being a major constraint in the CMIP5 LUH product and expanded the discussion about the missing land cover information in CMIP5 LUH in the section *Allocation of managed land in TBMs* (page 9, lines 31 ff.). We do not think that the discussion about uncertainty emerging from the fact, that only land use is provided to DGVMs/ESMs and they utilize different implementation schemes on their background land cover, would fit in section 2 where we discuss the uncertainties arising during land-use modeling and the harmonization.

2) We agree that the spatial configuration of the individual models even increases the uncertainty range and appreciate the reference to the Di Vittorio et al. (2016) paper. We included this additional uncertainty issue by rephrasing page 6, lines 11-14 to:

*'Land-use change model intercomparisons and sensitivity studies, however, indicate that the uncertainty range emerging from different assumptions in the models, input data, and spatial configuration substantially impacts the model results (Alexander et al., 2016; Di Vittorio et al., 2016; Schmitz et al., 2014).'*

page 6, line 2: How were the variables normalized? Could the dominance of initial pasture area be due to it just being the largest difference in relation to the other variables? Also, it would be more clear if you were specific in the text and the caption in describing that the "starting point" and "initial" are the pasture area in relation to fao in 2010, and that "model" is actually model type and presumably the spatial resolution/configuration.

Response: We included a clear mentioning how the data were processed in the supplementary material (Supplement S2.1) and improved the caption of Figure 2 as suggested. The areas are indeed in absolute terms, which indeed makes it likely to have larger deviations, if regression coefficients were interpreted directly. However, in the text we describe the relative contribution of each group of variables in the regression model (initial, model, scenario and residual; Table S2) to the total variation in pasture areas. The relative contribution is derived by dividing the sum of squares related to groups by the total sum of squares obtained from an ANOVA (type II sum of squares, i.e. not dependent in which variables are considered in the model; see Alexander et al., 2016) applied to the regression results. These are not affected by the larger difference in the initial deviation.

page 6, lines 6-14: I completely agree! While recent feedback on LUMIP has prompted the provision of LU-forest uncertainty along with the CMIP6 LUH product, it still falls short of the comprehensive approach discussed here.

Response: We thank the reviewer and also appreciate initial consideration of uncertainty along with LULCC data provided to LUMIP, e.g. that additional high and low estimates of historical land use will be included (Lawrence et al., 2016). However, we think a more ambitious approach should be taken into consideration for future MIPs.

page 6, lines 15-22: The separation of land use and land cover is a critical factor omitted from this discussion. While land use and land cover are often said in the same breath and the LULC(C) acronym is widely used, in nearly all cases people are referring to either land use or land cover. Research is clearly split along these lines, and land use data are remarkably inconsistent with land cover data. Land use and land cover need to be studied together, as an integrated process, in order to reduce LULCC uncertainties and inconsistencies between these two groups of data.

Response: We agree with the reviewer that there is a large discrepancy between land use and land cover research and data products and that this will be an important challenge to overcome in future by integrating the research lines. In fact, model integration is one of the key points of our conclusions. We highlighted this point more clearly in the revised manuscript by rephrasing our three conclusion points in section 5 (*Recommendations for improving the current LULCC representation across models*) and additionally discuss the model integration in section 6 (*Outlook: towards model integration across disciplines*).

Considering gross land use changes

How does this relate to the three issues in the previous section? Is this really a major driver of the 3 issues, or something along for the ride? It is clearly present in issues 1 and 2 (although the present day isn't discussed, only past and future), while its absence in IAM projection may be the relevant link (as the transitions are determined by a single independent model, which is part of issue 1)

Response: See earlier responses. We clarified in the text that we do not argue that missing gross transition representation is a major driver of the uncertainty in land-use data sets used for the harmonized time series. Separate from the uncertainties discussed in the previous section we argue that gross transitions are an issue that have not got enough attention throughout the communities apart from shifting cultivation in the tropics.

page 7, lines 21-23: Just a note: You are well aware that gross transition information is highly uncertain, and current work suggests that the CMIP5 LUH data product may actually overestimate gross transitions in the tropical regions.

Response: Indeed, gross transition information is uncertain, especially if it is provided by models that have multiple other sources of uncertainty (as discussed in section 2) or simple assumptions about the spatial distribution of shifting cultivation have to be made (as in the CMIP5 harmonization). For this reason, we call for additional work on this topic, based on empirical data, such as the updated shifting cultivation estimate for the CMIP6 LUH product (which the reviewer probably refers to here) or the recent work by Fuchs et al.

page 7, line 32: there are no land cover categories in CMIP5 LUH, only primary and secondary land. wood harvest is associated with forest or non-forest, but this land cover designation is based on a threshold of a potential biomass model, rather than more commonly used land cover or potential vegetation data sets.

Response: We removed 'land-cover' from the sentence, as it is indeed misleading along with the CMIP5 LUH data. We additionally moved the discussion about the derivation of explicit transitions to section 4 to distinguish between gross changes and explicit transitions. The sentence now reads (page 12, line 24-25):

*'For example, urban expansion is applied proportionally to cropland, pasture and (secondary) natural vegetation. Upon transitions between natural vegetation and agricultural land, choices in the model configuration have to be made, whether primary or secondary land is converted preferentially.'*

Moreover, even if forest/non-forest land cover is provided for CMIP6, the decision how to derive the transitions (e.g., forest to crop or non-forest to crop) has to be made? We thus believe our main argument in this paragraph about the simplistic assumptions to derive transitions remain unaffected.

page 8, line 14: "…increasingly been captured…"

Response: Corrected accordingly.

Allocation of managed land in ESMs and DGVMs

Ah, finally! This aspect of separate land use and land cover information/modeling is a factor in all 3 of your main sources of uncertainty, and as such needs to be mentioned up front and related to these uncertainty sources.

Response: See the general response, previous responses and the response to reviewer #2 how we clarified the structure in the revised manuscript to address this comment. In short, section 2 now focuses on the uncertainties within the land-use and land-cover data provider community ('data quality') and its implications for climate assessments upon coupling to DGVMs/ESMs. Section 3 picks up an important, but so far hardly considered challenge (gross changes), while section 4 discusses the issues arising when using external land-use data in DGVMs/ESMs, including the underlying source of the separated land-use and land-cover research lines.

page 8, lines 27-28: and scenarios and over relatively short time periods (see Di Vittorio et al 2014)

Response: We rephrased this paragraph. It now reads (page 10, lines 4-9):

*'The decision is important as it impacts the distribution of the natural vegetation in a grid cell, as well as the mean length of time that land has been under a particular use, with consequences for both the biogeochemical and biophysical properties (Reick et al., 2013). For example, new cropland expanding on forest would lead to a large and relatively rapid loss of ecosystem carbon due to deforestation, while cropland expanding on former grassland would have a less immediate impact on ecosystem carbon stocks through soils. Likewise, the albedo and partitioning of energy differs strongly between forest and grassland land covers (Mahmood et al., 2014; Pielke et al., 2011).'*

page 9, lines 26-27: this is consistent with land cover being studied separately from land use, and your examples also relate to your second main source of uncertainty.

Response: The reviewer is correct that the overview of these studies also show that there is a mix of approaches in studying current land-use and land-cover changes that necessarily leads to some kind of uncertainty. However, this statement was intended to emphasize that we did not conduct a systematic review, but rather use the studies as an illustration to support our argument that the change pattern is spatially heterogeneous.

pages 9-10, lines 30-12: glad you did this! But what determines the source of land use in CLUMondo? It is important to clearly state how this model differs in this selection versus those that use the methods by which you classify the changes. Generally, more info is needed regarding how the different classified algorithms are defined, in relation to how they are implemented in dgvms/esms. The reader should be able to understand what is going on without digging through the supplemental material. Maybe a table of the definitions?

Response: We included some more detail in the main text (page 11, line 10 onwards; see below) and added a table with the definitions (Table 3). In the original manuscript we aimed to keep methodological descriptions as short as possible, as we apply the analysis mainly for illustrative purposes.

*'CLUMondo models the spatial distribution of land systems over time, instead of land use and land cover directly. Land systems are among others characterized by a mosaic of land use and land cover within each grid cell. The land systems are allocated to the grid in each time step 'based on local suitability, spatial restrictions, and the competition between land systems driven by the demands for different goods and services' (Eitelberg et al., 2016; Van Asselen and Verburg, 2013). Thus, the determination of the source land use or land cover upon cropland expansion can be interpreted as a complex algorithm taking into account external demands, the land-use distribution of the previous time step, local suitability in a grid cell and neighborhood effects. This strategy differs from the one in TBMs in a way that not one simple rule is applied to each grid cell equally, but accounts for the spatial heterogeneity of drivers of land-use change.*

*In order to compare the sources of cropland expansion in CLUMondo to the globally applied rules in TBMs, we reclassified the outputs of a CLUMondo simulation (FAO3D, Eitelberg et al., 2016) according to their dominant land-use or land-cover type to derive transitions (Table S6) and classified the changes within each ca. 0.5 x 0.5 degree grid cell as either grassland first, forest first, proportional, or a complex reduction pattern (Table 3;Figure S2-3 and additional explanation in Supplement S2.4).'*

page 10, lines 5-8: what about the undefined category, which is the dominant category according to the figures (not the complex)? what does it stand for? are you grouping this with the complex category?

Response: Good point. We seem to have missed the undefined category somewhere on the line and added this piece of information to the text (page 11, line 22; see below), including a table that explains the individual categories (Table 3). In principle, undefined means that it was not possible to detect on of the algorithms (forest first, grassland first, etc.). We are not grouping it with the complex category, as this category basically entails the opposite: all major land-cover types have been available, but there is no priority which one is reduced upon cropland expansion.

*'Additionally, a grid cell was labeled 'undefined', if grassland or forest was not available in the source map.'*

page 10, line 16: the IAM community has been projecting land use AND land cover for some time, although not necessarily gross transitions.

Response: Agreed. But it is about gross transitions and transition matrices here and as IAMs/LUCMs provide the transition matrices, they need to derive the exact source category somehow as well. How will this be done? As the reviewer mentioned earlier, there are not many IAMs which actually project land use and land cover at a regular grid scale, but instead resample/downscale aggregated results. Thus, it very much depends on the sophistication of the allocation procedure, how 'accurate' the derived transition matrices will be. Additionally, these downscaled maps are usually not evaluated against observational data. Thus, by using transition matrices (and the ability of DGVMs/ESMs to incorporate them), the issue would be solved for ESMs/DGVMs, but did not disappear and requires further research. We now elaborate in more detail on this issue in section 4.5 (*Open issues of transition matrices*).

Conclusions and recommendations

page 11, lines 17-20: this is an important point, but it hasn't been clearly demonstrated in the text, largely because the paper generally refers to LULCC as a whole.

Response: We resolved this issue by applying the revisions mentioned in our previous responses and the response to reviewer #2, i.e. refining land-use and land-cover terminology and reorganization of the sections. Additionally, we moved the discussion about model integration into section 6 (*Outlook: towards model integration across* disciplines) thus separating it from the more short-term recommendations.

page 11, lines 20-23: while not an individual, agent-based behavioral model, GCAM has been integrated with CESM as the iESM, implementing two-way feedbacks between the human and environmental systems, particularly for terrestrial systems and the effects on land projection. See the oft-noted paper above, which runs the iESM, and: Collins et al, 2015, The integrated Earth system model version 1: formulation and functionality. Geoscientific model development, 8, :2203-2219. There should also be a paper coming out soon on a complete experiment using the iESM to examine the effects of the feedbacks.

Response: We accommodated this comment in section 6 (*Outlook: towards model integration across disciplines*; page 15, lines 15-20):

*'Integration of these different types of models, where anthropogenic activity on the land system is considered as an integral part of ESMs, instead of an external boundary condition, might help to reduce these uncertainties, although it will certainly further complicate the interpretation of model responses. For example, Di Vittorio et al. (2014) report first results of the iESM (Collins et al., 2015), an advanced coupling of an IAM and an ESM, implementing two-way feedbacks between the human and environmental systems, and show how this improved coupling can increase the accuracy of information exchange between the individual model components.'*

page 12, lines 1-19: It isn't clear how these suggestions relate to your three proposed primary issues with LULCC (which should also be restated in this conclusion – 1) uncertainty in lulcc data products is lacking due to not enough different products generated, 2) present-day lulc data are inconsistent and thus contain high uncertainty, and 3) uncertainty in lulcc projections is largely driven by initial data uncertainty over other model-specific sources). Issues 2 and 3 appear to mainly consist of data quality issues. Also note that your second main source on page 5 does not refer to gross or subgrid transitions at all, just to inconsistencies in present day data. Please make your conclusions/suggestions more consistent with the theme of the paper.

Response: See previous responses. The main challenges are meant to be (1) *Provision of a spatially explicit, continuous and consistent time series of land-use change*, (2) *Considering gross land-use changes* and (3) *Allocation of managed land in TBMs.* Related to these challenges we suggest to (1) Develop enhanced harmonized time series including uncertainty, (2) supplement the time series with estimations of gross transitions and (3) the development of transition matrices based on sophisticated land-use models and empirical data. We now explicitly mention them in the abstract, discuss each of them in one section (2-4) of the manuscript and propose 'ways forward' in section 5 (*Recommendations for improving the current LULCC representation across models*) for each challenge.

Supplemental material

Figure S1 seems a lot more complicate with more steps than described in the text, which makes more sense.

Response: Please see response to comment pages 9-10, lines 30-12.

**Reviewer #2 (Anonymous)**

The authors discuss in their manuscript the major uncertainties and shortcomings associated with the implementation of land-use and land-cover changes (LULCC) in climate change assessments. Additionally, three major challenges are identified and the reasons for them are discussed.

General Comment: Generally, I think this paper raises some important issues related to implementation of LULCC in the modelling community. Raising awareness to this issues with the help of an extended literature review will be beneficial in tacking this problems.

Response: We thank the reviewer for the kind words and the overall positive evaluation of our manuscript.

However, I think the manuscript would benefit from not only raising awareness of the issues and the reasons behind it but also providing a 'way forward'. This is sometimes done in the individual sections but I think the suggestions get lost in the wealth of information presented in the manuscript. I therefore suggest adding a separate recommendation section in which the authors clearly state in short and precise form what could/should be done to overcome the challenges and which part of the community they see in a better position to take the lead (if possible), instead of having the conclusions and recommendations combined in one section.

Response: Indeed, the manuscript intends to provide guidance for the involved communities (i.e., IAM, LUCM, DGVM, ESM, remote sensing) on the 'ways forward' based on the different challenges identified. We aimed to do this by discussing each of the challenges within one section and bringing them together in the conclusion section with additional recommendations. However, we see that this could be done better with an improved guidance for the reader throughout the manuscript. We think that the reviewer makes useful suggestions how to reorganize the manuscript in a way that the intended structure becomes clearer. We therefore decided to follow these suggestions and improved the manuscript by:

- Streamlining the discussion within the individual sections into 'description of the challenge and underlying reasons', 'examples', 'current solutions' and 'shortcomings of current solutions'
- Adding subheadings accordingly. For example, section 2 (*Challenge 1: Spatially explicit, continuous and consistent time series of land-use change*) has been re-structured into the following sub-sections:
    - *2.1 Background and emergence*
    - *2.2 Current approach to provide consistent data: The Land Use Harmonization (LUH)*
    - *2.3 Open issues in the LUH data and their implications for climate change assessments*
- Moving the recommendations from the individual sections to section 5 (*Recommendations for improving the current LULCC representation across models*), whereas each sub-section deals with one of the challenges identified in sections 2-4 and, including an indication which part of the community we think is in charge of taking the lead

We feel that this reorganization of the structure helped to simultaneously clarify the main concerns of reviewer #1.

The abstract and P3L5-10 ('overall objectives) do not match. Please make sure that these points are consistent throughout the document.

Response: We checked for consistency throughout the document, rephrased the particular part of the abstract (page 1, lines 26 ff.) and added a paragraph giving guidance on the structure of the manuscript according to the response to reviewer #1 (page 3, lines 13 ff.).

This brings me to my general comment; The paper covers a lot of material but very often the structure of the document gets lot. I therefore suggest the authors to go through the document and make sure that the reader can follow the train of thought easily. This could be achieved for example by adding subheading to the sections and having similar structure within each of the sub-sections. This also applies to the Supplements which should also convey a clear structure within the sub-sections.

Response: We clarified the structure as outlined in the response to the reviewer's previous comments. Additionally, we clarified the structure of the supplementary material, which now provides additional information in line with the structure of the main text. To improve guidance throughout the supplements, we added a reference to the section/figure/table that the particular supplement refers to in the main text

Specific Comments: In the title of the manuscript it should become clear that the document is about 'anthropogenic' land-use and land cover changes and that the assessments are with regard to 'climate change'. I therefore suggest revising the title.

Response: We added 'anthropogenic' to the title and exchanged 'climate assessments' by 'climate change assessments' to be unambiguous. We revised the title (in addition to the changes suggested by reviewer #3) into

*'Current challenges of implementing anthropogenic land-use and land-cover change in models contributing to climate change assessments'*

P1L20-23: Suggested to split sentence

Response: We split the sentence into (page 1, lines 21-23):

*'However, the processes and drivers of anthropogenic land-use activity are still overly simplistically implemented in Terrestrial Biosphere Models (TBMs). The published results of these models are used in major assessments of processes and impacts of global environmental change, such as the reports of the Intergovernmental Panel on Climate Change (IPCC).'*

P2L22-26: I think this is an interdisciplinary journal, it would be beneficial for readers outside of the community, if examples of the uncertainties could be given (e.g. it might not be clear to the readers what 'definition issues' are (see also P5L 16)).

Response: We agree that the 'definition issues' were not sufficiently explained in the text. We thus added an explanation and examples. The paragraph reads now (in line with changes according to main comment 1), reviewer #1 (page 2, lines 23 ff.):

*'[…] For example, carbon fluxes related to land-use change that increase the atmospheric concentration of greenhouse gases are the largest source of uncertainty in the global carbon budget (Ballantyne et al., 2015; Le Quéré et al., 2015). […] This current land-use representation is, amongst others, sensitive to the definition of individual land-use categories (e.g., what exactly defines a 'pasture'), inconsistencies in the definition of the land-use carbon flux (Pongratz et al., 2014; Stocker and Joos, 2015), the implementation and parameterization of land-use in TBMs (Brovkin et al., 2013; de Noblet-Ducoudre et al., 2012; Di Vittorio et al., 2014; Hibbard et al., 2010; Jones et al., 2013; Pitman et al., 2009; Pugh et al., 2015), the structural*

*differences across IAMs and LUCMs (Alexander et al., 2016; Prestele et al., 2016; Schmitz et al., 2014), and the uncertainty about land-use history (Ellis et al., 2013; Klein Goldewijk and Verburg, 2013; Meiyappan et al., 2012).'*

Accordingly page 5, lines 33 ff.:

*'The land-cover maps in turn disagree about extent and spatial pattern of agricultural land (Congalton et al., 2014; Fritz et al., 2011) due to both inconsistent definitions of individual land-use and land-cover categories (e.g., Sexton et al., 2015) and difficulties in identifying them from the spectral response (Friedl et al., 2010).*

P3L1: which 'sources'? Please specify

Response: The sources are listed in the previous paragraph. We added a reference to make this unambiguous (page 3, lines 4-5).

*'Currently reported uncertainties of the outputs of land use – climate interaction studies may be underestimated by insufficiently accounting for the aforementioned sources of uncertainty.'*

P3L27-31 Please be more specific and elaborate on the dataset

Response: Unfortunately, there is not a single dataset we could elaborate on here and we rephrased the paragraph to emphasize the variety of datasets provided from the various land-use change models (page 3, lines 32 ff.). We argue that the diversity of models produce a tremendous amount of data, which is not necessarily consistent and comparable due to the underlying reasons listed in the preceding paragraph (i.e., two independent land-use change modeling communities (historical, future), different data sources, varying assumptions and drivers in the models.)

P4L6&L9: What does 'LUH' and 'HYDE' stand for?

Response: The reviewer is absolutely correct that we missed to spell out the acronyms. We included the full terms in the revised manuscript (LUH: page 4, line 11; HYDE: page 4, line 24).

P5L18: 'large differences' in which variable?

Response: We refer to global land-cover products in this sentence, i.e. the variable is land cover or the individual classes (such as forest, grassland, etc.), respectively. We will rephrase to remove ambiguity. See the response to comment P2L22-26.

P5L23: what are the 'all relevant processes' to the authors? Elaborate. What is still missing.

Response: This is indeed a reasonable comment and the elaboration on this question would probably easily fill another publication, which we think is out of the scope of our manuscript. The paragraph the comment refers to has been removed due to the re-structuring of the manuscript. However, we addressed this comment by including discussion about separated history of land-use and land-cover research in section 4 and the model integration in section 6.

P5L26: It is not clear what a 'marker scenario' entails.

Response: We added a clarifying sentence and additional references in this paragraph (page 6, lines 8-11):

*'A 'marker scenario' entails the implementation of a SSP scenario by one IAM that was elected to represent the characteristics of the qualitative SSP storyline best, while additional implementations of the same SSP in other IAMs are 'non-marker scenario (Popp et al., 2016; Riahi et al., 2016)'.*

P5L33: 'large variations' in which variable?

Response: The variable is 'pasture areas'. We rephrased the sentence accordingly to make this clear (page 6, lines 18-19).

*'For example, the projections of 11 IAMs and LUCMs show large variations in pasture areas in 2030 for many world regions (Figure 2, background map).'*

P6L5: The differences in output arising from different models the input and calibration etc. is not only an issue in the assessment of LULCC but generally applies to all models...Maybe if you look at other modelling communities and how they quantify these uncertainties.

Response: We entirely agree that these issues are not unique to the LULCC modeling community. However, it is not the purpose of the paper to describe detailed methods/metrics of how the uncertainties could be quantified. But in section 5 (*Recommendations for improving current LULCC representation across models*), we outline what would conceptually be needed to (1) quantify the uncertainty range and (2) provide suggestions how it could be reduced.

We argue that little attention has been paid to the evaluation of land use models and the quantification of the uncertainty in their projections. This leads to a situation where often model and input data related differences dominate the scenario related uncertainties. We acknowledge that there are activities starting towards quantifying the uncertainties related to model assumptions (Riahi et al., 2016). However, two main issues are not resolved yet: (1) the 'marker' implementation of SSPS are intended to be used for climate change assessments without providing an error/uncertainty range due to different model interpretations, which would allow to quantify the uncertainty propagation into the final assessments and (2) all 'marker scenarios' are interpreted by a very similar kind of models (IAMs), which – especially when it comes to the spatial pattern of LULCC projections (= input to DGVMs/ESMs) – largely differ from alternative realizations of land-use allocation models (Prestele et al., 2016).

P 6L11: the problem is how to define 'plausible' realisations.

Response: It is indeed a difficult task. However, there are different products available for global land cover only (e.g., the ESA-CCI product, the MODIS product) or integrated with land-use statistics (e.g., Fritz et al., 2015; Klein Goldewijk et al., 2011; Ramankutty et al., 2008), which show substantial variations in the extent and especially the spatial pattern of land-use distribution. To elect one of these to the 'best' might be difficult due to technical and analytical constraints. Nonetheless, we would assume them to be 'plausible' since they are based on sound science (either remote sensing directly and/or remote sensing and statistics) and we added this deliberation to the manuscript in section 5 (page 13, lines 24-28).

*'To properly account for and disentangle the individual contribution of different historical reconstructions, the multitude of present-day land-use products and varying future land-use change modeling approaches, one would need a multi model ensemble design. Different future scenario models would need to be connected to different instances of historical reconstructions, both constrained by different plausible realizations (i.e., based on previously published, peer-reviewed approaches) of current land use and land cover.'*

P7L 2: Elaborate why sub-grid dynamics 'have been shown important'

Response: We added a few examples summarizing the results in the referenced publications (page 7, lines 15-24):

*'These sub-grid dynamics have been shown to be of importance when modeling change of carbon and nutrient stocks in response to land-use change in recent TBM studies (Bayer et al., in press; Fuchs et al., 2015b; Stocker et al., 2014; Wilkenskjeld et al., 2014). For example, Bayer et al. (in press) found the global cumulative land-use carbon emission to be ~33 % higher over the time period 1700-2014. Stocker et al. (2014) likewise report increased carbon emissions in recent decades and for all RCPs when accounting for shifting cultivation and wood harvest. Similarly, Wilkenskjeld et al. (2014) found a 60 % increase in the annual land-use emission for the historical period (1850-2005) and a range of 16-34 % increase for future scenarios, when accounting for gross changes. Recently, Arneth et al. (2017) demonstrated uniformly larger historical land-use change carbon emissions across a range of TBMs when shifting cultivation and wood harvest were included, which has implications for understanding of the terrestrial carbon budget as well as for estimates of future carbon mitigation potential in regrowing forest.'*

P7L6: what are 'under-determined mathematical systems' in this context?

Response: We removed this expression in the revised manuscript. Instead, we are now describing in section 3.3 (*Current approaches to provide gross change information: LUH and analysis of empirical data*) in more detail how the approach of Hurtt et al. (2011) works.

P7L8: what are 'minimum-transitions' in this context?

Response: Minimum transitions entail the changes between LULC categories only accounting for one-directional changes and derived from previously modeled net change time series. We added this explanation to the text (page 8, lines 17-18).

P7L21-23: Rephrase sentence

Response: We moved the statement to section 3.4 (*Open issues in the current approaches*) and rephrased to (page 9, lines 3-5):

*'[…]. However, our analysis of CLUMondo output (Figure 3), along with the European analysis of Fuchs et al. (2015a), suggests substantial amounts of gross changes (below the 0.5 degree LUH grid) also in the temperate zone and the high latitudes.'*

P10: rephrase 'allocation issue' do you mean the shifts between the communities and their perceived responsibilities

Response: 'Allocation issue' actually refers to the decision which land-cover type is replaced upon cropland and pasture expansion in the model. We moved the statement to section 4.5 (*Open issues of transition matrices*) and rephrased to (page 12, lines 14-16):

*'The provision of transition matrices, however, generally brings up a sequence of additional challenges, which we illustrate using the example of LUH in the following. First, the decision which land-cover type should be replaced upon cropland or pasture expansion (or introduced in case of abandonment) is in fact only shifted from the TBM community to the IAM/LUCM community […].'*

P10 Maybe you would like to add to your decision that 'satellite data' is also not 'directly measured data' but also goes through a mathematical conversion process.

Response: That is indeed a good point and we added a sentence that explicitly mentions that satellite data is not directly measured data, too (page 14, lines 16-17; see below). We think compared to the modeled LULCC data, satellite data entails a much more 'directly measured' component in most cases and can contribute to the evaluation of land-use change models.

*'Although satellite data is also not directly measured empirical data, but goes through a mathematical conversion process prior to a final land-cover product, it can improve representations of present-day land cover.'*

P11L24: Can you elaborate what 'improved communication' should entail in an ideal case. Additionally, I think it is not only the 'understanding' but first the 'awareness' of different assumptions and constrains needs to be achieved and before one can understand and tackle the problems.

Response: We believe improved communication eventually entails engaging in model integration as outlined in section 6 (*Outlook: towards model integration across disciplines*). Simultaneously the individual challenges discussed in the manuscript need to be resolved through joint engagement across the communities in the individual tasks (see recommendations now in section 5). Indeed, raising awareness of different assumptions and constraints is a first step and a major objective of the manuscript. By reorganizing the manuscript as described in our previous responses, we clarified this in the revised version, including recommendations which part of the community could most likely take the lead.

Figures: P24: add a 'log' label to the legend and add colour areas without change in grey to make the light yellow areas better stand out.

Response: We added the grey background. We did not include a log label, since the legend labels are given in km².

Supplement SP2L7: can you provide more details on the updated version, i.e. reference

Response: We added reference as far as possible (SI page 2, line 8).

The updated version is based on Ramankutty et al. (2008) for the static map of cropland distribution. There was however no additional publication related to the updated dataset. The dataset was available from http://www.geog.mcgill.ca/nramankutty/Datasets/Datasets.html, but the webpage has recently been removed.

SP5: Can you comment on the uncertainties associated with the CORINE data

Response: We added a short paragraph of uncertainties related to the CORINE data (SI page 9, lines 23 ff.). We do not think that it would add important information to the manuscript, if we elaborate comprehensively on the uncertainty in the CORINE data. We actually only use the change product for illustration in terms of cropland transition trajectories. These aggregated results are probably not heavily affected by uncertainty in the CORINE data.

SP6L8: Can you elaborate on what the 'thematic accuracy' entails.

Response: 'Thematic accuracy' entails the capability of CORINE land cover maps to represent the 'true' land-cover class as compared to an independent validation data set (EEA, 2006). We added an explanatory

sentence and the reference in the respective section of the supplementary material (SI page 9, lines 30-32).

**Reviewer #3 (Anonymous)**

The manuscript by Prestele et al., "Current challenges of implementing land-use and land-cover change in climate assessments", provides an overview of recent publications on interactions among land-use, carbon cycling, and different aspects of climate. First, the manuscript aims "…to identify existing shortcomings of the current LULCC representations within DGVMs and ESMs, reveal the underlying mechanisms and constraints that have hampered improved representations until now, and propose pathways to improve current representations" (page 3, lines 5-7). Second, based on the literature review, the manuscript attributes the lack of progress in including LULCC into climate assessments, to 1) the failure to account for uncertainty in reconstruction and future scenarios of gridded LULCC; 2) resolving sib-grid changes in land-use activities (e.g. gross transitions); 3) allocation of primary lands to managed lands in DGVMs and ESMs. Manuscript reviews a number of studies and discusses a wide range of limitations, specifically in CMIP5 historical reconstruction and future scenario. It has interesting discussion of how to use remote sensing data in improving treatment of LULCC processes in scenario development and its implementation into DGVMs and ESMs.

Response: The reviewer presents a good summary of the objectives of our manuscript. We thank the reviewer for the time spent on our manuscript and we appreciate the positive evaluation of our discussion about implementing remote sensing data in land – climate interaction studies.

For some of the following comments we sometimes split the original comments of the reviewer to address individual points.

However, the title is not appropriate because climate assessments such as IPCC do not implement LULCC – IPCC assessments review literature. CMIPs are not part of the IPCC, although their model simulations provide input to IPCC.

Response: The reviewer has a valid point here. Our current title is misleading, although we did not intend to equate the CMIP simulations with the IPCC assessment. We thus changed the title in addition to the changes suggested by reviewer #2 into:

*'Current challenges of implementing anthropogenic land-use and land-cover change in models contributing to climate change assessments'*

The manuscript has four major shortcomings: 1) While the manuscript reviews and synthesizes a number of recent studies on the development of scenarios of LULCC and of LULCC for climate and carbon cycling, it does not actually provide new insights or synthesis of LULCC implementation in ESMs and DGVMs. The manuscript provides a discussion of how the CMIP5 scenario was constructed and its limitations, but does not discuss differences in land use components of different ESMs or DGVMs. Or how they implemented the CMIP5 LULCC scenario. Table 1 gives 4 examples: 3 DGVMs (2 of which are variants of LPJ model) and a new HadGEM2-Jules ESM. There is no comprehensive analysis of CMIP5 ESMs or TRENDY DGVMs used in the AR5 in respect to LULCC. Thus, the manuscript's first goal is not supported by new insights beyond those previously published in literature.

Response: The reviewer raises a valid point that the wording of our main objectives leaves room for interpretation which requires clarification. It is not the purpose of the manuscript to focus on the technical details of land modules and/or LULCC implementation of ESMs/DGVMs. This has been done in related publications (e.g., De Noblet-Ducoudré et al., 2012) as the reviewer states correctly. Instead, we provide a synthesis of issues arising along the chain of activities regarding land-use in climate change assessments (remote sensing, modeling, scenario development, implementation in DGVMs/ESMs). For that purpose we review the literature and identify the three challenges that comprise our three main sections. For each challenge, we show why it is a challenge, what implications this challenge can have for carbon cycling and climate assessments, and discuss limitations in the current approach to overcome this particular challenge.

In this way, the manuscript provides important guidance to the communities involved, as we bring together the individual points for the first time, including recommendations for potential ways forward. In fact, Table 1 should be regarded as an illustration to support our argument rather than a comprehensive analysis. We clarified this by extending the wording in reference to Table 1 (page 10, line 3):

*'This has resulted in a range of different strategies, which we show as an illustration in Table 1 for a non-exhaustive list of models.'*

We feel accommodating changes as outlined in the response to the other two reviewers will better clarify the value of our manuscript.

Additionally, please note: We intentionally did not submit a 'research article' but a 'short communication' type of manuscript, since it is intended as a guidance or perspective for future research, rather than new fundamental research.

2) The manuscript claims that the limited characterization of uncertainty in CMIP5 and CMIP6 LU reconstructions and scenarios is responsible for the lack of progress on LULCC in climate assessments. There is no reason to believe that's true. CMIP is designed to compare climate models and ESMs under a common set of forcings and capture model structural uncertainty. CMIPs never claimed to capture all uncertainty due to input forcing. It's a well-established practice in climate MIPs to provide a standard scenario for all forcings – greenhouse gases, short-lived species, solar, constants, volcanoes and LULCC, particularly over historical periods. Such GCM or ESM simulations are extremely computationally expensive. Permutation of alternative forcings datasets is not likely something that many climate centers will be able to engage and afford. The idea of multiple LULCC reconstructions advocated by the paper for CMIPs is not practical. If some modeling group/center wants to explore uncertainty due to LULCC, there is more than one scenario that is available even from the GLM model: Hurtt et al. (2006) included both scenarios based on SAGE and HYDE datasets. Hurtt et al. 2011 examines different assumptions in the GLM model.

Response: We do not seek to claim that 'the limited characterization of uncertainty in LU reconstructions and scenarios is responsible for lack of progress on LULCC in climate assessments' in our manuscript. We argue that the current characterization of uncertainty is insufficient and errors unaccounted for propagate into climate assessments. For example, LUMIP (contributing to the goals of CMIP6) aims to answer the scientific question '*What are the global and regional effects of land-use and land-cover change on climate and biogeochemical cycling (past-future)?*' (Lawrence et al., 2016), which we think can only be done if there is a sufficient quantification of uncertainty in the land-use forcing data set in place as well. We now present the uncertainty in the land-use forcing as a source of uncertainty in the current coupling in section

5 (*Recommendations for improving the current coupling strategies*) and propose pathways how to reduce this uncertainty.

We agree with the point that CMIP is designed to compare climate models instead of forcing data sets, but it is similarly true that CMIP – due to its highly structured design – acts as a prototype for activities outside of CMIP and its forcing datasets (and as such the LUH) are widely used as a standard outside of CMIP, too. Please note, we do not restrict our arguments to the CMIP comparisons, but use it as an example at several places in the manuscript due to its high impact and pioneer role in the community.

We added at several locations in the revised manuscript that we do not restrict our arguments to CMIP and use LUH as an example (e.g., page 3, line 16; page 12, lines 14-15). In our view a 'well-established' practice is not necessarily the same as 'best practice'. The fact that climate modeling centers cannot (or in some cases do not prioritize to) explore uncertainty in LULCC does not necessarily imply that it should not be done at all. If not practical in such a comprehensive way as proposed in section 5 of the revised manuscript, then the communities need to come up with alternative strategies to tackle these uncertainties, e.g. determining a limited set of simulations that appears to significantly affect climate and carbon cycling using less computationally expensive DGVMs or offline land surface models (see our conclusion section). We now include this discussion explicitly in section 5 (page 14, lines 1-6):

*'The high computational demands of complex ESMs probably do not allow for multiple runs including all the uncertainties in land-use forcing. However, to derive robust results from climate model intercomparisons, a sufficient quantification of uncertainty in the land-use forcing dataset is urgently required. If this proves impractical through ESM simulations, we recommend to utilize less computational expensive models such as DGVMs and offline LSMs to assess the full range of uncertainty and determine a limited set of simulations, which appears to significantly affect biogeochemical cycles and climate. These can be subsequently used to test the uncertainty range in ESMs.'*

Harmonization to a common input for climate models is a major first step to have LULCC included in the climate simulations; in a next step the communities need to find a way to systematically approach the related uncertainties. Here our recommendations and conclusions could provide guidance on how to move forward and we rephrased them to be more specific (see sections 5 and 6).

The main bottleneck for improving LULCC characterization in the CMIP is poor representation of LULCC processes in GCMs and ESMs. Most CMIP5 ESMs or TRENDY DGVMs can't use the information available in CMIP5 or CMIP6 historical reconstructions or future scenarios. For example, most of the CMIP5 models use only information about land use fractions, and not gross transitions provided by the Hurtt et al. (2011) data set. With the exception of very few models, ESMs do not represent shifting cultivation or wood harvesting.

Response: We agree that one of the major issues in land use – climate interaction studies is the 'poor representation' of LULCC in GCMs and ESMs. In fact, we identify it as one of the main issues in our manuscript as well (section 4) and emphasized it additionally following the suggestions of reviewer #1 and #2. Given the low certainty in inputs of additional products such as wood harvest and shifting cultivation (Erb et al., 2016; Hurtt et al., 2011), inclusion of the processes in ESMs is not necessarily the only bottleneck. Simultaneously, we do not agree that a 'main bottleneck' (e.g., the poor representation) justifies neglecting other important issues (such as the uncertainty in LU modeling and the gross transitions) we raise in the manuscript. Instead, the LU modeling community should clearly communicate these issues as well and take the lead on improving the products (see section 5).

Another unsupported assumption in the manuscript is that, by making additional ESMs or GCMs with alternative representations of LULCC history, one would get a better handle on the uncertainty in climate feedback of LULCC. It's not necessarily true: most studies with and without LULCC typically find a small difference in global climate and small regions with statistically distinguishable differences in climate characteristics. One would need a large ensemble of such simulations to find differences between the biogeophysical effects of alternative LULCC reconstructions and scenarios, unless they are really different as in future scenarios. Biogeophysical differences should be more pronounced, but the problem is that CMIP5 or even CMIP6 ESMs are incapable of representing major LU processes such as shifting cultivation, wood or crop harvesting..

Response: Regarding biogeochemical effects it has been shown that alternative reconstructions make a large difference to carbon emissions (e.g., Bayer et al., in press; Meiyappan et al., 2015). From these findings we derive that alternative reconstructions are 'likely to substantially impact' LULCC – climate interactions and propose to determine a limited set of contrasting simulations (using DGVMs, offline LSMs) that could be tested in ESMs (see section 5). For example, the work of Kaplan et al. (2011) and Fuchs et al. (2013) has shown that historical reconstructions can substantially differ from HYDE, thus we think it is an important scientific question how alternative reconstructions could affect the climate signal. Additionally, the uptake of a high/low estimate of historical land use in LUMIP (Lawrence et al., 2016) after feedback from the community (see the open discussion of the LUMIP paper, Lawrence et al. (2016), in *Geoscientific Model Development*), indicates that the uncertainty in LU products is indeed an important issue that needs to be further explored.

3) The manuscript questions assumptions in CMIP5 Hurtt et al. 2011 reconstruction and future scenario. The Hurtt et al. (2011) effort, for the first time, harmonized historical reconstruction with the 4 Representative Concentration pathways (RCP) scenario and took into account gross transitions between different LU types in both tropics and extra-tropics. The authors are mistaken in their assumption that no-shifting cultivation in the extra-tropics implies no gross-transitions in the extra-tropics; for example, non-zero transitions between pastures to crops and crops to pastures. Furthermore, for CMIP6 (Lawrence et al. 2016), there will be a focus and additional LUH reconstructions available, as well as more details about the relationship between land cover and land use categories. I think a lot of criticism of the CMIP5 LULCC reconstruction and scenario is valid but the authors are overlooking improvements in the new reconstruction for CMIP6, which is publicly available now on the CMIP6 website.

Response: We acknowledge the effort of the Hurtt et al. (2011) harmonization activity and its contribution to enhance LULCC representation in land use – climate interaction studies. But, as the reviewer states later on, there are also limitations in the CMIP5 product. We do not assume that gross transitions can only appear in the tropics due to shifting cultivation (page 8, lines 16 ff.), but argue that due to the resolution of the minimum transitions, gross transitions, especially in the temperate zone and the high latitudes, might be missed (page 9, lines 3 ff.).

In terms of CMIP6 and the LUH2 product, we do not overlook the improvements, but explicitly mention the update and now included a specific mentioning why we refer to the CMIP5 product throughout the manuscript (page 4, lines 15-20):

'It has recently been updated for the upcoming 6th Phase of the Coupled Model Intercomparison Project (CMIP6; Eyring et al., 2016; Lawrence et al., 2016) and data for the historical period have been published (hereafter referred to as LUH2). Due to the lack of comprehensive documentation of the updated version at the time this paper was written and as, to our best knowledge, the points we demonstrate using LUH

*will be still valid with the new product, we primarily refer to the CMIP5 version in the remainder of this paper.'*

We admit that it is extremely difficult at the moment to follow the improvements compared to the LUH CMIP5 product, since documentation of the new products is restricted to a rather generic description (Lawrence et al., 2016) and thus we might miss some details. We are aware that the new historical product is publicly available, but the final dataset does not allow to trace back how the individual processes were implemented. To our best knowledge – apart from the indisputable improvements between the two products (e.g., additional focus on land management, improved shifting cultivation estimate) – some of our main criticisms (e.g., gross transitions in the extra-tropics, derivation of LU transitions) will be untouched even with the new product.

While it's possible to construct more detailed scenarios for recent periods with satellite coverage or for specific countries (e.g., Table 2 in the manuscript), particularly in the Northern Hemisphere, it is difficult if not impossible to develop multi-century reconstructions on a global scale with consistent sets of assumptions. Making simple assumptions in ESM is not an unreasonable approach for global, multi-century analyses. Assuming transitions based on the satellite era for the entire CMIP-style experiments may be problematic, as well, for pre-industrial or future periods.

Response: We agree that making simple assumptions is a reasonable approach to get started with the land-use implementation in DGVMs/ESMs and that the satellite era might not be representative for multi-century analysis as well. We included this thought in section 3.4 (*Open issues in the current approaches*; page 9, lines 8-13):

*'The data-based approach avoids the process uncertainty which hinders high-resolution model projections of land use, but is limited to the time period where empirical data through remote sensing is available. Additional sources such as historical land-use and land-cover maps and statistics (Fuchs et al., 2015c) may contribute to cover larger time periods, although with limited spatio-temporal resolution and spatial coverage, and an associated increase in uncertainty. It is thus difficult to develop multi-century reconstructions or future scenarios including gross changes using data-based approaches, since the derived gross/net ratios are only valid for periods of data coverage and are expected to change over time (Fuchs et al., 2015a).'*

However, at least for this era the model assumptions should be carefully evaluated. Based on sufficient transition information for present-day, these assumptions could be, e.g., gradually replaced over time (backward and forward) based on scenario assumptions or regional characteristics (e.g., Fuchs et al., 2015). Sensitivity analysis could provide additional insights how individual decisions affect the land-use pattern, even over long time periods. In such a way the models would account for spatio-temporal variability in land-use transitions.

4) The rationale for including analysis from the CLUMondo model is not clear – it demonstrates how spatio-temporal variations could be different within the grid. It does not show that such patterns will affect climate or carbon cycling. Besides the CLU-Mondo analysis, there is now new analysis in this manuscript. So, there are no new insights/analysis, just a synthesis of other studies, which are already partially covered by the authors in related publications (e.g., Alexander et al. 2016, Bayer et al, 2016, Prestele et al. 2016).

Response: We added a sentence to our objectives, which explicitly mentions the illustrative purpose of the CLUMondo analysis (page 3, lines 11-13).

*'We review recent literature from the land use, land cover, carbon cycle and climate modeling communities and support our arguments by illustrative analysis of satellite land-cover products and outputs of the land-use change model CLUMondo (Van Asselen and Verburg, 2013).'*

Specifically, the rationale for including CLUMondo analysis is to show that a simple allocation algorithm (such as forest will be cleared upon cropland expansion) applied globally might not sufficiently account for the spatio-temporal heterogeneity in the change patterns. Previous publications have shown that these decisions can affect regional climate and carbon cycling (e.g., de Noblet-Ducoudre, 2012), and thus our analysis should be taken as an illustration that further research is required on how these decisions affect the ESM results. We emphasized the illustrative purpose of the CLUMondo analysis by moving it to a separated sub-section (4.3, *Example: Spatial heterogeneity of cropland transitions in the CLUMondo model*).

As mentioned in previous responses, the manuscript does not aim to present comprehensive new analysis, but rather use illustrative analysis using the CLUMondo model to support our arguments. In doing so, we provide a synthesis of currently untackled, or insufficiently tackled, challenges at the interface of land-use and climate modeling, and try to present guidance for the communities involved.

I think the most interesting part of the paper is the section on remotely sensed data (high and low resolution) in development of new diagnostics for evaluation of global LULCC reconstructions or models. Perhaps the authors can re-frame their analysis and demonstrate how such data can be used to improve or evaluate reconstructions (e.g. the one in CMIP6) or to create new diagnostics to evaluate ESMs and DGVMS.

Response: In our view, our manuscript brings together three major challenges/issues at the interface of land-use and climate modeling, which can serve as a guidance on the 'ways forward' to the communities involved – we therefore do not share the opinion that the remotely-sensed data should be the chief focus of the paper. Certainly there is also a need to develop new diagnostics as mentioned by the reviewer, but this is beyond the scope of this current paper. However, the reviewer raises a fair point and we added this need to section 5 (*Recommendations for improving the current LULCC representation across models*) of the revised manuscript.

*'Simultaneously, the land-use and remote-sensing communities should engage to reduce uncertainties in land-use and land-cover products by:*

(1) *Developing diagnostics for the evaluation of land-use reconstructions based on satellite data and additional proxy data such as pollen reconstructions (Gaillard et al., 2010) or archeological evidence of early land use (Kaplan et al., 2016)*

(2) *[…]'*

[revised manuscript text omitted]

**Supplementary Information**

**S1 Overview of historical land use reconstructions**

Several approaches have been published within the last two decades to reconstruct the history of human utilization of land to meet their needs of food, fiber and space for settlement on a global scale. Depending on the objective of the particular study they cover different time periods, spatial resolutions and methods of reconstruction (Table S1). In the following paragraphs we summarize the methodologies of four spatially explicit historical reconstructions. For details, please see the original publications.

**HYDE**

The History Database of the Global Environment (HYDE) was originally developed by Klein Goldewijk (2001), covering spatially explicit historical population estimates and land-use patterns for the past 300 years at 0.5° resolution. Several updates and extensions led to version HYDE 3.1, which was used for the Land Use Harmonization (LUH ) in CMIP5 (Klein Goldewijk et al. (2011); this is the version we refer to here and in the article). Recently there has been a update to version HYDE 3.2, which now covers a time period from 10 000 BC to 2015 AD at 5 arcminute spatial resolution and includes further agricultural management layers (such as irrigation) (Klein Goldewijk, 2016).

The underlying principle of the HYDE reconstruction is the relationship between human population and agricultural activity expressed in a per capita use of cropland and pasture area, leading to a spatial dependency of land-use activities to human settlements. Klein Goldewijk et al. (2010) first derived time series of population numbers from a vast number of sources on a subnational or national scale (depending on data availability, e.g.. McEvedy and Jones (1978), Livi Bacci (2007) and Maddison (2001); see Klein Goldewijk (2001) and Klein Goldewijk et al. (2011) for details) and translated them to population density maps using patterns from Landscan (2006) for recent time and a combination of suitability maps for historic time. For the period 1961-2000, the per capita use of cropland and pasture was calculated from FAO statistics on country or subnational level. Prior to 1961 the per capita land- use numbers were dynamically estimated country by country following Ruddiman and Ellis (2009) and adjusted accounting for low population numbers (= higher per capita land use), but also limitations in technology and a  maximum area of land that can be cultivated by a subsistence farmer (= lower per capita land use). Using the per capita usage of cropland and pasture to estimate cropland and pasture total areas on a (sub-)national level for every time step, spatial allocation of the total areas to the 5 arcminute  grid was implemented using two sets of weighing maps: On the one side, present distribution of cropland and pasture was derived by integrating FAO statistics and additional subnational statistics for the USA and China with two satellite derived land cover products representative for recent time (DISCover version 2, Loveland et al. (2000); GLC2000, Bartholome and Belward (2005)). The weighing map for historical time, on the other

side, was constructed by combining the earlier described population density maps and different biophysical suitability parameters, namely soil quality, distance to rivers, steepness of terrain, and thresholds for annual mean temperature. Both maps were subsequently used to allocate (sub-) national totals of agricultural areas to specific grid cells, while the influence of the historic map gradually increases when going further into the past.

5   S1.2

**Ramankutty and Foley (1999)**

Ramankutty and Foley (1999) apply a hindcast modeling technique to derive global scale spatial patterns of cropland on a global scale for the period 1700-1992. The original reconstruction did not includeinlcude pasture areas. A revised and updated version[1] covers the years up to 2007, both for cropland and pasture at 5 arcminute spatial resolution. The starting point for the

10  reconstruction is represented by the integration of satellite derived land cover products (DISCover in original data set (Loveland and Belward, 1997); BU-MODIS (Friedl et al., 2002) and GLC2000 (Bartholome and Belward, 2005) in the updated version) and FAO statistics. The national and subnational totals of cropland and pasture were calibrated to the spatial distribution of cropland and pasture areas in the earth observation product applying a linear fitting approach. This resulted in a global, 5 arcminute resolution cropland and pasture map for the year 2000, representing the spatial distribution of cropland

15  and pasture areas (Ramankutty et al., 2008). In a second step, a comprehensive data base of historical agricultural areas on (sub-) national level was compiled from different sources. FAO statistics were used for the time period from 1961 to the end point. Prior to 1961, the data base first accounts for census data. Whenever census data were not available, cropland conversion rates of Houghton and Hackler (1995) were applied to the cropland map of Richards (1990) for 1980 with some regional adjustments to avoid unrealistic agricultural areas in particular regions. The spatial allocation of the cropland areas is

20  implemented by applying a simple hindcast model, which preserves the cropland pattern of the start map within each unit of the inventory data base for the whole time period to 1700. For that a change factor between two subsequent years is calculated from the inventory database, dividing the cropland area in the target year by the cropland area in the starting year, which is thereafter applied to each grid cell within a unit.

**S1.3 *Pongratz et al. (2008)**

25  Pongratz et al. (2008) extended the reconstruction of Ramankutty and Foley (1999) back to 800 AD and presented the first consistent and spatially explicit cropland and pasture reconstruction for pre-industrial times at the date of publication. For the period 1700-1992, the cropland time series is, apart from smaller regional adjustments and updates, the same than the Ramankutty and Foley (1999) data. Since they further had not published their pasture time series at that point, Pongratz et al. (2008) combined the pasture map for 1992 with change rates taken from the HYDE data base to extend it back to 1700. Unlike

30  the pattern maintaining approach applied by Ramankutty and Foley (1999), pasture was spatially distributed around existing
* * *
[1] The updated version is based on the global cropland and pasture maps published in Ramankutty et al. (2008) and the methodology described in Ramankutty et al. (1999). There was, however, no additional publication related to the updated dataset. The dataset was available from http://www.geog.mcgill.ca/nramankutty/Datasets/Datasets.html, but the webpage has been recently removed.

cropland while maintaining the pattern of total agricultural area rather than the individual shares of cropland and pasture to allow also for cropland expansion into pasture areas.

Based on this two time series covering the years 1700-1992, an extrapolation to 800 AD was applied on (sub-)-national level, while using population data from McEvedy and Jones (1978) as a proxy for land-use change. Similar to HYDE, the simple measure of per capita usage of crop and pasture area was assumed to be the best approximation. However, in this case, per capita use was calculated from the 1700 maps and held constant for the whole period prior to 1700. Spatial distribution of agricultural areas was assumed to represent the patterns of 1700 for the period 800 to 1700. Besides, changes in agricultural patterns, e.g. following the European colonization in North and South America, were especially accounted for by altering the patterns in particular regions. Both time series were aggregated to a 0.5° resolution.

**KK10**

Kaplan et al. (2010) introduce a non-linear relationship between population numbers and area of forest clearance to calculate total areas affected by human land-use change. The basic assumption of this approach is a decreasing per capita land use over time due to intensification of already converted areas rather than expand land use to new areas when population densities increase. With the objective to build an empirical, non-linear model, population time series for the period 6050 BC to AD 1850 were compiled first. Data from McEvedy and Jones (1978) were utilized for the period 1000 BC to AD 1850 with some regional adjustments and subsequently extended back to 6050 BC by a modelling approach (Global Land USE and Technological Evolution Simulator (GLUES, see Lemmen (2009) and Wirtz and Lemmen (2003) for details). Population density was normalized to cultivatable land to prevent the model extending cropland areas into unsuitable land. A sigmoidal log-linear model was fitted to a set of empirical data from various European countries to derive a relationship between forest cover and population density accounting also for different stages of technological development over time (Kaplan et al., 2009). Concurrently, Kaplan et al. (2010) integrated different climatic and biophysical variables to indices of suitability for cropland and pasture on a 5 arcminute grid following a method of Ramankutty et al. (2002). Combining the regional level estimates of historical forest cover with the suitability datasets led to a spatially explicit representation of area affected by land-use change over time. The integration was done by allocating cropland to high quality and suitable areas first, followed by pasture. As the forest cover – population relationship originally was derived for Europe, it has been adjusted for tropical and boreal regions in the global approach by including a threshold of net primary production, where productivity of agricultural lands is higher and therefore demand for new land lower.

**Table S1: Summary of historical LULCC reconstructions.**

| Reference | Spatial resolution | Temporal coverage and resolution | Input data | Allocation |
|---|---|---|---|---|
| KK10, Kaplan et al. (2010) | 5 x 5 arcminute | 6050 BC to AD 1850, annual | population estimates, land suitability maps | based on non-linear population density – forest clearance relationship, high quality land cleared first |
| HYDE 3.1, Klein Goldewijk et al. (2011) | 5 x 5 arcminute | 10 000 BC to AD 2005, variable resolution | population estimates, FAO statistics, satellite derived products | dynamic per capita use of cropland and pasture; combination of weighing maps derived from satellite products, population and environmental parameters |
| Pongratz et al. (2008) | 0.5 x 0.5 degree | AD 800 – AD 1992 | adjusted Ramankutty and Foley (1999), HYDE 2.0, population data | constant per capita use of cropland & pasture prior to 1700, constant spatial pattern of agriculture prior to 1700 |
| Ramankutty and Foley (1999) | 5 x 5 arcminute | AD 1700 – AD 1992; update AD 1700 – 2007 | census data and estimates of agricultural area, FAO statistics, satellite derived products | hindcast model, preserving agricultural pattern of 1992 within aggregated units |

**S2 Data and Methods**

Several data and methods have been used to support our arguments in the manuscript and create the related tables and figures. To ensure readability we decided to provide methodological details in the Supplementary Information rather than in the main text of the manuscript. In the following we provide an overview of the data used, details of the data processing and how

5 analysis was conducted. In each section heading we indicate the relation to the main text and the figures and tables that were derived from individual steps of analysis.

**S2.1 Attribution of uncertainty in land use change projections (Section 2; Figure 2)**

Multiple linear regression analysis followed by an ANOVA was used to decompose the variability of 43 projections of regional pasture areas for the year 2030 simulated by 11 global scale IAMs and LUCMs (Alexander et al., 2016; Prestele et al., 2016).

10 Every individual projection has been parameterized according to 9 variables (Table S2) that characterize the model structure (model type classification, model resolution), the scenario (socioeconomic and climate scenario variables) and the initial condition (deviation of absolute pasture area from value reported by FAOSTAT (2015) in the year 2010) prior to the regression analysis. The modeled pasture area in 2030 was assumed to be a function of these 9 variables. To balance performance and complexity of the resulting regression model, variables were rejected using the Akaike information criterion.

15 Subsequently an ANOVA was conducted on the regression results to identify relative contribution of the variables to the total variation in the regression model of the 2030 pasture areas. The type II[2] sum of squares were calculated for each variable and divided by the total sum of squares. Subsequently, the relative contributions of the individual variables were summarized according to the grouping in Table S2. The residual term thus covers all variation that could not be explained by these 9 variables.
* * *
[2] Type II sum of squares have been used since they are not dependent on the order in which the variables are considered in the model, which has been shown suitable for unbalanced data as in our analysis (Langsrud, 2003). See Alexander et al. (2016) for details.

**Table S2: Overview of variables used in the regression analysis and ANOVA (table adopted and modified according to Prestele et al. (2016)).)**

| Variable | Data type | Group |
|---|---|---|
| Initial condition delta | Continuous (deviation of model areas from FAO areas in 2010 (FAOSTAT, 2015) | Initial |
| Model type | Categorical (CGE, PE, Rule-based, Hybrid) | Model |
| Number of model cells (log) | Continuous | Model |
| $CO_2$ concentration 2100 | Continuous | Scenario |
| Population 2100 | Continuous | Scenario |
| GDP growth rate to 2100 | Continuous | Scenario |
| Inequality ratio 2100 | Continuous | Scenario |
| Technology change | Discrete (0=None, 1=Slow, 2=Medium, 3=Rapid) | Scenario |
| International trade | Discrete (1=Constrained, 2=Moderate, 3=High) | Scenario |

**S2.2 Derivation of gross vs. net changes due to re-gridding from a CLUMondo simulation (Section 3; Figure 3)**

To identify the difference between net and gross changes due to re-gridding of high-resolution modeled land-use change information, we utilized data from a simulation of the CLUMondo model (Van Asselen and Verburg, 2013) based on the FAO 3 demand scenario (Eitelberg et al., 2016). These data are available at a 9.25 x 9.25 km regular grid (~5 arcminute) in an equal area projection and are based on the land system classification described in van Asselen and Verburg (2012). Land systems are characterized by land-cover composition, livestock numbers and land-use intensity. Each grid cell can thus be expressed as a mosaic of five LULC types (cropland, grassland, forest, urban, and bare) which varies with the world region. Upon a change from one land system to another, these characteristics also change.

We used the fractions of these five LULC types to track areal changes per grid cell at the original 9.25 x 9.25 km resolution over the whole simulation period (2000-2040). The total area changed at this resolution (sum of gains and losses for each LULC type) was assumed to be the gross changes in our analysis. In a second step, we aggregated the maps to ca. 0.5 x 0.5 degree and calculated the changes between two time steps. Due to bi-directional changes at the higher resolution (which offset each other) the total area affected by change at 0.5 x 0.5 degree resolution is usually smaller. The areal changes at 0.5 x 0.5 degree resolution were assumed to be the net changes in our analysis. By adding up the net changes and gross changes across all five LULC types and over the whole simulation period, we identified the amount of actually changed area that would be missed in a net change representation at 0.5 x 0.5 degree for this simulation ().

[Figure]

**Figure S1: Preprocessing workflow of CLUMondo output for gross change analysis. Rectangles represent processing steps, parallelograms represent data. Grey shaded items emphasize aggregated data at ca. 0.5 x 0.5 degree resolution.**

To derive dominant sources of cropland expansion from remote sensing products, we analyzed high resolution LULCC data from Europe (CORINE, 100 m spatial resolution) and North America (NLCD, 30 m spatial resolution) (Table S3). We downloaded CORINE data from http://land.copernicus.eu/pan-european/corine-land-cover. NLCD data were obtained through http://www.mrlc.gov/.

**S2.3.1 Data: CORINE**

CORINE was produced by computer assisted visual interpretation of satellite images processed on a country-bycountry basis and subsequently merged to a comprehensive European database (EEA, 2007). It covers the years 1990, 2000, 2006 and most recently 2012 with different number of participating countries leading to different overlapping areas between the years. The land-cover classification was derived from different sensors dependent on the final year of the product (1990: Landsat-4/5 TM single date, 2000: Landsat-7 ETM single date; 2006: SPOT-4 and/or IRS P6 LISS III dual date; 2012: IRS P6 LISS III and RapidEye dual date). CORINE is provided at a spatial resolution of 100 m and 250 m in raster data format as well as in vector format. The minimum mapping unit is 25 ha. Besides the products for the years mentioned above, special LULCC products have been produced and are currently available for the periods 1990 to 2000 and 2000 to 2006. For the change products an enhanced minimum mapping unit of 5 ha was applied. The change products have been used for derivation of agricultural transitions in our analysis, thus covering all changes to agricultural areas larger than 5 ha between start and end year. All CORINE products are accompanied by a three level land-use and land-cover nomenclature varying in detail across the levels (Table S4). The first level only provides very general classes (e.g. artificial surfaces; agricultural areas; forests; etc.). The second level distinguishes 15 different categories and the highest detail is given by the 44 classes at level 3. For our analysis we used a merger of the different levels, as e.g. forests and shrubland could be only differentiated at level 2, while natural grassland could be only identified at level 3 (Table S4). See Bossard et al. (2000) for a detailed description of the legend and distinction of individual classes. Although CORINE provides a consistent framework of European land cover mapping, uncertainties in the final products are necessarily apparent. For example, the country-by-country processing of data can introduce uncertainty due to different treatment of the individual legend items during visual interpretation of the satellite imagery. However, clearly defined mapping guidelines aim to minimize these effects (Bossard et al., 2000). Moreover, the minimum mapping unit of 5 ha (in case of the change product that was used in our analysis) ignores changes on smaller areas. Thus, additional uncertainty can be introduced in areas where less changes appear. The thematic accuracy of the 2000 to 2006 change product is indicated with larger than 85%, while the accuracy for the 1990 to 2000 change product has not been assessed (see http://land.copernicus.eu/pan-european/corine-land-cover). Thematic accuracy entails the capability of CORINE land cover maps to represent the 'true' land-cover class as compared to an

independent validation dataset (EEA, 2006). Although these uncertainties may propagate into our analysis of cropland transition trajectories (Table 2, Figure 4), we do not expect them to substantially change the order of source LULCC categories at the aggregated European scale.

**S2.3.2.2 Data: NLCD**

5 The National Land Cover Database (NLCD) is a high resolution (30 m) land-cover product for the USA. This Landsat- derived product has been provided for the years 1992, 2001, 2006 and 2011 at the latest. For our analysis the 2001, 2006 and 2011 products have been considered, as they are provided in a harmonized collection with special change products. The NLCD dataset is classified according to a 16-class land-cover classification for the United States, developed in the 1970s by Anderson et al. (1976). The classification system distinguishes two agricultural classes, *(81) Pasture/Hay* and *(82) Cultivated*

10 *Crops* (Table S5). Stehman et al. (2003) report an accuracy level of 55.7 % for the 1992 dataset. Accuracy assessment is not yet available for the 2011 data, but as 2001 and 2006 data showed significantly improved accuracy levels (78.7 % and 78.0 %, Wickham et al. (2010) and Wickham et al. (2013)) a similar (or even better) quality can be assumed for the 2011 data.

**Table S3: Summary of land-cover products used for our analysis.**

| Product | Temporal coverage | Spatial resolution / Coverage | Legend | Sensor | Classification |
|---------|-------------------|-------------------------------|--------|--------|----------------|
| CORINE | 1990, 2000, 2006, (2012) | 100m / Europe | 44 classes, 3 hierarchical levels | Landsat-4/5 TM, Landsat-7 ETM, SPOT-4, IRS P6 LISS III, RapidEye | change product, supervised, expert knowledge |
| NLCD | (1992), 2001, 2006, 2011 | 30m / USA | 16 classes | Landsat | change product, spectral and knowledge based change detection |

15 **S2.2.3.3 Change detection**

We used the dedicated change products for our analysis, which hold information about source and target classes upon land-use change. Areas of agricultural expansion were identified by every pixel that has an agricultural label (based on the inherent legend) at time t2, but not at time t1. We calculated the total expansion of agricultural areas by the difference of pixels which were assigned an agricultural label at time t2 and time t1. Subsequently, combining the areas of cropland expansion with the

20 map of time t₁t 1 resulted in a map of sources of agricultural area. The source maps were classified and summarized considering the underlying original legend into grassland, forest, mixed grassland/forest and unvegetated land origin (Table S4, Table S5).

**Table S4: CORINE land-cover legend (Bossard et al., 2000) and aggregation applied in our analysis.**

| Level 1 | Level 2 | Level 3 | Aggregation |
|---------|---------|---------|-------------|

| (1) Artificial surfaces | (11) Urban fabric; (12) Industrial, commercial and transport units; (13) Mine, dump and construction sites; (14) Artificial, non-agricultural vegetated areas | (111) Continuous urban fabric; (112) Discountinuous urban fabric; (121) Industrial and commercial units; (122) Road and rail networks and associated land; (123) Port areas; (124) Airports; (131) Mineral extraction sites; (132) Dump sites; (133) Construction sites; (141) Green urban areas; (142) Sport and leisure facilities | Other |
|---|---|---|---|
| (2) Agricultural areas | (21) Arable land; (22) Permanent crops; (23) Pastures; (24) Hegerogeneous agricultural areass | (211) Non-irrigated arable land; (212) Permanently irrigated land; (213) Rice fields; (221) Vineyards; (222) Fruit trees and berry plantations; (223) Olive groves; (231) Pastures; (241) Annual cropas associated with permanent crops; (242) Complex cultivation patterns; (243) Land principally occupied by agriculture, with significant areas of natural vegetation; (244) Agro-forestry areas | Agricultural areas |
| (3) Forest and semi natural areas | (31) Forests; (32) Scrub and/or herbaceous vegetation associations; (33) Open spaces with little or no vegetation | (311) Broad-leaved forest; (312) Coniferous forest; (313) Mixed forest; (321) Natural grasslands; (322) Moors and heathland; (323) Sclerophyllous vegetation; (324) Transitional woodland-shrub; (331) Beaches, dunes, sands; (332) Bare rocks; (333) Sparsely vegetate areas; (334) Burnt areas; (335) Glaciers and perpetual snow | (311)-(313) Forest (321) Grassland (322)-(324) Shrubland (331)-(335) Other |
| (4) Wetlands | (41) Inland wetlands; (42) Maritime wetlands | (411) Inland marshes; (412) Peat bogs; (421) Salt marshes; (422) Salines; (423) Intertidal flats | Other |
| (5) Water bodies | (51) Inland waters; (52) Marine waters | (511) Water courses; (512) Water bodies; (521) Coastal lagoons; (522) Estuaries; (523) Sea and ocean | Other |

**Table S5: National Land Cover Database (NLCD) classification system according to Anderson et al. (1976) and aggregation applied in our analysis.**

| Value | Label | Description | Aggregation |
|---|---|---|---|
| 11 | Open Water | All areas of open water, generally with less than 25 % cover or vegetation or soil | Other |
| 12 | Perennial Ice/Snow | All areas characterized by a perennial cover of ice and/or snow, generally greater than 25 % of total cover | Other |
| 21 | Developed, Open Space | Includes areas with a mixture of some constructed materials, but mostly vegetation in the form of lawn grasses. Impervious surfaces account for less than 20 % of total cover. These areas most commonly include large-lot single-family housing units, parks, golf courses, and vegetation planted in developed settings for recreation, erosion control, or aesthetic purposes | Other |
| 22 | Developed, Low Intensity | Includes areas with a mixture of constructed materials and vegetation. Impervious surfaces account for 20-49 % of total cover. These areas most commonly include single-family housing units. | Other |
| 23 | Developed, Medium Intensity | Includes areas with a mixture of constructed materials and vegetation. Impervious surfaces account for 50-79 % of the total cover. These areas most commonly include single-family housing units. | Other |
| 24 | Developed, High Intensity | Includes highly developed areas where people reside or work in high numbers. Examples include apartment complexes, row houses and commercial/industrial. Impervious surfaces account for 80-100 % of the total cover. | Other |
| 31 | Barren Land (Rock/Sand/Clay) | Barren areas of bedrock, desert pavement, scarps, talus, slides, volcanic material, glacial debris, sand dunes, strip mines, gravel pits and other accumulations of earthen material. Generally, vegetation accounts for less than 15 % of total cover. | Other |
| 41 | Deciduous Forest | Areas dominated by trees generally greater than 5 meters tall, and greater than 20 % of total vegetation cover. More than 75 % of the tree species shed foliage simultaneously in response to seasonal change. | Forest |
| 42 | Evergreen Forest | Areas dominated by trees generally greater than 5 meters tall, and greater than 20 % of total vegetation cover. More than 75 % of the tree species maintain their leaves all year. Canopy is never without green foliage. | Forest |
| 43 | Mixed Forest | Areas dominated by trees generally greater than 5 meters tall, and greater than 20 % of total vegetation cover. Neither deciduous nor evergreen species are greater than 75 % of total tree cover. | Forest |
| 52 | Shrub/Scrub | Areas dominated by shrubs; less than 5 meters tall with shrub canopy typically greater than 20 % of total vegetation. This class includes true shrubs, young trees in an early successional stage or trees stunted from environmental conditions. | Shrubland |
| 71 | Grassland/Herbaceous | Areas dominated by grammanoid or herbaceous vegetation, generally greater than 80 % of total vegetation. These areas are not subject to intensive management such as tilling, but can be utilized for grazing. | Grassland |
| 81 | Pasture/Hay | Areas of grasses, legumes, or grass-legume mixtures planted for livestock grazing or the production of seed or hay crops, typically on a perennial cycle. Pasture/hay vegetation accounts for greater than 20 % of total vegetation. | Pasture |
| 82 | Cultivated Crops | Areas used for the production of annual crops, such as corn, soybeans, vegetables, tobacco, and cotton, and also perennial woody crops such as | Cropland |

| 90 | Woody Wetlands | Areas where forest or shrub land vegetation accounts for greater than 20 % of vegetative cover and the soil or substrate is periodically saturated with or covered with water. | Other |
| 95 | Emergent Herbaceous Wetlands | Areas where perennial herbaceous vegetation accounts for greater than 80 % of vegetative cover and the soil or substrate is periodically saturated with or covered with water. | Other |

orchards and vineyards. Crop vegetation accounts for greater than 20 % of total vegetation. This class also includes all land being actively tilled.

**S2.3 Derivation of gross vs. net changes due to re-gridding from a CLUMondo simulation**

To identify the difference between net and gross changes due to re-gridding of high resolution modeled land use change information, we utilized data from a simulation of the CLUMondo model (Van Asselen and Verburg, 2013) based on the FAO

5  3 demand scenario (Eitelberg et al., 2016). These data are available at a 9.25 x 9.25 km regular grid (~ 5 arcminute) in an equal area projection and are based on the land system classification described in van Asselen and Verburg (2012). Land systems are characterized by land-cover composition, livestock numbers and land-use intensity. Each grid cell can thus be expressed as a mosaic of five LULC types (cropland, grassland, forest, urban, and bare), whose exact fractions vary with the world region. Upon a change from one land system to another, these characteristics also change.

10 We used the fractions of these five LULC types to track areal changes per grid cell at the original 9.25 x 9.25 km resolution over the whole simulation period (2000-2040). The total area changed at this resolution (sum of gains and losses for each LULC type) was assumed to be the gross changes in our analysis. In a second step, we aggregated the maps to ca. 0.5 x 0.5 degree and calculated the changes between two time steps. Due to bi-directional changes at the higher resolution (which offset each other) the total area affected by change at 0.5 x 0.5 degree resolution is usually smaller. The areal changes at 0.5 x 0.5

15 degree resolution were assumed to be the net changes in our analysis. By adding up the net changes and gross changes across all five LULC types and over the whole simulation period, we identified the amount of actually changed area that would be missed in a net change representation at 0.5 x 0.5 degree for this simulation (Figure S1).

**S2.4 CLUMondo land-use change priority analysis (Section 4; Figure 5)**

The CLUMondo data originate from a simulation based on the FAO 3 demand scenario (Eitelberg et al., 2016) and cover the time period from 2000 to 2040 with annual temporal resolution. Data are available at a 9.25 x 9.25 km regular grid (~5 arcminute) in an equal area projection and are based on the land system classification system described in van Asselen and Verburg (2012) (Table S6). In order to detect a particular algorithm, which is valid within a ca. 0.5 x 0.5 degree grid cell, the model output required several steps of preprocessing (Figure S2):

- Aggregation of the CLUMondo land systems legend and reclassification of each map following the PFT scheme of DGVMs to cropland, grassland, forest, and mosaics of them. We also kept the bare and artificial classes, since they would have confused the other classes otherwise (Table S6).
- Identification of grid cells with cropland expansion by overlaying maps of two subsequent time steps. Cropland expansion was identified as changes from any other class to the reclassified cropland class or changes from any other classes except than the reclassified cropland class to the reclassified mosaic cropland classes.
- Tracking of change trajectories, i.e., identification of classes that contributed to cropland expansion. The cropland expansion from the last step was used as a mask to keep only grid cells where cropland actually expanded between two time steps. This step yielded the information, which LULC type was converted to cropland (= 'contributing source').
- Aggregation to ca. 0.5 x 0.5 degree grid. This step yielded the proportion of new cropland that originates in a particular LULC type within each ca. 0.5 x 0.5 degree grid cell.
- Tracking how much of the original LULC type at t1 within a ca. 0.5 x 0.5 degree grid cell was converted to cropland in t2 (= 'available source').
- Division of 'contribution source' by 'available source'. By applying this step we could distinguish grid cells which did not contain a particular LULC type at t1 (division not defined) from grid cells where a particular LULC type was available, but not converted to cropland (division result equals 0).

As a result of the preprocessing we obtained maps, where each grid cell contained the fraction of the original LULC type at t1 that was converted to cropland in t2. Subsequently we searched across these maps for priority algorithms of LULCC within ca. 0.5 x 0.5 degree grid cells for decadal time steps following a set of rules (Figure S3). A grid cell was classified as

- UNDEFINED, if either forest or grassland were not available at t1. For these cells a classification was not possible, since it is not clear which source class was converted with higher priority. For example, if the grid cell only contains grassland at time t1, grassland is logically converted to cropland. However, a forest first algorithm would be also true for this grid cell (and just not executed, because there was no forest to convert). The mosaic class was excluded here, since even it is not available, all algorithms could be detected with the following rules.

- UNVEGETATED FIRST, if urban or bare classes in a grid cell were converted completely, while at the same time all other sources were available, but not or only partially converted. Additionally, grid cells where urban or bare classes were partially converted, while at the same time all other sources were available, but not converted.

- FOREST FIRST, if more than 90% of the available forest in a grid cell was converted to cropland, while at the same time grassland was available, but less than 90% of it was converted. Additionally, grid cells where less than 90% of the available forest was converted, while at the same time grassland or mosaic classes were available, but not converted.

- GRASSLAND FIRST, if more than 90% of the available grassland in a grid cell was converted to cropland, while at the same time forest was available, but less than 90% of it was converted. Additionally, grid cells where less than 90% of the available grassland was converted, while at the same time forest or mosaic classes were available, but not converted.

- PROPORTIONAL, if the mosaic class was converted, while at the same time grassland and forest were available, but not converted. Additionally, grid cells where the ratio of converted grassland and forest was between 0.5 and 1.5 were considered as an indicator for proportional reduction.

- COMPLEX, if at least forest and grassland were available as a source, but neither a preferential conversion nor a proportional conversion could be detected.

[Figure]

Figure S1: Preprocessing workflow of CLUMondo output for gross change analysis. Rectangles represent processing steps, parallelograms represent data. Grey shaded items emphasize aggregated data at ca. 0.5 x 0.5 degree resolution.

**Table S6: CLUMondo land system classification and reclassification to broader LULC types.**

| LS code | Land system name | Reclassification |
|---|---|---|
| 0 | Cropland; extensive with few livestock | Cropland |
| 1 | Cropland; extensive with bovines, goats & sheep | Cropland |
| 2 | Cropland; medium intensive with few livestock | Cropland |
| 3 | Cropland; medium intensive with bovines, goats & sheep | Cropland |
| 4 | Cropland; intensive with few livestock | Cropland |
| 5 | Cropland; intensive with bovines, goats & sheep | Cropland |
| 6 | Mosaic cropland and grassland with bovines, goats & sheep | Mosaic cropland/grassland |
| 7 | Mosaic cropland (extensive) and grassland with few livestock | Mosaic cropland/grassland |
| 8 | Mosaic cropland (medium intensive) and grassland with few livestock | Mosaic cropland/grassland |
| 9 | Mosaic cropland (intensive) and grassland with few livestock | Mosaic cropland/grassland |
| 10 | Mosaic cropland (extensive) and forest with few livestock | Mosaic cropland/forest |
| 11 | Mosaic cropland (medium intensive) and forest with few livestock | Mosaic cropland/forest |
| 12 | Mosaic cropland (intensive) and forest with few livestock | Mosaic cropland/forest |
| 13 | Dense forest | Forest |
| 14 | Open forest with few livestock | Forest |
| 15 | Mosaic grassland and forest | Mosaic grassland/forest |
| 16 | Mosaic grassland and bare | Grassland |
| 17 | Natural grassland | Grassland |
| 18 | Grassland with few livestock | Grassland |
| 19 | Grassland with bovines, goats and sheep | Grassland |
| 20 | Bare | Bare |
| 21 | Bare with few livestock | Bare |
| 22 | Peri-urban & villages | Urban |
| 23 | Urban | Urban |

[Figure]

**Figure S2: Preprocessing workflow of CLUMondo output for land-use change priority analysis. Rectangles represent processing steps, parallelograms represent data. Grey shaded items emphasize aggregated data at ca. 0.5 x 0.5 degree resolution.**

[Figure]

[Figure]

**Figure S3: Classification rules applied to each ca. 0.5 x 0.5 degree grid cell to identify a predominant reduction of a particular source LULC type.**

**S3 Additional Results**

**2000-2010**

[Figure]

**2010-2020**

[Figure]

[Figure]

[Figure]

**2020-2030**

34 %    41 %

9 %    4 %    12 %

**2030-2040**

Algorithm
- Undefined
- Forest first
- Grassland first
- Proportional
- Complex
- Land area

24 %    53 %

6 %

6 %

12 %

**Figure S4: Transitions from natural vegetation to cropland as shown by the CLUMondo model (FAO 3 demand scenario) from 2000 to 2040 in decadal time steps. Colored grid cells represent areas with at least 10 % of cropland expansion within a ca. 0.5 x 0.5 degree grid cell. Grid cells are classified to forest first (yellow), grassland first (cyan), proportional (magenta) and complex reduction (red) algorithm as described in the text (for details see SI). Black grid cells denote areas where the validity of none algorithm could -be detected.**

---

## Editor Decision (ED1)

**ESD-2016-39: Editor Decision**

December 28$^{th}$, 2016

Dear authors,

Thank you very much for your thoughtful responses to the reviewer concerns raised during the open discussion stage of the peer review process.

The open stage brought out a fertile discussion that is very well appreciated. Insightful, constructive reviews have been formulated raising valid concerns, and the authors aptly responded to those concerns with thoughtful arguments and pledges to substantially improve the manuscript − including revising the title, structure and relevant contents as suggested.

Aside from mutually agreed improvements to the manuscript, the authors may also wish to consider mentioning particular aspects of open debate, which are welcome to be reflected in the revised manuscript. For instance, the authors may provide brief mention of potential caveats and controversy in their statements (e.g. "it may be argued that ... however, we argue that ... based on ..."), along with further arguments justifying and supporting the authors' views. In doing so, the authors transform potential controversy from a caveat to an asset in the paper, aptly discussing open issues as has been done in the response to the reviewers.

Moreover, while further scientific technical innovation would always rank high in every readers' wishlist, the readers should be made aware that the submitted manuscript type entails a platform for concise, incisive and potentially controversial arguments to be presented (a "short communication", affine to an opinion paper). As such, the manuscript is essentially formulated as a communication of partial review and opinion/vision nature, with supporting technical (e.g. modelling, analysis) and literature evidence as deemed necessary to support the conveyed arguments. By further enhancing the exposition of the authors' visions and perspectives to tackle the raised challenges, the natural vocation of the manuscript will be further highlighted and strengthened.

Acknowledging the potential shown during the open discussion stage for a substantially improved manuscript to emerge, and in order to enable a thorough revision process along the aforementioned lines, my editorial decision entails *Major Revisions*.

If you have any further questions, please do not hesitate to contact us.

With very best wishes,

Rui Perdigão
(ESD Editor)

---

## Author Response (AR2)

Dear Dr. Perdigão,

Thank you very much again for the time and effort spent on our manuscript.

We revised the manuscript according to the minor revisions requested.

Please find below a detailed point-to-point reply to all individual comments and a marked-up version showing all changes eventually made to the manuscript. Page/line references in our response refer to the clean version of the revised manuscript.

With kind regards,

Reinhard Prestele

**Reviewer #1 (A. Di Vittorio)**

The authors have done a tremendous job of revising this manuscript. It is better organized, more clear and consistent, and overall a very good paper that explains, demonstrates, and proposes advances to three major challenges of global lulcc in the context of earth system modeling. The authors responded very well to all the reviewers' comments, and I don't have any major concerns with this version. I suggest some minor clarifications/edits below, and am looking forward to seeing this paper published soon.

Response: We thank the reviewer for the positive evaluation of our revised manuscript. We addressed all comments as detailed in the point-to-point reply below.

specific suggestions/comments:

Introduction

page 3, lines 13-16: This sentence is awkward. Should there be an "and" in place of the comma between "studies" and "reviews" (two serial clauses), or is there a third serial clause associated with "…in the context…" that needs a verb? It also may be helpful to state the three challenges here.

Response: Thanks for pointing to this. We rephrased the sentences for clarity (page 3, lines 14-17).

Challenge 1

page 3, line 26: "e.g." is not needed here. If you are concerned that this implies that nothing else can 'jump,' you can try something like "Corresponding jumps in carbon and nutrient…"

Response: Agreed. We replaced '*Consequently, jumps e.g. in carbon and nutrient pools…*' by '*Corresponding jumps in carbon and nutrient pools…*' (page 3, line 27)

page 4, line 5: Figure 1 doesn't really illustrate this point (it may work better for the next sentence). More appropriate would be table of the independent data sets (refer to Table S1 and maybe add the future projections to it?).

Response: We agree that the reference was misplaced and moved it to the subsequent sentence. We additionally included a reference to the supplementary information and Table S1. (page 4, line 5 & 7).

'*[…] and a variety of independent datasets at spatially explicit or world region level is provided to the user community (e.g. climate modeling) (see Supplement S1 and Table S1 for the example of the historical data).*'

page 4, lines 32-33: Awkward sentence. If it is 'within' the data, how is it not considered?

Response: We rephrased the sentence (page 4, lines 33-34):

'*Beyond this inarguable success, several uncertainties are to date not, or only partially, addressed in the LUH data.*'

page 5, line 20: "…amounts…"

Response: Changed accordingly (page 5, line 22).

page 6, line 2: "providing land-use data to climate models" isn't necessary here

Response: We removed '*providing land-use data to climate models*' as suggested (page 6, lines 3-4).

Challenge 2

page 7, lines 10-11: This more general definition does not make sense to me. Just because an area change is not included in a product does not mean it is a gross change. It could be that a particular category is not represented, which is different than a gross change.

Response: We agree that this statement was too vague. We added explanation to account for the case mentioned by the reviewer (i.e., if a category is not represented). It now reads (page 7, lines 11-13):

'A more general definition would include all area changes (i.e., gains and losses of all categories represented in a product) that are not depicted in land-use change products.'

page 8, line 6: Is there a context for this 40-year period? Is it based on current day statistics/assumptions? Is it a projection of historical baselines? Is it a future projection?

Response: We now explicitly mention that the 40-year period is based on the availability of data for this CLUMondo scenario, which simulates annual land system changes between 2000 and 2040 (page 8, lines 6-7).

'We tracked all changes between five land-use and land-cover categories (cropland, pasture, forest, urban, and bare) at the original resolution over the time period 2000 to 2040.'

Challenge 3

page 11, line 19: Is this the same simulation as above?

Response: It is indeed the same simulation. We rephrased the sentence to add this piece of information as following (page 11, line 22):

'[…] we reclassified the outputs of the same CLUMondo simulation utilized in section 3.2 […]'

page 13, lines 8-14: This seems incomplete, and the example given is likely not universal, and is not correct depending on the maps in question. First of all, there are more cases: exact match, lu ag on natural lc, lc ag on natural lu, and spatial combinations/redistributions of the latter two. It would seem that transition matrices would work best with the exact match, and would pose challenges for all other cases. But the result (in terms of which cover to convert) of applying of the matrices to the mismatched land cover in each cell would be dependent on the given transitions and land use/cover in each cell, even if using a single set of rules to apply the matrices (these rules will also have uncertainty).

Response: We agree with the reviewer. We rephrased the paragraph in a way that we still mention the issue (exact match vs. mismatched land cover/land use), but do not present a universal solution, which is indeed dependent on the background land cover, the land use data and the specific model in question (page 13, lines 11-19).

Recommendations

page 15, lines 4-5: This isn't necessarily the case if the DGVM is designed to accommodate integrated land cover and land use dynamics.

Response: We think the reviewer has a valid point here. At the same time, there are currently hardly any (if none?) DGVMs designed to accommodate integrated land cover and land use dynamics available. Moreover, in this section we still discuss the situation where data is passed between land-use change models and TBMs, i.e. no integrated land cover/use. We thus feel the statement is correct and decided not to change.

Outlook

page 15, lines 9-17: The introduction of the 'offline' strategy is a bit confusing here, especially since the next sentence does not explain it, but rather asks for an integrated framework. It sounds like you refer the 'ways forward' as the decoupled strategy, then want to discuss 'offline' coupling which needs to be defined here, all leading up to full integration. Maybe the topic sentence should present this progression, then the text can walk through each in turn. Currently, the "land use, land cover and the climate system…" sentence is out of place.

Response: We rephrased the paragraph to emphasize that we indeed aimed at presenting the progression from the 'offline' coupling to an integrated modeling framework (page 15, lines 12ff).

We discuss the recommended improvements in the 'offline' coupling in section 5, acknowledging that full integration will also take some time. In section 6 we then give an outlook what should be done, in our point of view, to improve the understanding of land use – climate interactions, i.e. model integration.

page 15, line 17: I recommend the terminology of "preliminary results of the iESM." The "first" results, based on a completed feedback coupling experiment, are still in in review.

Response: Changed as suggested (page 15, line 21).

**Reviewer #2 (Anonymous)**

The authors have made considerable improvements during the revision phase and the comments from my first review were addressed satisfactory. The current manuscript is now better structured and much clearer also to 'non-specialists' in this field. Based on my current review, I suggest minor revisions in which the comments below should be addressed before the paper could be accepted for publication.

Response: We thank the reviewer for the positive evaluation of our revised manuscript. We accommodated the additional comments raised by the reviewer as detailed in the point-to-point reply below.

General Comments:

I would like to ask the authors to make sure that it will be clear throughout the manuscript that the content of this paper is based on the authors' opinion (e.g. with regard to the challenges and reasons for the difficulties are and how to they could be overcome).

Response: We have rephrased some additional sentences clearly indicating that the suggested ways forward are perspectives of the authors (see examples below). Moreover, during the first revision of the manuscript, we put major efforts in highlighting the 'opinion' character of the manuscript (e.g., by the extensive usage of 'could', 'should' and 'might' constructions in section 5 and section 6). However, the identified problems and underlying reasons are based on literature review and analysis and not part of the authors' opinion.

Examples:

*'Simultaneously, we suggest that the land-use and remote-sensing communities should engage to reduce uncertainties in land-use and land-cover products by: […]*' (page 14, lines 10-11)

*'If not yet possible at the global scale […], we recommend the implementation of regional scale evaluation schemes using smaller scale, high accuracy remote sensing products as a starting point for later integration into global applications.'* (page 14, 20-23)

*'Rather than improving de-coupled data products and models on an individual basis and connecting them 'offline' through the exchange of files, we argue that land use, land cover and the climate system need to be studied in an integrated modeling framework.'* (page 15, lines 13-15)

In the manuscript, many different acronyms are being used which are difficult to recall throughout the paper. As the journal is aimed at an interdisciplinary audience, I suggest that the authors add a list of acronyms to the supplementary information and mention the existence of such a list in the introduction section of the paper.

Response: We do acknowledge that the use of acronyms might be difficult for the reader (but is essential for readability) and added a list to the SI as suggested by the reviewer (SI page 1, lines 1-19). Additionally, we refer to this list at the first time, when we introduce an acronym in the introduction (page 2, line 6).

*'Anthropogenic land-use and land-cover change (LULCC; for a list of acronyms used in the paper see Supplement S0) is a key cause […]'*

I suggest using Roman numerals throughout the text to identify the three Challenges and to distinguish the challenges from section numbers. E.g. The caption for the second P3 L19 would then read '2 Challenge I: Spatial explicit …'

Response: Good point. We changed the section headers accordingly. Additionally, Roman numerals are now used when listing the challenges in the abstract (page 1, lines 28 & 30).

Instead of just saying 'for details see SI' or similar when refereeing to the Supplementary Information, please provide the section number within the SI to make it easier for the reader too find the information.

Response: We apologize that we were not consistent in referencing the supplementary material throughout the manuscript. We adjusted all references to the SI by adding the section number (e.g., page 6, line 20: *'[…] (Figure 2, background map; Supplement S2.1)'*).

Specific Comments:

P1L23-25: Please split this long sentence.

Response: Done. We split the sentence into (page 1, lines 23-26):

*'Fully coupled models of climate, land use and biogeochemical cycles to explore land use – climate interactions across spatial scales are currently not available. Instead, information on land use is provided as exogenous data from the land-use change modules of Integrated Assessment Models (IAMs) to TBMs.'*

P1L29: suggest replacing 'due to' with 'associated with'.

Response: Changed accordingly (page 1, line 29).

P1L30-31: Can you provide some more elaboration on what the 'allocation strategy' entails, as this might not be clear from the abstract.

Response: Given the brief, concise character of the abstract and to keep the balance in the abstract we see no possibility to do this. We feel the statement summarizes the issue quite well in a short, concise way and we are explaining it in detail in the paper.

P2L17: Maybe add 'effected' before 'land'

Response: We thank the reviewer for the suggestion. However, the 'potential net source of GHGs' does not only refer to land affected by change, but to 'the land' as a whole. We therefore prefer not to change this sentence.

P2L21: 'the short history', please add details what 'short' entails (e.g. since when)

Response: We added *'(~10 years; Canadell et al., 2007)'* to the sentence to add detail to *'the short history'* (page 2, line 22).

P2L21: please add examples what 'external data' are.

Response: We added *'(e.g. maps of global cropland or pasture distribution)'* to the sentence (page 2, line 23).

P2L22: '… short history …' … '…along with…' … '…led to issues that render… ' … '… uncertain'. This sentence is not clear to me; how can the points mention 'render' something 'uncertain'. Please consider rephrasing.

Response: We rephrased the sentence for clarity (page 2, lines 22-25).

*'The short history [...], along with the need to include external data [...], have led to several issues that complicate the quantification of land-use change impacts on climate and biogeochemical cycles using TBMs.'*

P2L27: Pease specify what an ''offline' coupling' entails.

Response: We specify 'offline' coupling in the subsequent sentence and now connected the sentences to make this more clear (page 2, lines 28-31). 'Offline' coupling refers to the situation, where data is passed between different types of models (IAMs/LUCMs and TBMs) without feedbacks between the different types of models.

P3L27: 'reliably determining' please consider rephrasing

Response: We rephrased the sentence (page 3, lines 27-28). It now reads:

*'Corresponding jumps in carbon and nutrient pools in the transition period would distort legacy fluxes working on decadal to centennial time scale, rendering the simulations useless for the quantification of climate impacts'*

P4L23: 'Smoothly connected'; what does this mean, is this similar to Figure1? Please elaborate.

Response: We added a reference to Figure 1 (page 4, line 27). Indeed this figure is intended to visualize the 'smooth connection' between historical and future land use time series (schematically).

P4L26 'the harmonization'; which one? LUH?

Response: We added the word 'process' to clarify (page 4, line 27). 'Harmonization' describes the process of connecting the historical and future land use data in this sentence.

P4L32-32: … 'serves as a basis to implement anthropogenic impact on … Please consider rephrasing

Response: We rephrased the sentence to improve readability (page 4, lines 33-34). It now reads:

*'The harmonization ensured for the first time consistent land-use input for climate model intercomparisons and thus facilitated the implementation of anthropogenic impact on the land in climate models.'*

P5L23: Consider rephrasing the sentence.

Response: We rephrased the sentence aiming at improved readability into (page 5, lines 24-25):

*'[…], an inappropriate representation of the uncertainty about land-use history is likely to affect model outcomes regarding changes in local to regional climate.'*

P6L17: Please specify what 'cropland development' is.

Response: Cropland development refers to the dynamics of future cropland expansion / abandonment. We rephrased the sentence (page 6, lines 17-18). It now reads:

*'Model comparison further revealed that while land-use change models represent the future development of cropland area more consistently, […]'*

P6L19: How were these projections derived? Please clarify where they come from (i.e. reference or refer to section in SI)

Response: We provide the reference to the projections in the subsequent sentence (Prestele et al., 2016). Additionally, we now include a reference to the section in the SI, where we describe the analysis underlying Figure 2 (page 6, lines 20-21).

P7L31: 'meaning that …?... many are' I think there is a word missing.

Response: Agreed, thanks. We added '*TBMs*' to the sentence (page 7, line 32).

P9L10: replace 'larger' with 'longer'

Response: Changed as suggested (page 9, line 11).

P10L8: Please elaborate in more detail what 'through soils' means.

Response: We rephrased the sentence and added a reference to be more specific (page 10, lines 9-11).

*'[…] while cropland expanding on former grassland would have a less immediate impact on ecosystem carbon stocks due to the long time lag (years to centuries) for the resulting changes in soil carbon to be realized (Pugh et al., 2015).'*

P10L29: 'They'? Who? Gibbs? Please clarify.

Response: We replaced '*They*' by '*Gibbs et al. (2010)'* (page 10, line 31).

P11L6: Please elaborate in what these products are better.

Response: We admit that 'better products' was a very vague phrasing and replaced it with the following to be more specific (page 11, lines 8-9):

'*[…] though products with higher resolution (up to ~30m), more frequent temporal coverage, and increasing thematic detail are just emerging; Ban et al., 2015 […]'*

P11L15: What are 'neighborhood effects'?

Response: We agree that neighborhood effects were not sufficiently defined and added a short explanation (page 11, line 18). The sentence now reads:

'*[…] neighborhood effects (i.e., cropland expansion in a grid cell also depends on the availability of suitable land in the surrounding grid cells).'*

P12L4: replace 'complex interplays' with 'the complex interplay'

Response: Replaced as suggested (page 12, line 8).

P13L8: move the reference at the end of the sentence.

Response: We feel that moving the reference, which explicitly refers to the translation of CCI-LC categories to PFTs and not the background vegetation map per se, would rather confuse the reader. We therefore decided not to change this sentence.

P13L10: Please check the grammar of the sentence.

Response: We checked the grammar and rephrased the sentence according to our response to comment 'page 13, lines 8-14' of reviewer #1.

P13L20: To make things even more clear, I suggest restating the three sources again.

Response: We accommodated this comment by rephrasing the first part of the sentence and splitting it into (page 13, 23- 26):

'*As we have shown in section 2, three major sources of uncertainty, which include the uncertainty about land-use history, inconsistencies in present-day land-use estimates, and structural differences across IAMs and LUCMs, are poorly addressed through the almost exclusive implementation of the LUH dataset within the climate modeling community.'*

P13L22: replace 'impact' with 'influence' or 'change'

Response: We replaced '*impact*' by '*influence*' (page 13, line 26).

P13L24-26: Move the last part of the sentence to the beginning.

Response: Rephrased as suggested (page 13, lines 29-31).

P14L18: Remove 'our'

Response: Removed (page 14, line 21).

P14L26-18: Split sentence.

Response: We split the sentence into (page 14, lines 30-32):

*'Based on such analyses, multi-century reconstructions and projections for climate and ecosystem assessments could be enhanced for at least the satellite era. As models extend further into the past, the detailed information could be gradually replaced by model assumptions, supported by additional reference data such as historical maps and statistics.'*

P15L3: Please rephrase 'have to ensure to use'

Response: We replaced '*have to*' by '*must*' to increase readability (page 15, line 6).

P15L14: please replace 'very large level of uncertainties' with a more appropriate phrase.

Response: The sentence has been rephrased into (page 15, line 18):

*'[…] accumulating an increasing level of uncertainty along the modeling chain.'*

Figures:
Figure1: Please provide information for the interdisciplinary audience how the spatial patterns are being distorted. Maybe with a schematic showing grid cells or discuss more explicit in the text.

Response: We agree that this statement might be confusing and thus removed it from the figure caption (page 28, line 5). As we discuss in the text (page 4, line 27), the harmonization 'tries to conserve the spatial pattern of the IAM', which however cannot fully be achieved due to the different land-use pattern in the historical and future maps (as discussed in section 2). How and to what extent a 'distortion of the spatial pattern' happens depends strongly on the IAM maps in question. To properly describe and quantify, one would need to compare the original IAM maps with the LUH maps, which we believe is outside the scope of this paper.

Figure2: due to the different values of the Initial variations it is impossible to properly compare the values shown in the right bar plots for different regions. To make the bar plots comparable I suggest adding the % in the categories or normalising the values shown on the right by the initial value before generating the bar plots on the right.

Response: We improved the caption for clarity (page 29, lines 2ff), but do not think that adding % labels to the bar plot would increase the readability of the figure. The purpose of the figure is to highlight the broad patterns of the variation (as discussed in section 2.3; page 6, lines 17-27), rather than comparing discrete values of individual regions.

The reviewer is absolutely correct that the initial variation varies across the regions, as well as the total variation does. Thus, we use relative values to compare the contribution of different variance components across regions. The purpose of the right bar plots is mainly to visualize the distribution of model, scenario and residual components in regions where their summed contribution is small (e.g., Canada, USA, Australia,…). They are comparable across regions in a way that they represent the % of the components model, scenario and residual on the part of variation that is not explained by the initial variation.

Figure 3: add to caption that this is based on singe model.

Response: We accommodated this comment by adding *'[…] one realization of a single LUCM (CLUMondo; FAO 3 demand scenario)'* to the caption (page 30, lines 2-3).

Figure 4: To make the values of the individual bars comparable, I suggest showing annual averages instead of sums over periods of different length.

Response: The reviewer has a valid point that the time periods of NLCD and CORINE differ in their length. However, we think the suggested changes would be misleading. We present the sum of changes (in terms of area) only to indicate that the total agricultural expansion is different across products and time periods where change products are available. At the same time, we present the relative contribution of the sources to these total agricultural expansion, which we also use for 'comparison' in the text (page 10, lines 18 ff). Showing annual averages would add detail to the data, we actually cannot obtain from CORINE and NLCD. Moreover, such a processing would not change the relative contribution of the individual sources (since we do not have annual information), which we refer to in our analysis.

Supplementary Information Section

P5L14: Please add reference to the 'Akaike information criterion'.

Response: We added the following reference (SI page 6, line 14):

*Akaike, H.: Information theory and an extension of the maximum likelihood principle, in Second International Symposium on Information Theory, edited by B. N. Petrov and B. F. Csaki, pp. 267–281, Budapest., 1973.*

P7L16: Please add correct Reference.

Response: Figure S1 depicts the workflow described in this paragraph. We thus believe it is the correct reference. Additionally, we added Figure 3 as a reference, which presents the final product of the analysis in the main text of the paper (SI page 8, line 16).

P15L1, P26Tabel3 and P18FigureS3: The category 'unvegetated fist' is not shown in Figure 5 and Figure S4. Please either add this category to the Figures or give detailed reasons why this is not shown

Response: The reviewer is absolutely correct, that we did not include the 'unvegetated first' category in Figures 5 and S4. Thanks for pointing to the missing explanation. We added explanation accordingly to the figure captions (page 34, line 5; SI page 23, line 5). The reason for not including it into the figure is that the share of this category is negligible small (< 0.1 %) and would thus not add any useful information.

P18FigureS3: Please use same colours for the classifications as in Figure 5 and Figure S4.

Response: We changed the colors accordingly (SI page 19).

[revised manuscript text omitted]

**Supplementary Information**

**S1 Overview of historical land use reconstructions**

Several approaches have been published within the last two decades to reconstruct the history of human utilization of land to meet their needs of food, fiber and space for settlement on a global scale. Depending on the objective of the particular study they cover different time periods, spatial resolutions and methods of reconstruction (Table S1). In the following paragraphs we summarize the methodologies of four spatially explicit historical reconstructions. For details, please see the original publications.

*HYDE*

The History Database of the Global Environment (HYDE) was originally developed by Klein Goldewijk (2001), covering spatially explicit historical population estimates and land-use patterns for the past 300 years at 0.5° resolution. Several updates and extensions led to version HYDE 3.1, which was used for the Land Use Harmonization (LUH) in CMIP5 (Klein Goldewijk et al. (2011); this is the version we refer to here and in the article). Recently there has been a update to version HYDE 3.2, which now covers a time period from 10 000 BC to 2015 AD at 5 arcminute spatial resolution and includes further agricultural management layers (such as irrigation) (Klein Goldewijk, 2016).

The underlying principle of the HYDE reconstruction is the relationship between human population and agricultural activity expressed in a per capita use of cropland and pasture area, leading to a spatial dependency of land-use activities to human settlements. Klein Goldewijk et al. (2010) first derived time series of population numbers from a vast number of sources on a subnational or national scale (depending on data availability, e.g. McEvedy and Jones (1978), Livi Bacci (2007) and Maddison (2001); see Klein Goldewijk (2001) and Klein Goldewijk et al. (2011) for details) and translated them to population density maps using patterns from Landscan (2006) for recent time and a combination of suitability maps for historic time. For the period 1961-2000, the per capita use of cropland and pasture was calculated from FAO statistics on country or subnational level. Prior to 1961 the per capita land use numbers were dynamically estimated country by country following Ruddiman and Ellis (2009) and adjusted accounting for low population numbers (= higher per capita land use), but also limitations in technology and a maximum area of land that can be cultivated by a subsistence farmer (= lower per capita land use). Using the per capita usage of cropland and pasture to estimate cropland and pasture total areas on a (sub-)national level for every time step, spatial allocation of the total areas to the 5 arcminute grid was implemented using two sets of weighing maps: On the one side, present distribution of cropland and pasture was derived by integrating FAO statistics and additional subnational statistics for the USA and China with two satellite derived land cover products representative for recent time (DISCover version 2, Loveland et al. (2000); GLC2000, Bartholome and Belward (2005)). The weighing map for historical time, on the other side, was constructed by combining the earlier described population density maps and different biophysical suitability parameters, namely soil quality, distance to rivers, steepness of terrain and thresholds for annual mean temperature. Both maps were subsequently used to allocate (sub-) national totals of agricultural areas to specific grid cells, while the influence of the historic map gradually increases when going further into the past.

**Ramankutty and Foley (1999)**

Ramankutty and Foley (1999) apply a hindcast modeling technique to derive spatial patterns of cropland on a global scale for the period 1700-1992. The original reconstruction did not inlcude pasture areas. A revised and updated version[1] covers the years up to 2007 both for cropland and pasture at 5 arcminute spatial resolution. The starting point for the reconstruction is
* * *
[1] The updated version is based on the global cropland and pasture maps published in Ramankutty et al. (2008) and the methodology described in Ramankutty et al. (1999). There was, however, no additional publication related to the updated dataset. The dataset was available from http://www.geog.mcgill.ca/nramankutty/Datasets/Datasets.html, but the webpage has been recently removed.

represented by the integration of satellite derived land cover products (DISCover in original data set (Loveland and Belward, 1997); BU-MODIS (Friedl et al., 2002) and GLC2000 (Bartholome and Belward) 2005 in the updated version) and FAO statistics. The national and subnational totals of cropland and pasture were calibrated to the spatial distribution of cropland and pasture areas in the earth observation product applying a linear fitting approach. This resulted in a global 5 arcminute resolution cropland and pasture map for the year 2000, representing the spatial distribution of cropland and pasture areas (Ramankutty et al., 2008). In a second step, a comprehensive data base of historical agricultural areas on (sub-)national level was compiled from different sources. FAO statistics were used for the time period from 1961 to the end point. Prior to 1961 the data base first accounts for census data. Whenever census data were not available, cropland conversion rates of Houghton and Hackler (1995) were applied to the cropland map of Richards (1990) for 1980 with some regional adjustments to avoid unrealistic agricultural areas in particular regions. The spatial allocation of the cropland areas is implemented by applying a simple hindcast model, which preserves the cropland pattern of the start map within each unit of the inventory data base for the whole time period to 1700. For that a change factor between two subsequent years is calculated from the inventory database, dividing the cropland area in the target year by the cropland area in the starting year, which is thereafter applied to each grid cell within a unit.

**Pongratz et al. (2008)**

Pongratz et al. (2008) extended the reconstruction of Ramankutty and Foley (1999) back to 800 AD and presented the first consistent and spatially explicit cropland and pasture reconstruction for pre-industrial times at the date of publication. For the period 1700-1992 the cropland time series is, apart from smaller regional adjustments and updates, the same than the Ramankutty and Foley (1999) data. Since they further had not published their pasture time series at that point, Pongratz et al. (2008) combined the pasture map for 1992 with change rates taken from the HYDE data base to extend it back to 1700. Unlike the pattern maintaining approach applied by Ramankutty and Foley (1999), pasture was spatially distributed around existing cropland while maintaining the pattern of total agricultural area rather than the individual shares of cropland and pasture to allow also for cropland expansion into pasture areas.

Based on this two time series covering the years 1700-1992, an extrapolation to 800 AD was applied on (sub)-national level, while using population data from McEvedy and Jones (1978) as a proxy for land-use change. Similar to HYDE, the simple measure of per capita usage of crop and pasture area was assumed to be the best approximation. However, in this case, per capita use was calculated from the 1700 maps and held constant for the whole period prior to 1700. Spatial distribution of agricultural areas was assumed to represent the patterns of 1700 for the period 800 to 1700. Besides, changes in agricultural patterns, e.g. following the European colonization in North and South America, were especially accounted for by altering the patterns in particular regions. Both time series were aggregated to a 0.5° resolution.

**KK10**

Kaplan et al. (2010) introduce a non-linear relationship between population numbers and area of forest clearance to calculate total areas affected by human land-use change. The basic assumption of this approach is a decreasing per capita land use over time due to intensification of already converted areas rather than expand land use to new areas when population densities increase. With the objective to build an empirical, non-linear model, population time series for the period 6050 BC to AD 1850 were compiled first. Data from McEvedy and Jones (1978) were utilized for the period 1000 BC to AD 1850 with some regional adjustments and subsequently extended back to 6050 BC by a modelling approach (Global Land USE and Technological Evolution Simulator (GLUES, see Lemmen (2009) and Wirtz and Lemmen (2003) for details). Population density was normalized to cultivatable land to prevent the model extending cropland areas into unsuitable land. A sigmoidal log-linear model was fitted to a set of empirical data from various European countries to derive a relationship between forest cover and population density accounting also for different stages of technological development over time (Kaplan et al., 2009). Concurrently, Kaplan et al. (2010) integrated different climatic and biophysical variables to indices of suitability for cropland and pasture on a 5 arcminute grid following a method of Ramankutty et al. (2002). Combining the regional level estimates of historical forest cover with the suitability datasets led to a spatially explicit representation of area affected by land-use change over time. The integration was done by allocating cropland to high quality and suitable areas first, followed by pasture. As the forest cover – population relationship originally was derived for Europe, it has been adjusted for tropical and boreal regions in the global approach by including a threshold of net primary production, where productivity of agricultural lands is higher and therefore demand for new land lower.

**Table S1: Summary of historical LULCC reconstructions**

| Reference | Spatial resolution | Temporal coverage and resolution | Input data | Allocation |
|---|---|---|---|---|
| KK10, Kaplan et al. (2010) | 5 x 5 arcminute | 6050 BC to AD 1850, annual | population estimates, land suitability maps | based on non-linear population density – forest clearance relationship, high quality land cleared first |
| HYDE 3.1, Klein Goldewijk et al. (2011) | 5 x 5 arcminute | 10 000 BC to AD 2005, variable resolution | population estimates, FAO statistics, satellite derived products | dynamic per capita use of cropland and pasture; combination of weighing maps derived from satellite products, population and environmental parameters |
| Pongratz et al. (2008) | 0.5 x 0.5 degree | AD 800 – AD 1992 | adjusted Ramankutty and Foley (1999), HYDE 2.0, population data | constant per capita use of cropland & pasture prior to 1700, constant spatial pattern of agriculture prior to 1700 |
| Ramankutty and Foley (1999) | 5 x 5 arcminute | AD 1700 – AD 1992; update AD 1700 – 2007 | census data and estimates of agricultural area, FAO statistics, satellite derived products | hindcast model, preserving agricultural pattern of 1992 within aggregated units |

**S2 Data and Methods**

Several data and methods have been used to support our arguments in the manuscript and create the related tables and figures. To ensure readability we decided to provide methodological details in the Supplementary Information rather than in the main text of the manuscript. In the following we provide an overview of the data used, details of the data processing and how

5  analysis was conducted. In each section heading we indicate the relation to the main text and the figures and tables that were derived from individual steps of analysis.

**S2.1 Attribution of uncertainty in land use change projections (Section 2; Figure 2)**

Multiple linear regression analysis followed by an ANOVA was used to decompose the variability of 43 projections of regional pasture areas for the year 2030 simulated by 11 global scale IAMs and LUCMs (Alexander et al., 2016; Prestele et al., 2016).

10  Every individual projection has been parameterized according to 9 variables (Table S2) that characterize the model structure (model type classification, model resolution), the scenario (socioeconomic and climate scenario variables) and the initial condition (deviation of absolute pasture area from value reported by FAOSTAT (2015) in the year 2010) prior to the regression analysis. The modeled pasture area in 2030 was assumed to be a function of these 9 variables. To balance performance and complexity of the resulting regression model, variables were rejected using the Akaike information criterion

15  (Akaike, 1973). Subsequently an ANOVA was conducted on the regression results to identify relative contribution of the variables to the total variation in the regression model of the 2030 pasture areas. The type II[2] sum of squares were calculated for each variable and divided by the total sum of squares. Subsequently, the relative contributions of the individual variables were summarized according to the grouping in Table S2. The residual term thus covers all variation that could not be explained by these 9 variables.
* * *
[2] Type II sum of squares have been used since they are not dependent on the order in which the variables are considered in the model, which has been shown suitable for unbalanced data as in our analysis (Langsrud, 2003). See Alexander et al. (2016) for details.

**Table S2: Overview of variables used in the regression analysis and ANOVA (table adopted and modified according to Prestele et al. (2016))**

| Variable | Data type | Group |
|---|---|---|
| Initial condition delta | Continuous (deviation of model areas from FAO areas in 2010 (FAOSTAT, 2015) | Initial |
| Model type | Categorical (CGE, PE, Rule-based, Hybrid) | Model |
| Number of model cells (log) | Continuous | Model |
| $CO_2$ concentration 2100 | Continuous | Scenario |
| Population 2100 | Continuous | Scenario |
| GDP growth rate to 2100 | Continuous | Scenario |
| Inequality ratio 2100 | Continuous | Scenario |
| Technology change | Discrete (0=None, 1=Slow, 2=Medium, 3=Rapid) | Scenario |
| International trade | Discrete (1=Constrained, 2=Moderate, 3=High) | Scenario |

**S2.2 Derivation of gross vs. net changes due to re-gridding from a CLUMondo simulation (Section 3; Figure 3)**

To identify the difference between net and gross changes due to re-gridding of high-resolution modeled land-use change information, we utilized data from a simulation of the CLUMondo model (Van Asselen and Verburg, 2013) based on the FAO 3 demand scenario (Eitelberg et al., 2016). These data are available at a 9.25 x 9.25 km regular grid (~5 arcminute) in an equal area projection and are based on the land system classification described in van Asselen and Verburg (2012). Land systems are characterized by land-cover composition, livestock numbers and land-use intensity. Each grid cell can thus be expressed as a mosaic of five LULC types (cropland, grassland, forest, urban, and bare) which varies with the world region. Upon a change from one land system to another, these characteristics also change.

We used the fractions of these five LULC types to track areal changes per grid cell at the original 9.25 x 9.25 km resolution over the whole simulation period (2000-2040). The total area changed at this resolution (sum of gains and losses for each LULC type) was assumed to be the gross changes in our analysis. In a second step, we aggregated the maps to ca. 0.5 x 0.5 degree and calculated the changes between two time steps. Due to bi-directional changes at the higher resolution (which offset each other) the total area affected by change at 0.5 x 0.5 degree resolution is usually smaller. The areal changes at 0.5 x 0.5 degree resolution were assumed to be the net changes in our analysis. By adding up the net changes and gross changes across all five LULC types and over the whole simulation period, we identified the amount of actually changed area that would be missed in a net change representation at 0.5 x 0.5 degree for this simulation (**Error! Reference source not found.**; Figure 3).

[Figure]

**Figure S1: Preprocessing workflow of CLUMondo output for gross change analysis. Rectangles represent processing steps, parallelograms represent data. Grey shaded items emphasize aggregated data at ca. 0.5 x 0.5 degree resolution.**

**S2.3 Analysis of remote sensing products (Section 4; Figure 4; Table 2)**

To derive dominant sources of cropland expansion from remote sensing products, we analyzed high resolution LULCC data from Europe (CORINE, 100 m spatial resolution) and North America (NLCD, 30 m spatial resolution) (Table S3).
We downloaded CORINE data from http://land.copernicus.eu/pan-european/corine-land-cover. NLCD data were obtained through http://www.mrlc.gov/.

**S2.3.1 Data: CORINE**

CORINE was produced by computer assisted visual interpretation of satellite images processed on a country-by-country basis and subsequently merged to a comprehensive European database (EEA, 2007). It covers the years 1990, 2000, 2006 and most recently 2012 with different number of participating countries leading to different overlapping areas between the years. The land-cover classification was derived from different sensors dependent on the final year of the product (1990: Landsat-4/5 TM single date, 2000: Landsat-7 ETM single date; 2006: SPOT-4 and/or IRS P6 LISS III dual date; 2012: IRS P6 LISS III and RapidEye dual date). CORINE is provided at a spatial resolution of 100 m and 250 m in raster data format as well as in vector format. The minimum mapping unit is 25 ha. Besides the products for the years mentioned above, special LULCC products have been produced and are currently available for the periods 1990 to 2000 and 2000 to 2006. For the change products an enhanced minimum mapping unit of 5 ha was applied. The change products have been used for derivation of agricultural transitions in our analysis, thus covering all changes to agricultural areas larger than 5 ha between start and end year. All CORINE products are accompanied by a three level land-use and land-cover nomenclature varying in detail across the levels (Table S4). The first level only provides very general classes (e.g. artificial surfaces; agricultural areas; forests; etc.). The second level distinguishes 15 different categories and the highest detail is given by the 44 classes at level 3. For our analysis we used a merger of the different levels, as e.g. forests and shrubland could be only differentiated at level 2, while natural grassland could be only identified at level 3 (Table S4). See Bossard et al. (2000) for a detailed description of the legend and distinction of individual classes. Although CORINE provides a consistent framework of European land cover mapping, uncertainties in the final products are necessarily apparent. For example, the country-by-country processing of data can introduce uncertainty due to different treatment of the individual legend items during visual interpretation of the satellite imagery. However, clearly defined mapping guidelines aim to minimize these effects (Bossard et al., 2000). Moreover, the minimum mapping unit of 5 ha (in case of the change product that was used in our analysis) ignores changes on smaller areas. Thus, additional uncertainty can be introduced in areas where less changes appear. The thematic accuracy of the 2000 to 2006 change product is indicated with larger than 85%, while the accuracy for the 1990 to 2000 change product has not been assessed (see http://land.copernicus.eu/pan-european/corine-land-cover). Thematic accuracy entails the capability of CORINE land cover maps to represent the 'true' land-cover class as compared to an independent validation dataset (EEA, 2006). Although

these uncertainties may propagate into our analysis of cropland transition trajectories (Table 2, Figure 4), we do not expect them to substantially change the order of source LULCC categories at the aggregated European scale.

**S2.3.2 Data: NLCD**

The National Land Cover Database (NLCD) is a high resolution (30 m) land-cover product for the USA. This Landsat derived product has been provided for the years 1992, 2001, 2006 and 2011 at the latest. For our analysis the 2001, 2006 and 2011 products have been considered, as they are provided in a harmonized collection with special change products. The NLCD dataset is classified according to a 16-class land-cover classification for the United States, developed in the 1970s by Anderson et al. (1976). The classification system distinguishes two agricultural classes, *(81) Pasture/Hay* and *(82) Cultivated Crops* (Table S5). Stehman et al. (2003) report an accuracy level of 55.7 % for the 1992 dataset. Accuracy assessment is not yet available for the 2011 data, but as 2001 and 2006 data showed significantly improved accuracy levels (78.7 % and 78.0 %, Wickham et al. (2010) and Wickham et al. (2013)) a similar (or even better) quality can be assumed for the 2011 data.

**Table S3: Summary of land-cover products used for our analysis**

| Product | Temporal coverage | Spatial resolution / Coverage | Legend | Sensor | Classification |
|---|---|---|---|---|---|
| CORINE | 1990, 2000, 2006, (2012) | 100m / Europe | 44 classes, 3 hierarchical levels | Landsat-4/5 TM, Landsat-7 ETM, SPOT-4, IRS P6 LISS III, RapidEye | change product, supervised, expert knowledge |
| NLCD | (1992), 2001, 2006, 2011 | 30m / USA | 16 classes | Landsat | change product, spectral and knowledge based change detection |

**S2.3.3 Change detection**

We used the dedicated change products for our analysis, which hold information about source and target classes upon land-use change. Areas of agricultural expansion were identified by every pixel that has an agricultural label (based on the inherent legend) at time t2, but not at time t1. We calculated the total expansion of agricultural areas by the difference of pixels which were assigned an agricultural label at time t2 and time t1. Subsequently, combining the areas of cropland expansion with the map of time t 1 resulted in a map of sources of agricultural area. The source maps were classified and summarized considering the underlying original legend into grassland, forest, mixed grassland/forest and unvegetated land origin (Table S4, Table S5).

**Table S4: CORINE land-cover legend (Bossard et al., 2000) and aggregation applied in our analysis**

| Level 1 | Level 2 | Level 3 | Aggregation |
|---|---|---|---|

| | | | |
|---|---|---|---|
| (1) Artificial surfaces | (11) Urban fabric; (12) Industrial, commercial and transport units; (13) Mine, dump and construction sites; (14) Artificial, non-agricultural vegetated areas | (111) Continuous urban fabric; (112) Discountinuous urban fabric; (121) Industrial and commercial units; (122) Road and rail networks and associated land; (123) Port areas; (124) Airports; (131) Mineral extraction sites; (132) Dump sites; (133) Construction sites; (141) Green urban areas; (142) Sport and leisure facilities | Other |
| (2) Agricultural areas | (21) Arable land; (22) Permanent crops; (23) Pastures; (24) Hegerogeneous agricultural areass | (211) Non-irrigated arable land; (212) Permanently irrigated land; (213) Rice fields; (221) Vineyards; (222) Fruit trees and berry plantations; (223) Olive groves; (231) Pastures; (241) Annual cropas associated with permanent crops; (242) Complex cultivation patterns; (243) Land principally occupied by agriculture, with significant areas of natural vegetation; (244) Agro-forestry areas | Agricultural areas |
| (3) Forest and semi natural areas | (31) Forests; (32) Scrub and/or herbaceous vegetation associations; (33) Open spaces with little or no vegetation | (311) Broad-leaved forest; (312) Coniferous forest; (313) Mixed forest; (321) Natural grasslands; (322) Moors and heathland; (323) Sclerophyllous vegetation; (324) Transitional woodland-shrub; (331) Beaches, dunes, sands; (332) Bare rocks; (333) Sparsely vegetate areas; (334) Burnt areas; (335) Glaciers and perpetual snow | (311)-(313) Forest (321) Grassland (322)-(324) Shrubland (331)-(335) Other |
| (4) Wetlands | (41) Inland wetlands; (42) Maritime wetlands | (411) Inland marshes; (412) Peat bogs; (421) Salt marshes; (422) Salines; (423) Intertidal flats | Other |
| (5) Water bodies | (51) Inland waters; (52) Marine waters | (511) Water courses; (512) Water bodies; (521) Coastal lagoons; (522) Estuaries; (523) Sea and ocean | Other |

**Table S5: National Land Cover Database (NLCD) classification system according to Anderson et al. (1976) and aggregation applied in our analysis**

| Value | Label | Description | Aggregation |
|---|---|---|---|
| 11 | Open Water | All areas of open water, generally with less than 25 % cover or vegetation or soil | Other |
| 12 | Perennial Ice/Snow | All areas characterized by a perennial cover of ice and/or snow, generally greater than 25 % of total cover | Other |
| 21 | Developed, Open Space | Includes areas with a mixture of some constructed materials, but mostly vegetation in the form of lawn grasses. Impervious surfaces account for less than 20 % of total cover. These areas most commonly include large-lot single-family housing units, parks, golf courses, and vegetation planted in developed settings for recreation, erosion control, or aesthetic purposes | Other |
| 22 | Developed, Low Intensity | Includes areas with a mixture of constructed materials and vegetation. Impervious surfaces account for 20-49 % of total cover. These areas most commonly include single-family housing units. | Other |
| 23 | Developed, Medium Intensity | Includes areas with a mixture of constructed materials and vegetation. Impervious surfaces account for 50-79 % of the total cover. These areas most commonly include single-family housing units. | Other |
| 24 | Developed, High Intensity | Includes highly developed areas where people reside or work in high numbers. Examples include apartment complexes, row houses and commercial/industrial. Impervious surfaces account for 80-100 % of the total cover. | Other |
| 31 | Barren Land (Rock/Sand/Clay) | Barren areas of bedrock, desert pavement, scarps, talus, slides, volcanic material, glacial debris, sand dunes, strip mines, gravel pits and other accumulations of earthen material. Generally, vegetation accounts for less than 15 % of total cover. | Other |
| 41 | Deciduous Forest | Areas dominated by trees generally greater than 5 meters tall, and greater than 20 % of total vegetation cover. More than 75 % of the tree species shed foliage simultaneously in response to seasonal change. | Forest |
| 42 | Evergreen Forest | Areas dominated by trees generally greater than 5 meters tall, and greater than 20 % of total vegetation cover. More than 75 % of the tree species maintain their leaves all year. Canopy is never without green foliage. | Forest |
| 43 | Mixed Forest | Areas dominated by trees generally greater than 5 meters tall, and greater than 20 % of total vegetation cover. Neither deciduous nor evergreen species are greater than 75 % of total tree cover. | Forest |
| 52 | Shrub/Scrub | Areas dominated by shrubs; less than 5 meters tall with shrub canopy typically greater than 20 % of total vegetation. This class includes true shrubs, young trees in an early successional stage or trees stunted from environmental conditions. | Shrubland |
| 71 | Grassland/Herbaceous | Areas dominated by grammanoid or herbaceous vegetation, generally greater than 80 % of total vegetation. These areas are not subject to intensive management such as tilling, but can be utilized for grazing. | Grassland |
| 81 | Pasture/Hay | Areas of grasses, legumes, or grass-legume mixtures planted for livestock grazing or the production of seed or hay crops, typically on a perennial cycle. Pasture/hay vegetation accounts for greater than 20 % of total vegetation. | Pasture |
| 82 | Cultivated Crops | Areas used for the production of annual crops, such as corn, soybeans, vegetables, tobacco, and cotton, and also perennial woody crops such as | Cropland |

| | | orchards and vineyards. Crop vegetation accounts for greater than 20 % of total vegetation. This class also includes all land being actively tilled. | |
|---|---|---|---|
| 90 | Woody Wetlands | Areas where forest or shrub land vegetation accounts for greater than 20 % of vegetative cover and the soil or substrate is periodically saturated with or covered with water. | Other |
| 95 | Emergent Herbaceous Wetlands | Areas where perennial herbaceous vegetation accounts for greater than 80 % of vegetative cover and the soil or substrate is periodically saturated with or covered with water. | Other |

**S2.4 CLUMondo land-use change priority analysis (Section 4; Figure 5)**

The CLUMondo data originate from a simulation based on the FAO 3 demand scenario (Eitelberg et al., 2016) and cover the time period from 2000 to 2040 with annual temporal resolution. Data are available at a 9.25 x 9.25 km regular grid (~5 arcminute) in an equal area projection and are based on the land system classification system described in van Asselen and Verburg (2012) (Table S6). In order to detect a particular algorithm, which is valid within a ca. 0.5 x 0.5 degree grid cell, the model output required several steps of preprocessing (Figure S2):

- Aggregation of the CLUMondo land systems legend and reclassification of each map following the PFT scheme of DGVMs to cropland, grassland, forest and mosaics of them. We also kept the bare and artificial classes, since they would have confused the other classes otherwise (Table S6).

- Identification of grid cells with cropland expansion by overlaying maps of two subsequent time steps. Cropland expansion was identified as changes from any other class to the reclassified cropland class or changes from any other classes except than the reclassified cropland class to the reclassified mosaic cropland classes.

- Tracking of change trajectories, i.e. identification of classes that contributed to cropland expansion. The cropland expansion from the last step was used as a mask to keep only grid cells where cropland actually expanded between two time steps. This step yielded the information, which LULC type was converted to cropland (= 'contributing source').

- Aggregation to ca. 0.5 x 0.5 degree grid. This step yielded the proportion of new cropland that originates in a particular LULC type within each ca. 0.5 x 0.5 degree grid cell.

- Tracking how much of the original LULC type at t1 within a ca. 0.5 x 0.5 degree grid cell was converted to cropland in t2 (= 'available source').

- Division of 'contribution source' by 'available source'. By applying this step we could distinguish grid cells which did not contain a particular LULC type at t1 (division not defined) from grid cells where a particular LULC type was available, but not converted to cropland (division result equals 0).

As a result of the preprocessing we obtained maps, where each grid cell contained the fraction of the original LULC type at t1 that was converted to cropland in t2. Subsequently we searched across these maps for priority algorithms of LULCC within ca. 0.5 x 0.5 degree grid cells for decadal time steps following a set of rules (Figure S3). A grid cell was classified as

- UNDEFINED, if either forest or grassland were not available at t1. For these cells a classification was not possible, since it is not clear which source class was converted with higher priority. For example, if the grid cell only contains grassland at time t1, grassland is logically converted to cropland. However, a forest first algorithm would be also true for this grid cell (and just not executed, because there was no forest to convert). The mosaic class was excluded here, since even it is not available all algorithms could be detected with the following rules.

- UNVEGETATED FIRST, if urban or bare classes in a grid cell were converted completely, while at the same time all other sources were available, but not or only partially converted. Additionally, grid cells where urban or bare classes were partially converted, while at the same time all other sources were available, but not converted.

- FOREST FIRST, if more than 90% of the available forest in a grid cell was converted to cropland, while at the same time grassland was available, but less than 90% of it was converted. Additionally, grid cells where less than 90% of the available forest was converted, while at the same time grassland or mosaic classes were available, but not converted.

- GRASSLAND FIRST, if more than 90% of the available grassland in a grid cell was converted to cropland, while at the same time forest was available, but less than 90% of it was converted. Additionally, grid cells where less than 90% of the available grassland was converted, while at the same time forest or mosaic classes were available, but not converted.

- PROPORTIONAL, if the mosaic class was converted, while at the same time grassland and forest were available, but not converted. Additionally, grid cells where the ratio of converted grassland and forest was between 0.5 and 1.5 were considered as an indicator for proportional reduction.

- COMPLEX, if at least forest and grassland were available as a source, but neither a preferential conversion nor a proportional conversion could be detected.

**Table S6: CLUMondo land system classification and reclassification to broader LULC types**

| LS code | Land system name | Reclassification |
|---------|------------------|------------------|
| 0 | Cropland; extensive with few livestock | Cropland |
| 1 | Cropland; extensive with bovines, goats & sheep | Cropland |
| 2 | Cropland; medium intensive with few livestock | Cropland |
| 3 | Cropland; medium intensive with bovines, goats & sheep | Cropland |
| 4 | Cropland; intensive with few livestock | Cropland |
| 5 | Cropland; intensive with bovines, goats & sheep | Cropland |
| 6 | Mosaic cropland and grassland with bovines, goats & sheep | Mosaic cropland/grassland |
| 7 | Mosaic cropland (extensive) and grassland with few livestock | Mosaic cropland/grassland |
| 8 | Mosaic cropland (medium intensive) and grassland with few livestock | Mosaic cropland/grassland |
| 9 | Mosaic cropland (intensive) and grassland with few livestock | Mosaic cropland/grassland |
| 10 | Mosaic cropland (extensive) and forest with few livestock | Mosaic cropland/forest |
| 11 | Mosaic cropland (medium intensive) and forest with few livestock | Mosaic cropland/forest |
| 12 | Mosaic cropland (intensive) and forest with few livestock | Mosaic cropland/forest |
| 13 | Dense forest | Forest |
| 14 | Open forest with few livestock | Forest |
| 15 | Mosaic grassland and forest | Mosaic grassland/forest |
| 16 | Mosaic grassland and bare | Grassland |
| 17 | Natural grassland | Grassland |
| 18 | Grassland with few livestock | Grassland |
| 19 | Grassland with bovines, goats and sheep | Grassland |
| 20 | Bare | Bare |
| 21 | Bare with few livestock | Bare |
| 22 | Peri-urban & villages | Urban |
| 23 | Urban | Urban |

[Figure]

**Figure S2: Preprocessing workflow of CLUMondo output for land-use change priority analysis. Rectangles represent processing steps, parallelograms represent data. Grey shaded items emphasize aggregated data at ca. 0.5 x 0.5 degree resolution.**

[Figure]

[Figure]

**Figure S3: Classification rules applied to each ca. 0.5 x 0.5 degree grid cell to identify a predominant reduction of a particular source LULC type.**

**2000-2010**

[Figure]

**2010-2020**

**2020-2030**

[Figure]

**2030-2040**

**Figure S4: Transitions from natural vegetation to cropland as shown by the CLUMondo model (FAO 3 demand scenario) from 2000 to 2040 in decadal time steps. Colored grid cells represent areas with at least 10 % of cropland expansion within a ca. 0.5 x 0.5 degree grid cell. Grid cells are classified to forest first (yellow), grassland first (cyan), proportional (magenta) and complex reduction (red) algorithm as described in the text (for details see SI). Black grid cells denote areas where the validity of none algorithm could be detected. Grid cells classified to unvegetated first (Table 3) are not shown due to very small contribution (< 0.1 %).**

---

## Editor Decision (ED2)

**ESD-2016-39R: Editor Decision**

March 29$^{th}$, 2017

Dear Authors,

Thank you very much for all your efforts and diligence in carefully preparing a thoroughly improved manuscript, taking into account the concerns raised in the peer-review process, and providing comprehensive responses to the evaluation reports.

The referees that kindly re-evaluated the manuscript acknowledged the vast improvements and were supportive of publication subject to minor revisions.

Having analysed the revised version, the author responses and referee reports, I will welcome the manuscript for publication at Earth System Dynamics subject to addressing the recommendations raised in the evaluation reports.

With very best wishes,

Rui Perdigão
(ESD Editor)